# Distance-Restricted Folklore Weisfeiler-Leman GNNs with Provable Cycle Counting Power

**Junru Zhou**[1]   **Jiarui Feng**[2]   **Xiyuan Wang**[1]   **Muhan Zhang**[1]

[1]Institute for Artificial Intelligence, Peking University
[2]Department of CSE, Washington University in St. Louis
zml72062@stu.pku.edu.cn, feng.jiarui@wustl.edu,
{wangxiyuan,muhan}@pku.edu.cn

## Abstract

The ability of graph neural networks (GNNs) to count certain graph substructures, especially cycles, is important for the success of GNNs on a wide range of tasks. It has been recently used as a popular metric for evaluating the expressive power of GNNs. Many of the proposed GNN models with provable cycle counting power are based on subgraph GNNs, i.e., extracting a bag of subgraphs from the input graph, generating representations for each subgraph, and using them to augment the representation of the input graph. However, those methods require heavy preprocessing, and suffer from high time and memory costs. In this paper, we overcome the aforementioned limitations of subgraph GNNs by proposing a novel class of GNNs—$d$-Distance-Restricted FWL(2) GNNs, or $d$-DRFWL(2) GNNs, based on the well-known FWL(2) algorithm. As a heuristic method for graph isomorphism testing, FWL(2) colors all node pairs in a graph and performs message passing among those node pairs. In order to balance the expressive power and complexity, $d$-DRFWL(2) GNNs simplify FWL(2) by restricting the range of message passing to node pairs whose mutual distances are at most $d$. This way, $d$-DRFWL(2) GNNs exploit graph sparsity while avoiding the expensive subgraph extraction operations in subgraph GNNs, making both the time and space complexity lower. We theoretically investigate both the discriminative power and the cycle counting power of $d$-DRFWL(2) GNNs. Our most important finding is that $d$-DRFWL(2) GNNs have provably strong cycle counting power even with $d = 2$: they can count all 3, 4, 5, 6-cycles. Since 6-cycles (e.g., benzene rings) are ubiquitous in organic molecules, being able to detect and count them is crucial for achieving robust and generalizable performance on molecular tasks. Experiments on both synthetic datasets and molecular datasets verify our theory. To the best of our knowledge, 2-DRFWL(2) GNN is the most efficient GNN model to date (both theoretically and empirically) that can count up to 6-cycles.

## 1 Introduction

Graphs are important data structures suitable for representing relational or structural data. As a powerful tool to learn node-level, link-level or graph-level representations for graph-structured data, graph neural networks (GNNs) have achieved remarkable successes on a wide range of tasks [54, 61]. Among the various GNN models, Message Passing Neural Networks (MPNNs) [27, 36, 50, 55] are a widely adopted class of GNNs. However, the expressive power of MPNNs has been shown to be upper-bounded by the Weisfeiler-Leman test [55]. More importantly, MPNNs even fail to detect some simple substructures (e.g., 3, 4-cycles) [33], thus losing great structural information in graphs.

37th Conference on Neural Information Processing Systems (NeurIPS 2023).

The weak expressive power of MPNNs has aroused a search for more powerful GNN models. To evaluate the expressive power of these GNN models, there are usually two perspectives. One is to characterize their discriminative power, i.e., the ability to distinguish between non-isomorphic graphs. It is shown in [14] that an equivalence exists between distinguishing all non-isomorphic graph pairs and approximating any permutation-invariant functions on graphs. Although the discriminative power partially reveals the function approximation ability of a GNN model, it fails to tell *what specific functions* a model is able to approximate. Another perspective is to directly characterize *what specific function classes* a model can approximate. In particular, we can study the expressive power of a GNN model by asking *what graph substructures it can count*. This perspective is often more practical since the task of counting substructures (especially cycles) is closely related to a variety of domains such as chemistry [18, 35], biology [37], and social network analysis [34].

Despite the importance of counting cycles, the task gets increasingly difficult as the length of cycle increases [2].To have an intuitive understanding of this difficulty, we first discuss why MPNNs fail to count any cycles. The reason is clear: MPNNs only keep track of the rooted subtree **around each node** [55], and without node identity, any neighboring node $v$ of $u$ has no idea which of its neighbors are also neighbors of $u$. Therefore, MPNNs cannot count even the simplest 3-cycles. In contrast, the 2-dimensional Folklore Weisfeiler-Leman, or FWL(2) test [12], uses *2-tuples of nodes* instead of nodes as the basic units for message passing, and thus natually **encodes closed walks**. For example, when a 2-tuple $(u, v)$ receives messages from 2-tuples $(w, v)$ and $(u, w)$ in FWL(2), $(u, v)$ **gets aware of the walk** $u \to w \to v \to u$, whose length is $d(u, w) + d(w, v) + d(u, v) =: l$. As long as the intermediate nodes do not overlap, we immediately get an $l$-cycle passing both $u$ and $v$. In fact, FWL(2) can provably count up to 7-cycles [5, 25].

However, FWL(2) has a space and time complexity of $O(n^2)$ and $O(n^3)$, which makes it impractical for large graphs and limits its real-world applications. Inspired by the above discussion, we naturally raise a question: **to what extent can we retain the cycle counting power of FWL(2) while reducing the time and space complexity?** Answering this question requires another observation: **cycle counting tasks are intrinsically *local***. For example, the number of 6-cycles that pass a given node $u$ has nothing to do with the nodes that have a shortest-path distance $\geqslant 4$ to $u$. Therefore, if we only record the embeddings of *node pairs with mutual distance $\leqslant d$* in FWL(2), where $d$ is a fixed positive integer, then the ability of FWL(2) to count substructures with diameter $\leqslant d$ should be retained. Moreover, given that most real-world graphs are sufficiently sparse, this simplified algorithm runs with complexities much lower than FWL(2), since we only need to store and update a small portion of the $n^2$ embeddings for all 2-tuples. We call this simplified version $d$-**Distance-Restricted FWL(2)**, or $d$-**DRFWL(2)**, reflecting that only 2-tuples with *restricted* distances get updated.

**Main contributions.** Our main contributions are summarized as follows:

1. We propose $d$-DRFWL(2) tests as well as their neural versions, $d$-DRFWL(2) GNNs. We study how the hyperparameter $d$ affects the discriminative power of $d$-DRFWL(2), and show that a **strict expressiveness hierarchy** exists for $d$-DRFWL(2) GNNs with increasing $d$, both theoretically and experimentally.

2. We study the cycle counting power of $d$-DRFWL(2) GNNs. Our major results include:

    - 2-DRFWL(2) GNNs can already count up to 6-cycles, covering common structures like benzene rings in organic chemistry. The time and space complexities of 2-DRFWL(2) are $O(n \deg^4)$ and $O(n \deg^2)$ respectively, making it the **most efficient one to date** among other models with 6-cycle counting power such as $I^2$-GNN [33].

    - With $d \geqslant 3$, $d$-DRFWL(2) GNNs can count up to 7-cycles, **fully retaining** the cycle counting power of FWL(2) but with complexities **strictly lower** than FWL(2). This finding also confirms the existence of GNNs with the same cycle counting power as FWL(2), but strictly weaker discriminative power.

3. We compare the performance and empirical efficiency of $d$-DRFWL(2) GNNs (especially for $d = 2$) with other state-of-the-art GNNs on both synthetic and real-world datasets. The results verify our theory on the counting power of $d$-DRFWL(2) GNNs. Additionally, for the case of $d = 2$, the amount of GPU memory required for training 2-DRFWL(2) GNNs is greatly reduced compared with subgraph GNNs like ID-GNN [57], NGNN [59] and $I^2$-GNN [33]; the preprocessing time and training time of 2-DRFWL(2) GNNs are also much less than $I^2$-GNN.

## 2 Preliminaries

### 2.1 Notations

For a simple, undirected graph $G$, we use $\mathcal{V}_G$ and $\mathcal{E}_G$ to denote its node set and edge set respectively. For every node $v \in \mathcal{V}_G$, we define its *k-th hop neighbors* as $\mathcal{N}_k(v) = \{u \in \mathcal{V}_G : d(u,v) = k\}$, where $d(u,v)$ is the shortest-path distance between nodes $u$ and $v$. We further introduce the symbols $\mathcal{N}_{\leqslant k}(v) = \bigcup_{i=1}^{k} \mathcal{N}_i(v)$ and $\mathcal{N}(v) = \mathcal{N}_1(v)$. For any $n \in \mathbb{N}$, we denote $[n] = \{1, 2, \ldots, n\}$.

An $\ell$-*cycle* ($\ell \geqslant 3$) in the (simple, undirected) graph $G$ is a sequence of $\ell$ edges $\{v_1, v_2\}, \{v_2, v_3\}, \ldots, \{v_\ell, v_1\} \in \mathcal{E}_G$ with $v_i \neq v_j$ for any $i \neq j$ and $i, j \in [\ell]$. The $\ell$-cycle is said to pass a node $v$ if $v$ is among the nodes $\{v_i : i \in [\ell]\}$. An $\ell$-*path* is a sequence of $\ell$ edges $\{v_1, v_2\}, \{v_2, v_3\}, \ldots, \{v_\ell, v_{\ell+1}\} \in \mathcal{E}_G$ with $v_i \neq v_j$ for any $i \neq j$ and $i, j \in [\ell + 1]$, and it is said to start at node $v_1$ and end at node $v_{\ell+1}$. An $\ell$-*walk* from $v_1$ to $v_{\ell+1}$ is a sequence of $\ell$ edges $\{v_1, v_2\}, \{v_2, v_3\}, \ldots, \{v_\ell, v_{\ell+1}\} \in \mathcal{E}_G$ but the nodes $v_1, v_2, \ldots, v_{\ell+1}$ can coincide. An $\ell$-*clique* is $\ell$ nodes $v_1, \ldots, v_\ell$ such that $\{v_i, v_j\} \in \mathcal{E}_G$ for all $i \neq j$ and $i, j \in [\ell]$.

Let $\mathcal{G}$ be the set of all simple, undirected graphs. If $S$ is a graph substructure on $\mathcal{G}$, and $G \in \mathcal{G}$ is a graph, we use $C(S, G)$ to denote the number of inequivalent substructures $S$ that occur as subgraphs of $G$. Similarly, if $u$ is a node of $G$, we use $C(S, u, G)$ to denote the number of inequivalent substructures $S$ that pass node $u$ and occur as subgraphs of $G$.

Following [15, 33], we give the definition of *whether a function class $\mathcal{F}$ on graphs can count a certain substructure $S$*.

**Definition 2.1** (Graph-level count). Let $\mathcal{F}_{\text{graph}}$ be a function class on $\mathcal{G}$, i.e., $f_{\text{graph}} : \mathcal{G} \to \mathbb{R}$ for all $f_{\text{graph}} \in \mathcal{F}_{\text{graph}}$. $\mathcal{F}_{\text{graph}}$ is said to be able to *graph-level count* a substructure $S$ on $\mathcal{G}$ if for $\forall G_1, G_2 \in \mathcal{G}$ such that $C(S, G_1) \neq C(S, G_2)$, there exists $f_{\text{graph}} \in \mathcal{F}_{\text{graph}}$ such that $f_{\text{graph}}(G_1) \neq f_{\text{graph}}(G_2)$.

**Definition 2.2** (Node-level count). Let $\mathcal{G} \times \mathcal{V} = \{(G, u) : G \in \mathcal{G}, u \in \mathcal{V}_G\}$. Let $\mathcal{F}_{\text{node}}$ be a function class on $\mathcal{G} \times \mathcal{V}$, i.e., $f_{\text{node}} : \mathcal{G} \times \mathcal{V} \to \mathbb{R}$ for all $f_{\text{node}} \in \mathcal{F}_{\text{node}}$. $\mathcal{F}_{\text{node}}$ is said to be able to *node-level count* a substructure $S$ on $\mathcal{G}$ if for $\forall (G_1, u_1), (G_2, u_2) \in \mathcal{G} \times \mathcal{V}$ such that $C(S, u_1, G_1) \neq C(S, u_2, G_2)$, there exists $f_{\text{node}} \in \mathcal{F}_{\text{node}}$ such that $f_{\text{node}}(G_1, u_1) \neq f_{\text{node}}(G_2, u_2)$.

Notice that $C(S, G)$ can be calculated by $\sum_{u \in \mathcal{V}_G} C(S, u, G)$ divided by a factor only depending on $S$, for any given substructure $S$. (For example, for triangles the factor is 3.) Therefore, counting a substructure at node level is harder than counting it at graph level.

### 2.2 FWL($k$) graph isomorphism tests

**WL(1) test.** The 1-dimensional Weisfeiler-Leman test, or WL(1) test, is a heuristic algorithm for the graph isomorphism problem [52]. For a graph $G$, the WL(1) test iteratively assigns a color $W(v)$ to every node $v \in \mathcal{V}_G$. At the 0-th iteration, the color $W^{(0)}(v)$ is identical for every node $v$. At the $t$-th iteration with $t \geqslant 1$,

$$W^{(t)}(v) = \text{HASH}^{(t)}\left(W^{(t-1)}(v), \text{POOL}^{(t)}\left(\{\!\{W^{(t-1)}(u) : u \in \mathcal{N}(v)\}\!\}\right)\right), \tag{1}$$

where $\text{HASH}^{(t)}$ and $\text{POOL}^{(t)}$ are injective hashing functions, and $\{\!\{\cdot\}\!\}$ means a multiset (set with potentially identical elements). The algorithm stops when the node colors become stable, i.e., $\forall v, u \in \mathcal{V}_G, W^{(t+1)}(v) = W^{(t+1)}(v) \Leftrightarrow W^{(t)}(v) = W^{(t)}(u)$. We denote the *stable coloring* of node $v$ as $W^{(\infty)}(v)$; then the representation for graph $G$ is

$$W(G) = \text{READOUT}\left(\{\!\{W^{(\infty)}(v) : v \in \mathcal{V}_G\}\!\}\right), \tag{2}$$

where READOUT is an arbitrary injective multiset function.

**FWL($k$) tests.** For $k \geqslant 2$, the $k$-dimensional Folklore Weisfeiler-Leman tests, or FWL($k$) tests, define a hierarchy of algorithms for graph isomorphism testing, as described in [12, 32, 42]. For a graph $G$, the FWL($k$) test assigns a color $W(\mathbf{v})$ for every *k-tuple* $\mathbf{v} = (v_1, \ldots, v_k) \in \mathcal{V}_G^k$. At the 0-th iteration, the color $W^{(0)}(\mathbf{v})$ is the *atomic type* of $\mathbf{v}$, denoted as $\text{atp}(\mathbf{v})$. If we denote the

subgraphs of $G$ induced by $\mathbf{v}$ and $\mathbf{v}'$ as $G(\mathbf{v})$ and $G(\mathbf{v}')$ respectively, then the function $\mathrm{atp}(\cdot)$ can be any function on $\mathcal{V}_G^k$ that satisfies the following condition: $\mathrm{atp}(\mathbf{v}) = \mathrm{atp}(\mathbf{v}')$ iff the mapping $v_i \mapsto v_i', i \in [k]$ (i.e., the mapping that maps each $v_i$ to its corresponding $v_i'$, for $i \in [k]$) induces an isomorphism from $G(\mathbf{v})$ to $G(\mathbf{v}')$.

At the $t$-th iteration with $t \geqslant 1$, FWL($k$) updates the color of $\mathbf{v} \in \mathcal{V}_G^k$ as

$$W^{(t)}(\mathbf{v}) = \mathrm{HASH}^{(t)}\left(W^{(t-1)}(\mathbf{v}), \mathrm{POOL}^{(t)}\left(\{\!\{\mathrm{sift}(W^{(t-1)}, \mathbf{v}, w) : w \in \mathcal{V}_G\}\!\}\right)\right), \qquad (3)$$

where $\mathrm{sift}(f, \mathbf{v}, w)$ is defined as

$$\mathrm{sift}(f, \mathbf{v}, w) = (f(\mathbf{v}[1 \to w]), f(\mathbf{v}[2 \to w]), \ldots, f(\mathbf{v}[k \to w])). \qquad (4)$$

Here we use $\mathbf{v}[j \to w]$ to denote $(v_1, \ldots, v_{j-1}, w, v_{j+1}, \ldots, v_k)$ for $j \in [k]$ and $w \in \mathcal{V}_G$.

$\mathrm{HASH}^{(t)}$ and $\mathrm{POOL}^{(t)}$ are again injective hashing functions. The algorithm stops when all $k$-tuples receive stable colorings. The stable coloring of $\mathbf{v} \in \mathcal{V}_G^k$ is denoted as $W^{(\infty)}(\mathbf{v})$. We then calculate the representation for graph $G$ as

$$W(G) = \mathrm{READOUT}\left(\{\!\{W^{(\infty)}(\mathbf{v}) : \mathbf{v} \in \mathcal{V}_G^k\}\!\}\right), \qquad (5)$$

where READOUT is an arbitrary injective multiset function.

## 3 $d$-Distance-Restricted FWL(2) GNNs

In this section, we propose the $d$-Distance Restricted FWL(2) tests/GNNs. They use 2-tuples like FWL(2), but restrict the distance between nodes in each 2-tuple to be $\leqslant d$, which effectively reduces the number of 2-tuples to store and aggregate while still retaining great cycle counting power.

### 3.1 $d$-DRFWL(2) tests

We call a 2-tuple $(u, v) \in \mathcal{V}_G^2$ a **distance-$k$ tuple** if the shortest-path distance between $u$ and $v$ is $k$ in the following. **$d$-Distance Restricted FWL(2) tests**, or **$d$-DRFWL(2) tests**, assign a color $W(u, v)$ for *every distance-$k$ tuple $(u, v)$ with $0 \leqslant k \leqslant d$*. Initially, the color $W^{(0)}(u, v)$ only depends on $d(u, v)$. For the $t$-th iteration with $t \geqslant 1$, $d$-DRFWL(2) updates the colors using the following rule,

For each $k = 0, 1, \ldots, d$,

$$W^{(t)}(u, v) = \mathrm{HASH}_k^{(t)}\left(W^{(t-1)}(u, v), \left(M_{ij}^{k(t)}(u, v)\right)_{0 \leqslant i, j \leqslant d}\right), \quad \text{if } d(u, v) = k, \quad (6)$$

where $\mathrm{HASH}_k^{(t)}$ is an injective hashing function for distance $k$ and iteration $t$, and $M_{ij}^{k(t)}(u, v)$ is defined as

$$M_{ij}^{k(t)}(u, v) = \mathrm{POOL}_{ij}^{k(t)}\left(\{\!\{\left(W^{(t-1)}(w, v), W^{(t-1)}(u, w)\right) : w \in \mathcal{N}_i(u) \cap \mathcal{N}_j(v)\}\!\}\right). \quad (7)$$

The symbol $\left(M_{ij}^{k(t)}(u, v)\right)_{0 \leqslant i, j \leqslant d}$ stands for $\left(M_{00}^{k(t)}(u, v), M_{01}^{k(t)}(u, v), \ldots, M_{0d}^{k(t)}(u, v), \ldots, M_{dd}^{k(t)}(u, v)\right)$. Each of the $\mathrm{POOL}_{ij}^{k(t)}$ with $0 \leqslant i, j, k \leqslant d$ is an injective multiset hashing function. Briefly speaking, the rules (6) and (7) **update the color of a distance-$k$ tuple $(u, v)$ using colors of distance-$i$ and distance-$j$ tuples**.

When every distance-$k$ tuple $(u, v)$ with $0 \leqslant k \leqslant d$ receives its stable coloring, denoted as $W^{(\infty)}(u, v)$, the representation of $G$ is calculated as

$$W(G) = \mathrm{READOUT}\left(\{\!\{W^{(\infty)}(u, v) : (u, v) \in \mathcal{V}_G^2 \text{ and } 0 \leqslant d(u, v) \leqslant d\}\!\}\right). \qquad (8)$$

Figure 1 illustrates how 2-DRFWL(2) updates the color of a distance-2 tuple $(u, v)$. Since only distance-$k$ tuples with $0 \leqslant k \leqslant 2$ are colored in 2-DRFWL(2) tests, there are 7 terms of the form $((W(w, v), W(u, w))$ (with $w \in \{u, v, x, y, z, t, r\}$) contributing to the update of $W(u, v)$. The 7 nodes $u, v, x, y, z, t, r$ are filled with different colors, according to their distances to $u$ and to $v$. For example, the *violet* node $u$ has distance 0 to $u$ and distance 2 to $v$, thus contributing to $M_{02}^2(u, v)$; the *green* nodes $x$ and $y$ have distance 1 to either $u$ or $v$, thus contributing to $M_{11}^2(u, v)$. Analogously, nodes with *red, blue, orange* and *pink* colors contribute to $M_{20}^2(u, v), M_{12}^2(u, v), M_{21}^2(u, v)$ and $M_{22}^2(u, v)$, respectively. Finally, the uncolored nodes $q$ and $s$ do not contribute to the update of $W(u, v)$, since

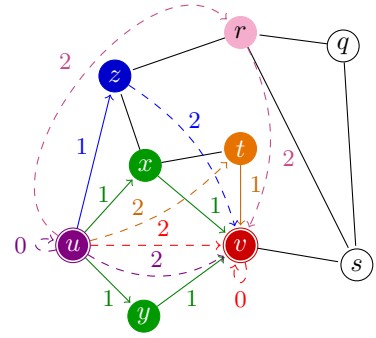

Figure 1: Neighbor aggregation in 2-DRFWL(2) for 2-tuple $(u, v)$.

they have distance 3 (which is greater than 2) to $u$. From the figure, we can observe that by redefining neighbors and sparsifying 2-tuples of FWL(2), $d$-DRFWL(2) significantly reduces the complexity and focuses only on local structures, especially on sparse graphs.

Now we study the expressive power of $d$-DRFWL(2) tests by comparing them with the WL hierarchy. First, we can prove that $d$-DRFWL(2) tests are strictly more powerful than WL(1), for every $d$.

**Theorem 3.1.** *In terms of the ability to distinguish between non-isomorphic graphs, the d-DRFWL(2) test is strictly more powerful than WL(1), for any $d \geqslant 1$.*

Next, we compare $d$-DRFWL(2) tests with FWL(2). Actually, since it is easy for the FWL(2) test to compute distance between every pair of nodes (we can initialize $W(u, v)$ as 0, 1 and $\infty$ for $u = v$, $(u, v) \in \mathcal{E}_G$ and all other cases, and iteratively update $W(u, v)$ with $\min\{W(u, v), \min_w\{W(u, w) + W(w, v)\}\}$), the FWL(2) test can use its update rule to simulate (6) and (7) by applying different $\text{HASH}_k^{(t)}$ and $\text{POOL}_{ij}^{k(t)}$ functions to different $(i, j, k)$ values. This implies that **$d$-DRFWL(2) tests are at most as powerful as the FWL(2) test**. Actually, this hierarchy in expressiveness is strict, due to the following theorem.

**Theorem 3.2.** *In terms of the ability to distinguish between non-isomorphic graphs, FWL(2) is strictly more powerful than d-DRFWL(2), for any $d \geqslant 1$. Moreover, $(d + 1)$-DRFWL(2) is strictly more powerful than d-DRFWL(2).*

The proofs of all theorems within this section are included in Appendix B.

### 3.2 $d$-**DRFWL(2) GNNs**

Based on the $d$-DRFWL(2) tests, we now propose $d$-DRFWL(2) GNNs. Let $G$ be a graph which can have node features $f_v \in \mathbb{R}^{d_f}, v \in \mathcal{V}_G$ and (or) edge features $e_{uv} \in \mathbb{R}^{d_e}, \{u, v\} \in \mathcal{E}_G$. A **$d$-DRFWL(2) GNN** is defined as a function of the form

$$f = M \circ R \circ L_T \circ \sigma_{T-1} \circ \cdots \circ \sigma_1 \circ L_1. \tag{9}$$

The input of $f$ is the *initial labeling* $h_{uv}^{(0)}, 0 \leqslant d(u, v) \leqslant d$. Each $L_t$ with $t = 1, 2, \ldots, T$ in (9) is called a **$d$-DRFWL(2) GNN layer**, which transforms $h_{uv}^{(t-1)}$ into $h_{uv}^{(t)}$ using rules

For each $k = 0, 1, \ldots, d$,

For each $(u, v) \in \mathcal{V}_G^2$ with $d(u, v) = k$,

$$a_{uv}^{ijk(t)} = \bigoplus_{w \in \mathcal{N}_i(u) \cap \mathcal{N}_j(v)} m_{ijk}^{(t)} \left( h_{wv}^{(t-1)}, h_{uw}^{(t-1)} \right), \tag{10}$$

$$h_{uv}^{(t)} = f_k^{(t)} \left( h_{uv}^{(t-1)}, \left( a_{uv}^{ijk(t)} \right)_{0 \leqslant i, j \leqslant d} \right), \tag{11}$$

where $m_{ijk}^{(t)}$ and $f_k^{(t)}$ are arbitrary learnable functions; $\bigoplus$ denotes a permutation-invariant aggregation operator (e.g., sum, mean or max). (10) and (11) are simply counterparts of (7) and (6) with $\text{POOL}_{ij}^{k(t)}$ and $\text{HASH}_k^{(t)}$ replaced with continuous functions. $\sigma_t, t = 1, \ldots, T - 1$ are entry-wise activation functions. $R$ is a permutation-invariant readout function, whose input is the multiset

$\{\!\!\{h_{uv}^{(T)} : (u,v) \in \mathcal{V}_G^2 \text{ and } 0 \leqslant d(u,v) \leqslant d\}\!\!\}$. Finally, $M$ is an MLP that acts on the graph representation output by $R$.

We can prove that (i) the representation power of any $d$-DRFWL(2) GNN is upper-bounded by the $d$-DRFWL(2) test, and (ii) there exists a $d$-DRFWL(2) GNN instance that has equal representation power to the $d$-DRFWL(2) test. We leave the formal statement and proof to Appendix B.

## 4   The cycle counting power of $d$-DRFWL(2) GNNs

By Theorem 3.2, the expressive power of $d$-DRFWL(2) tests (or $d$-DRFWL(2) GNNs) strictly increases with $d$. However, the space and time complexities of $d$-DRFWL(2) GNNs also increase with $d$. On one hand, since there are $O(n \deg^k)$ distance-$k$ tuples in a graph $G$, at least $O(n \deg^k)$ space is necessary to store the representations for all distance-$k$ tuples. For $d$-DRFWL(2) GNNs, this results in **a space complexity of** $O(n \deg^d)$. On the other hand, since there are at most $O(\deg^{\min\{i,j\}})$ nodes in $\mathcal{N}_i(u) \cap \mathcal{N}_j(v)$, $\forall 0 \leqslant i,j \leqslant d$, there are at most $O(\deg^d)$ terms at the RHS of (10). Therefore, it takes $O(d^2 \deg^d)$ time to update a single representation vector $h_{uv}^{(t)}$ using (10) and (11). This implies **the time complexity of $d$-DRFWL(2) GNNs is** $O(nd^2 \deg^{2d})$.

For scalability, $d$-DRFWL(2) GNNs with a relatively small value of $d$ are used in practice. **But how to find the $d$ value that best strikes a balance between expressive power and efficiency?** To answer this question, we need a *practical, quantitative* metric of expressive power. In the following, we characterize the *cycle counting* power of $d$-DRFWL(2) GNNs. We find that 2-DRFWL(2) GNNs are powerful enough to **node-level count up to 6-cycles**, as well as **many other useful graph substructures**. Since for $d = 2$, the time and space complexities of $d$-DRFWL(2) GNNs are $O(n \deg^4)$ and $O(n \deg^2)$ respectively, our model is **much more efficient** than I$^2$-GNN which requires $O(n \deg^5)$ time and $O(n \deg^4)$ space to count up to 6-cycles [33]. Moreover, with $d \geqslant 3$, $d$-DRFWL(2) GNNs are able to **node-level count up to 7-cycles**, already matching the cycle counting power of FWL(2).

Our main results are stated in Theorems 4.3–4.8. Before we present the theorems, we need to give revised definitions of $C(S,u,G)$ for some certain substructures $S$. This is because in those substructures not all nodes are structurally equal.

**Definition 4.1.** If $S$ is an $\ell$-path with $\ell \geqslant 2$, $C(S,u,G)$ is defined to be the number of $\ell$-paths in $G$ *starting from* node $u$.

Figure 2a illustrates Definition 4.1 for the $\ell = 4$ case.

**Definition 4.2.** The substructures in Figures 2b, 2c and 2d are called *tailed triangles*, *chordal cycles* and *triangle-rectangles*, respectively. If $S$ is a tailed triangle (or chordal cycle or triangle-rectangle), $C(S,u,G)$ is defined to be the number of tailed triangles (or chordal cycles or triangle-rectangles) that occur as subgraphs of $G$ and include node $u$ at a position shown in the figures.

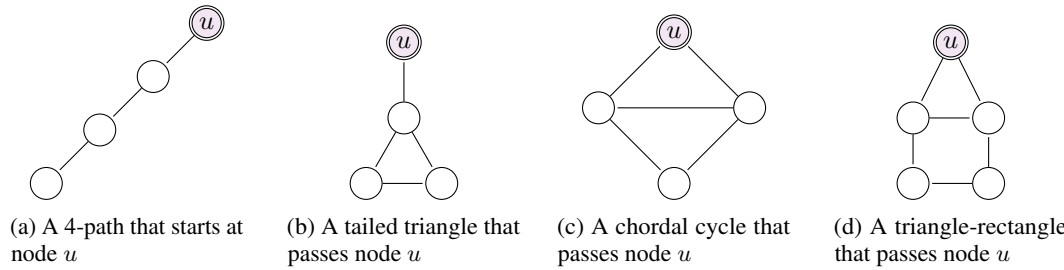

(a) A 4-path that starts at node $u$

(b) A tailed triangle that passes node $u$

(c) A chordal cycle that passes node $u$

(d) A triangle-rectangle that passes node $u$

Figure 2: Illustrations of node-level counts of certain substructures.

Now we state our main theorems. In the following, the definition for "node-level counting" is the same as Definition 2.2, but one should treat $C(S,u,G)$ differently (following Definitions 4.1 and 4.2) when $S$ is a path, a tailed triangle, a chordal cycle, or a triangle-rectangle. To define the output of a $d$-DRFWL(2) GNN $f$ on domain $\mathcal{G} \times \mathcal{V}$, we denote

$$f(G,u) = h_{uu}^{(T)}, \quad u \in \mathcal{V}_G, \tag{12}$$

i.e., we treat the embedding of $(u, u)$ as the representation of node $u$. For 1-DRFWL(2) GNNs, we have

**Theorem 4.3.** *1-DRFWL(2) GNNs can node-level count 3-cycles, but cannot graph-level count any longer cycles.*

For 2-DRFWL(2) GNNs, we investigate not only their cycle counting power, but also their ability to count many other graph substructures. Our results include

**Theorem 4.4.** *2-DRFWL(2) GNNs can node-level count 2, 3, 4-paths.*

**Theorem 4.5.** *2-DRFWL(2) GNNs can node-level count 3, 4, 5, 6-cycles.*

**Theorem 4.6.** *2-DRFWL(2) GNNs can node-level count tailed triangles, chordal cycles and triangle-rectangles.*

**Theorem 4.7.** *2-DRFWL(2) GNNs cannot graph-level count $k$-cycles with $k \geqslant 7$ or $k$-cliques with $k \geqslant 4$.*

For $d$-DRFWL(2) GNNs with $d \geqslant 3$, we have

**Theorem 4.8.** *For any $d \geqslant 3$, $d$-DRFWL(2) GNNs can node-level count 3, 4, 5, 6, 7-cycles, but cannot graph-level count any longer cycles.*

The proofs of all theorems within this section are included in Appendix C. To give an intuitive explanation for the cycle counting power of $d$-DRFWL(2) GNNs, let us consider, e.g., why 2-DRFWL(2) GNNs can count up to 6-cycles. The key reason is that **they allow a distance-2 tuple $(u, v)$ to receive messages from other two distance-2 tuples** $(u, w)$ **and** $(w, v)$, and are thus aware of closed 6-walks (since $6 = 2 + 2 + 2$). Indeed, if we forbid such kind of message passing, the modified 2-DRFWL(2) GNNs can no longer count 6-cycles, as experimentally verified in Appendix F.2.

## 5    Related works

**The cycle counting power of GNNs.** It is proposed in [33] to use GNNs' ability to count given-length cycles as a metric for their expressiveness. Prior to this work, Arvind et al. [5] and Fürer [25] have discussed the cycle counting power of the FWL(2) test: FWL(2) can and only can graph-level count up to 7-cycles. Huang et al. [33] also characterizes the node-level cycle counting power of subgraph MPNNs and I²-GNN. Apart from counting cycles, there are also some works analyzing the general substructure counting power of GNNs. Chen et al. [15] discusses the ability of WL($k$) tests to count general subgraphs or induced subgraphs, but the result is loose. Tahmasebi et al. [48] analyzes the substructure counting power of Recursive Neighborhood Pooling, which can be seen as a subgraph GNN with recursive subgraph extraction procedures.

**The trade-off between expressive power and efficiency of GNNs.** Numerous methods have been proposed to boost the expressive power of MPNNs. Many of the provably powerful GNN models have direct correspondence to the Weisfeiler-Leman hierarchy [12, 28], such as higher-order GNNs [44] and IGNs [41–43]. Despite their simplicity in theory, those models require $O(n^{k+1})$ time and $O(n^k)$ space in order to achieve equal expressive power to FWL($k$) tests, and thus do not scale to large graphs even for $k = 2$.

Recent works try to strike a balance between the expressive power of GNNs and their efficiency. Among the state-of-the-art GNN models with *sub-$O(n^3)$* time complexity, subgraph GNNs have gained much research interest [8, 17, 24, 33, 39, 46, 57–60]. Subgraph GNNs process a graph $G$ by 1) extracting a bag of subgraphs $\{G_i : i = 1, 2, \ldots, p\}$ from $G$, 2) generating representations for every subgraph $G_i, i = 1, 2, \ldots, p$ (often using a weak GNN such as MPNN), and 3) combining the representations of all subgraphs into a representation of $G$. Most commonly, the number $p$ of subgraphs extracted is equal to the number of nodes $n$ (called a *node-based subgraph extraction policy* [24]). In this case, the time complexity of subgraph GNNs is upper-bounded by $O(nm)$, where $m$ is the number of edges. If we further adopt the $K$-*hop ego-network policy*, i.e., extracting a $K$-hop subgraph $G_u$ around each node $u$, the time complexity becomes $O(n \deg^{K+1})$. Frasca et al. [24] and Zhang et al. [58] theoretically characterize the expressive power of subgraph GNNs, and prove that subgraph GNNs with node-based subgraph extraction policies lie strictly between WL(1) and FWL(2) in the Weisfeiler-Leman hierarchy. Despite the lower complexity, in practice subgraph GNNs still suffer from heavy preprocessing and a high GPU memory usage.

Apart from subgraph GNNs, there are also attempts to add sparsity to higher-order GNNs. Morris et al. [45] proposes $\delta$-$k$-LWL, a variant of the WL($k$) test that updates a $k$-tuple $\mathbf{u}$ only from the $k$-tuples with a component *connected* to the corresponding component in $\mathbf{u}$. Zhang et al. [58] proposes LFWL(2) and SLFWL(2), which are variants of FWL(2) that update a 2-tuple $(u, v)$ from nodes in either $\mathcal{N}(v)$ or $\mathcal{N}(u) \cup \mathcal{N}(v)$. Our model can also be classified into this type of approaches, yet we not only sparsify neighbors of a 2-tuple, but also 2-tuples used in message passing, which results in much lower space complexity. A detailed comparison between our method and LFWL(2)/SLFWL(2) is included in Appendix D.

## 6 Experiments

In this section, we empirically evaluate the performance of $d$-DRFWL(2) GNNs (especially for the case of $d = 2$) and verify our theoretical results. To be specific, we focus on the following questions:

**Q1:** Can $d$-DRFWL(2) GNNs reach their theoretical counting power as stated in Theorems 4.3–4.8?

**Q2:** How do $d$-DRFWL(2) GNNs perform compared with other state-of-the-art GNN models on open benchmarks for graphs?

**Q3:** What are the time and memory costs of $d$-DRFWL(2) GNNs on various datasets?

**Q4:** Do $d$-DRFWL(2) GNNs with increasing $d$ values construct a hierarchy in discriminative power (as shown in Theorem 3.2)? Further, does this hierarchy lie between WL(1) and FWL(2) empirically?

We answer **Q1**–**Q3** in 6.1–6.3, as well as in Appendix F. The answer to **Q4** is included in Appendix F.1. The details of our implementation of $d$-DRFWL(2) GNNs, along with the experimental settings, are included in Appendix E. Our code for all experiments, including those in Section 6 of the main paper and in Appendix F, is available at `https://github.com/zml72062/DR-FWL-2`.

### 6.1 Substructure counting

**Datasets.** To answer **Q1**, we perform node-level substructure counting tasks on the synthetic dataset in [33, 60]. The synthetic dataset contains 5,000 random graphs, and the training/validation/test splitting is 0.3/0.2/0.5. The task is to perform regression on the node-level counts of certain substructures. Normalized MAE is used as the evaluation metric.

**Tasks and baselines.** To verify Theorems 4.4–4.6, we use 2-DRFWL(2) GNN to perform node-level counting task on 9 different substructures: 3-cycles, 4-cycles, 5-cycles, 6-cycles, tailed triangles, chordal cycles, 4-cliques, 4-paths and triangle-rectangles. We choose MPNN, node-based subgraph GNNs (ID-GNN, NGNN, GNNAK+), PPGN, and $I^2$-GNN as our baselines. Results for all baselines are from [33].

To verify Theorem 4.8, we compare the performances of 2-DRFWL(2) GNN and 3-DRFWL(2) GNN on node-level cycle counting tasks, with the cycle length ranging from 3 to 7.

We also conduct ablation studies to investigate *what kinds of message passing are essential* (among those depicted in Figure 1) for 2-DRFWL(2) GNNs to successfully count up to 6-cycles and other substructures. The experimental details and results are given in Appendix F.2. The studies also serve as a verification of Theorem 4.3.

Table 1: Normalized MAE results of node-level counting cycles and other substructures on synthetic dataset. The colored cell means an error less than 0.01.

| Method | Synthetic (norm. MAE) | | | | | | | | |
|---|---|---|---|---|---|---|---|---|---|
| | 3-Cyc. | 4-Cyc. | 5-Cyc. | 6-Cyc. | Tail. Tri. | Chor. Cyc. | 4-Cliq. | 4-Path | Tri.-Rect. |
| MPNN | 0.3515 | 0.2742 | 0.2088 | 0.1555 | 0.3631 | 0.3114 | 0.1645 | 0.1592 | 0.2979 |
| ID-GNN | 0.0006 | 0.0022 | 0.0490 | 0.0495 | 0.1053 | 0.0454 | 0.0026 | 0.0273 | 0.0628 |
| NGNN | 0.0003 | 0.0013 | 0.0402 | 0.0439 | 0.1044 | 0.0392 | 0.0045 | 0.0244 | 0.0729 |
| GNNAK+ | 0.0004 | 0.0041 | 0.0133 | 0.0238 | 0.0043 | 0.0112 | 0.0049 | 0.0075 | 0.1311 |
| PPGN | 0.0003 | 0.0009 | 0.0036 | 0.0071 | 0.0026 | 0.0015 | 0.1646 | 0.0041 | 0.0144 |
| $I^2$-GNN | 0.0003 | 0.0016 | 0.0028 | 0.0082 | 0.0011 | 0.0010 | 0.0003 | 0.0041 | 0.0013 |
| 2-DRFWL(2) GNN | 0.0004 | 0.0015 | 0.0034 | 0.0087 | 0.0030 | 0.0026 | 0.0009 | 0.0081 | 0.0070 |

**Results.** From Table 1, we see that 2-DRFWL(2) GNN achieves less-than-0.01 normalized MAE on all 3, 4, 5 and 6-cycles, verifying Theorem 4.5; 2-DRFWL(2) GNN also achieves less-than-0.01 normalized MAE on tailed triangles, chordal cycles, 4-paths and triangle-rectangles, verifying Theorems 4.4 and 4.6.

It is interesting that 2-DRFWL(2) GNN has a very good performance on the task of node-level counting 4-cliques, which by Theorem 4.7 it cannot count in theory. A similar phenomenon happens for subgraph GNNs. This may be because 2-DRFWL(2) GNN and subgraph GNNs still learn some local structural biases that have strong correlation with the number of 4-cliques.

Table 2: Normalized MAE results of node-level counting $k$-cycles $(3 \leqslant k \leqslant 7)$ on synthetic dataset.

| Method | Synthetic (norm. MAE) | | | | |
|---|---|---|---|---|---|
| | 3-Cyc. | 4-Cyc. | 5-Cyc. | 6-Cyc. | 7-Cyc. |
| 2-DRFWL(2) GNN | **0.0004** | **0.0015** | **0.0034** | **0.0087** | 0.0362 |
| 3-DRFWL(2) GNN | 0.0006 | 0.0020 | 0.0047 | 0.0099 | **0.0176** |

From Table 2, we see that 3-DRFWL(2) GNN achieves a comparable performance to 2-DRFWL(2) GNN on the tasks of counting 3, 4, 5 and 6-cycles. Yet when it comes to counting 7-cycles, 3-DRFWL(2) GNN greatly outperforms 2-DRFWL(2) GNN, verifying Theorems 4.7 and 4.8.

Finally, from the last row of Table 7 we see that 1-DRFWL(2) GNN achieves less-than-0.01 normalized MAE on 3-cycles, but performs badly on 4, 5 and 6-cycles. This result verifies Theorem 4.3.

### 6.2 Molecular property prediction

**Datasets.** To answer **Q2**, we evaluate the performance of $d$-DRFWL(2) GNNs on four popular molecular graph datasets—QM9, ZINC, ogbg-molhiv and ogbg-molpcba. QM9 contains 130k small molecules, and the task is regression on 12 targets. One can refer to the page for the meaning of those 12 targets. ZINC [20], including a smaller version (ZINC-12K) and a full version (ZINC-250K), is a dataset of chemical compounds and the task is graph regression. The ogbg-molhiv (containing 41k molecules) and ogbg-molpcba (containing 438k molecules) datasets belong to the Open Graph Benchmark (OGB) [31]; the task on both datasets is binary classification. Details of the four datasets are given in Appendix E.2.2.

Table 3: MAE results on QM9 (smaller the better). The top two are highlighted as **First**, **Second**.

| Target | 1-GNN | 1-2-3-GNN | DTNN | Deep LRP | PPGN | NGNN | I$^2$-GNN | 2-DRFWL(2) GNN |
|---|---|---|---|---|---|---|---|---|
| $\mu$ | 0.493 | 0.476 | **0.244** | 0.364 | **0.231** | 0.428 | 0.428 | 0.346 |
| $\alpha$ | 0.78 | 0.27 | 0.95 | 0.298 | 0.382 | 0.29 | **0.230** | **0.222** |
| $\varepsilon_{\text{homo}}$ | 0.00321 | 0.00337 | 0.00388 | **0.00254** | 0.00276 | 0.00265 | 0.00261 | **0.00226** |
| $\varepsilon_{\text{lumo}}$ | 0.00355 | 0.00351 | 0.00512 | 0.00277 | 0.00287 | 0.00297 | **0.00267** | **0.00225** |
| $\Delta\varepsilon$ | 0.0049 | 0.0048 | 0.0112 | **0.00353** | 0.00406 | 0.0038 | 0.0038 | **0.00324** |
| $R^2$ | 34.1 | 22.9 | 17.0 | 19.3 | **16.07** | 20.5 | 18.64 | **15.04** |
| ZPVE | 0.00124 | 0.00019 | 0.00172 | 0.00055 | 0.0064 | 0.0002 | **0.00014** | **0.00017** |
| $U_0$ | 2.32 | **0.0427** | 2.43 | 0.413 | 0.234 | 0.295 | 0.211 | **0.156** |
| $U$ | 2.08 | **0.111** | 2.43 | 0.413 | 0.234 | 0.361 | 0.206 | **0.153** |
| $H$ | 2.23 | **0.0419** | 2.43 | 0.413 | 0.229 | 0.305 | 0.269 | **0.145** |
| $G$ | 1.94 | **0.0469** | 2.43 | 0.413 | 0.238 | 0.489 | 0.261 | **0.156** |
| $C_v$ | 0.27 | 0.0944 | 2.43 | 0.129 | 0.184 | 0.174 | **0.0730** | **0.0901** |

**Baselines.** For QM9, the baselines are chosen as 1-GNN, 1-2-3-GNN [44], DTNN [53], Deep LRP [15], PPGN, NGNN [59] and I$^2$-GNN [33]. Methods [3, 26, 40, 47] utilizing geometric features or quantum mechanic theory are omitted to fairly compare the graph representation power of the models. For ZINC and ogbg-molhiv, we adopt GIN, PNA [16], DGN [7], HIMP [23], GSN [10], Deep LRP [15], CIN [9], NGNN, GNNAK+, SUN [24] and I$^2$-GNN as our baselines. For ogbg-molpcba, the baselines are GIN, PNA, DGN, NGNN and GNNAK+. The experimental details are given in Appendix E.2.2. We leave the results on ZINC, ogbg-molhiv and ogbg-molpcba to Appendix F.3.

**Results.** On QM9, Table 3 shows that 2-DRFWL(2) GNN attains top two results on most (11 out of 12) targets. Moreover, 2-DRFWL(2) GNN shows a good performance on targets $U_0, U, H$ and $G$, where subgraph GNNs like NGNN or I$^2$-GNN have a poor performance. The latter fact actually reveals that our method has a stronger ability to capture *long-range interactions* on graphs than subgraph GNNs, since the targets $U_0, U, H$ and $G$ are macroscopic thermodynamic properties of

molecules and heavily depend on such long-range interactions. We leave the detailed analysis to Appendix F.4. Apart from QM9, Table 8 of Appendix F.3 shows that $d$-DRFWL(2) GNN outperforms CIN [9] on ZINC-12K; the performance on ogbg-molhiv is also comparable to baseline methods. These results show that although designed for cycle counting, $d$-DRFWL(2) GNN is also highly competitive on general molecular tasks.

It is interesting to notice that $d$-DRFWL(2) GNN shows an inferior performance on ogbg-molpcba, compared with baseline methods. We conjecture that the results on ogbg-molpcba might be insensitive to the gain in cycle counting power, and might prefer simple model architectures rather than highly expressive ones.

## 6.3 Empirical efficiency

To answer **Q3**, we compare the time and memory costs of 2-DRFWL(2) GNN with MPNN, NGNN and $I^2$-GNN on two datasets—QM9 and ogbg-molhiv. We use three metrics to evaluate the empirical efficiency of 2-DRFWL(2) GNNs: the maximal GPU memory usage during training, the preprocessing time, and the training time per epoch.

To make a fair comparison, we fix the number of 2-DRFWL(2) GNN layers and the number of message passing layers in all baseline methods to 5; we also fix the size of hidden dimension to 64 for QM9 and 300 for ogbg-molhiv. The subgraph heights for NGNN and $I^2$-GNN are both 3. The batch size is always 64.

Table 4: Empirical efficiency of 2-DRFWL(2) GNN.

| Method | QM9 | | | ogbg-molhiv | | |
|---|---|---|---|---|---|---|
| | Memory (GB) | Pre. (s) | Train (s/epoch) | Memory (GB) | Pre. (s) | Train (s/epoch) |
| MPNN | 2.28 | 64 | 45.3 | 2.00 | 2.4 | 18.8 |
| NGNN | 13.72 | 2354 | 107.8 | 5.23 | 1003 | 42.7 |
| $I^2$-GNN | 19.69 | 5287 | 209.9 | 11.07 | 2301 | 84.3 |
| 2-DRFWL(2) GNN | 2.31 | 430 | 141.9 | 4.44 | 201 | 44.3 |

**Results.** From Table 4, we see that the preprocessing time of 2-DRFWL(2) GNN is much shorter than subgraph GNNs like NGNN or $I^2$-GNN. Moreover, 2-DRFWL(2) GNN requires much less GPU memory while training, compared with subgraph GNNs. The training time of 2-DRFWL(2) GNN is comparable to NGNN (which can only count up to 4-cycles), and much shorter than $I^2$-GNN.

We also evaluate the empirical efficiency of 2-DRFWL(2) GNN on graphs of larger sizes. The results, along with details of the datasets we use, are given in Appendix F.5. From Table 14 in Appendix F.5, we see that 2-DRFWL(2) GNN easily scales to graphs with $\sim 500$ nodes, as long as the average degree is small.

## 7 Conclusion and limitations

Motivated by the analysis of why FWL(2) has a stronger cycle counting power than WL(1), we propose $d$-DRFWL(2) tests and $d$-DRFWL(2) GNNs. It is then proved that with $d = 2$, $d$-DRFWL(2) GNNs can already count up to 6-cycles, retaining most of the cycle counting power of FWL(2). Because $d$-DRFWL(2) GNNs explicitly leverage the *local* nature of cycle counting, they are much more efficient than other existing GNN models that have comparable cycle counting power. Besides, $d$-DRFWL(2) GNNs also have an outstanding performance on various real-world tasks. Finally, we have to point out that our current implementation of $d$-DRFWL(2) GNNs, though being efficient most of the time, still has difficulty scaling to datasets with a large average degree such as ogbg-ppa, since the preprocessing time is too long ($\sim$40 seconds per graph on ogbg-ppa). This also makes our method unsuitable for node classification tasks, since these tasks typically involve graphs with large average degrees. We leave the exploration of more efficient $d$-DRFWL(2) implementation to future work.

## Acknowledgement

The work is supported in part by the National Natural Science Foundation of China (62276003), the National Key Research and Development Program of China (No. 2021ZD0114702), and Alibaba Innovative Research Program.

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

# A  Message passing neural networks

Message passing neural networks (MPNNs) [27, 36, 50, 55] are a class of GNNs that iteratively updates node representations $h_u, u \in \mathcal{V}_G$ by aggregating information from neighbors. At the $t$-th ($t \geqslant 1$) iteration, the update rule for MPNNs is

$$h_u^{(t)} = f^{(t)} \left( h_u^{(t-1)}, \bigoplus_{v \in \mathcal{N}(u)} m^{(t)} \left( h_u^{(t-1)}, h_v^{(t-1)}, e_{uv} \right) \right), \tag{13}$$

where $h_u^{(t)}$ is the representation of node $u$ at the $t$-th iteration and $e_{uv}$ is the edge feature of $\{u, v\} \in \mathcal{E}_G$. $f^{(t)}$ and $m^{(t)}$ are learnable functions, and $\bigoplus$ is a permutation-invariant aggregation operator, such as sum, mean or max. The representation of graph $G$ is obtained by

$$h_G = R(\{\!\!\{ h_u : u \in \mathcal{V}_G \}\!\!\}), \tag{14}$$

where $R$ is a permutation-invariant function of multisets.

It is known [55] that the ability of MPNNs to discriminate between non-isomorphic graphs is upper-bounded by the WL(1) test. When it comes to counting cycles, *MPNNs cannot graph-level count any cycles*, as stated in [33].

# B  Proofs of theorems in Section 3

## B.1  Proof of Theorem 3.1

We restate Theorem 3.1 as following,

**Theorem B.1.** *In terms of the ability to distinguish between non-isomorphic graphs, the d-DRFWL(2) test is strictly more powerful than WL(1), for any $d \geqslant 1$.*

*Proof.* In Theorem 3.2 we will prove that $(d + 1)$-DRFWL(2) is strictly more powerful than $d$-DRFWL(2), for any $d \geqslant 1$. Therefore, we only need to show that 1-DRFWL(2) is strictly more powerful than WL(1). We will first prove that 1-DRFWL(2) can give an implementation of the WL(1) test. Actually, let

$$W^{(0)}(u, v) = \begin{cases} W_{\mathrm{WL(1)}}^{(0)}(u), & d(u, v) = 0, \\ \mathrm{NULL}, & d(u, v) = 1, \end{cases} \tag{15}$$

be the initial 1-DRFWL(2) colors of $(u, v) \in \mathcal{V}_G^2$ with $0 \leqslant d(u, v) \leqslant 1$. Here $W_{\mathrm{WL(1)}}^{(0)}(u)$ is the initial WL(1) color of node $u$ (which is identical for all $u \in \mathcal{V}_G$). It is obvious that $W^{(0)}(u, v)$ does only depend on $d(u, v)$. As for the update rule, at the $(2t - 1)$-th iteration with $t \geqslant 1$, we ask

$$\mathrm{POOL}_{01}^{1(2t-1)} \left( \{\!\!\{ (W^{(2t-2)}(u, v), W^{(2t-2)}(u, u)) \}\!\!\} \right) = W^{(2t-2)}(u, u), \tag{16}$$

and

$$\mathrm{POOL}_{10}^{1(2t-1)} \left( \{\!\!\{ (W^{(2t-2)}(v, v), W^{(2t-2)}(u, v)) \}\!\!\} \right) = W^{(2t-2)}(v, v), \tag{17}$$

for those $(u, v)$ with distance 1. Here, both $\mathcal{N}_0(u) \cap \mathcal{N}_1(v)$ and $\mathcal{N}_1(u) \cap \mathcal{N}_0(v)$ have only one element, so $\mathrm{POOL}_{01}^{1(2t-1)}$ and $\mathrm{POOL}_{10}^{1(2t-1)}$ simply select the second and the first component from the unique 2-tuple in the corresponding multisets, respectively. The $\mathrm{HASH}_k^{(2t-1)}$ functions for $k = 0$ or 1 are chosen as

$$\mathrm{HASH}_1^{(2t-1)} \left( W^{(2t-2)}(u, v), \left( M_{ij}^{1(2t-1)} \right)_{0 \leqslant i, j \leqslant 1} \right) = \mathrm{CONCAT} \left( M_{01}^{1(2t-1)}, M_{10}^{1(2t-1)} \right), \tag{18}$$

$$\text{for } d(u, v) = 1,$$

$$\mathrm{HASH}_0^{(2t-1)} \left( W^{(2t-2)}(u, u), \left( M_{ij}^{0(2t-1)} \right)_{0 \leqslant i, j \leqslant 1} \right) = W^{(2t-2)}(u, u), \tag{19}$$

$$\text{for } d(u, v) = 0.$$

What we did is to ask **the $(2t-1)$-th iteration of 1-DRFWL(2) to record the WL(1) colors** $\left(W_{\textbf{WL(1)}}^{(t-1)}(u), W_{\textbf{WL(1)}}^{(t-1)}(v)\right)$ **in the 1-DRFWL(2) color of node pair** $(u, v)$. In the $2t$-th iteration, 1-DRFWL(2) then uses this record to update the colors of $(u, u)$ and $(v, v)$:

$$\mathrm{HASH}_1^{(2t)}\left(W^{(2t-1)}(u, v), \left(M_{ij}^{1(2t)}\right)_{0 \leqslant i, j \leqslant 1}\right) = W^{(2t-1)}(u, v), \tag{20}$$
$$\text{for } d(u, v) = 1,$$

$$\mathrm{HASH}_0^{(2t)}\left(W^{(2t-1)}(u, u), \left(M_{ij}^{0(2t)}\right)_{0 \leqslant i, j \leqslant 1}\right) = M_{11}^{0(2t)}, \tag{21}$$
$$\text{for } d(u, v) = 0.$$

And we ask

$$M_{11}^{0(2t)}(u, v) = \mathrm{HASH}_{\mathrm{WL(1)}}^{(t)}\left(W^{(2t-2)}(u, u), \mathrm{POOL}_{\mathrm{WL(1)}}^{(t)}\left(\{\!\{W^{(2t-2)}(v, v) : v \in \mathcal{N}(u)\}\!\}\right)\right). \tag{22}$$

Here $W^{(2t-2)}(u, u)$ and $W^{(2t-2)}(v, v)$ are stored in $W^{(2t-1)}(u, v)$ in the last iteration (in the first and the second components respectively). $\mathrm{HASH}_{\mathrm{WL(1)}}^{(t)}$ and $\mathrm{POOL}_{\mathrm{WL(1)}}^{(t)}$ are the hashing functions used by WL(1) test, as in (1). It is easy to see the above implementation uses 2 iterations of 1-DRFWL(2) update to simulate 1 iteration of WL(1) update. Therefore, 1-DRFWL(2) is at least as powerful as WL(1) in terms of the ability to distinguish between non-isomorphic graphs.

To see why 1-DRFWL(2) is strictly more powerful than WL(1), we only need to find out a pair of graphs $G$ and $H$ such that WL(1) cannot distinguish between them while 1-DRFWL(2) can. Let $G$ be two 3-cycles and $H$ be one 6-cycle. Of course WL(1) cannot distinguish between $G$ and $H$. However, 1-DRFWL(2) can distinguish between them because there are no triangles in $H$ but two in $G$, as is made clear in the following. In the first iteration of 1-DRFWL(2), the $M_{11}^{1(1)}$ term collects common neighbors of every node pair $(u, v)$ with distance 1. In $G$ this results in non-empty common neighbor multisets for all distance-1 tuples $(u, v)$, while in $H$ all such multisets are empty. 1-DRFWL(2) then makes use of this discrepancy by properly choosing the $\mathrm{HASH}_1^{(1)}$ function that assigns different colors for distance-1 tuples in $G$ and in $H$. Therefore, 1-DRFWL(2) can distinguish between $G$ and $H$. $\qquad\square$

### B.2  Proof of Theorem 3.2

We restate Theorem 3.2 as following,

**Theorem B.2.** *In terms of the ability to distinguish between non-isomorphic graphs, FWL(2) is strictly more powerful than $d$-DRFWL(2), for any $d \geqslant 1$. Moreover, $(d+1)$-DRFWL(2) is strictly more powerful than $d$-DRFWL(2).*

*Proof.* First we show that FWL(2) can give an implementation of $d$-DRFWL(2) for any $d \geqslant 1$. The implementation is in three steps:

**Distance calculation.**    For any 2-tuple $(u, v) \in \mathcal{V}_G^2$, let its FWL(2) color $W^{(0)}(u, v)$ be 0 if $u = v$, 1 if $\{u, v\} \in \mathcal{E}_G$, and $\infty$ otherwise. In the first $(n-1)$ iterations, FWL(2) can use the following update rule to calculate distance between any pair of nodes,

$$W^{(t)}(u, v) = \min\left(W^{(t-1)}(u, v), \min_{w \in \mathcal{V}_G}\left(W^{(t-1)}(w, v) + W^{(t-1)}(u, w)\right)\right). \tag{23}$$

When $t = n - 1$, the FWL(2) color for any $(u, v) \in \mathcal{V}_G^2$ gets stable and becomes $d(u, v)$.

**Initial color generation.**    At the $n$-th iteration, FWL(2) transforms $W^{(n-1)}(u, v) = d(u, v)$ into

$$W^{(n)}(u, v) = \begin{cases} \mathrm{CONCAT}\left(d(u, v), W_{d\text{-DRFWL(2)}}^{(0)}(u, v)\right), & \text{if } 0 \leqslant d(u, v) \leqslant d, \\ \mathrm{CONCAT}\left(d(u, v), \mathrm{NULL}\right), & \text{otherwise.} \end{cases} \tag{24}$$

Here $W_{d\text{-DRFWL(2)}}^{(0)}(u, v)$ is the initial $d$-DRFWL(2) color of 2-tuple $(u, v)$. Because $W_{d\text{-DRFWL(2)}}^{(0)}(u, v)$ only depends on $d(u, v)$, the RHS in (24) is a function of $W^{(n-1)}(u, v) = d(u, v)$, and thus complies with the general form of FWL(2) as in (3).

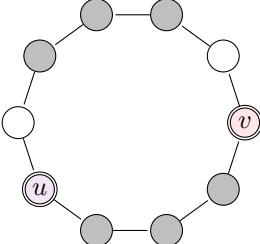 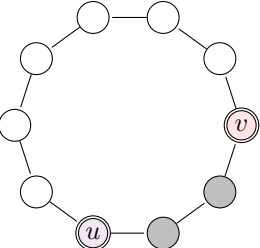

(a) When running 4-DRFWL(2) in a 10-cycle, there are 8 (marked as colored, with 3 on the inferior arc and 3 on the superior arc) nodes contributing to the update of any distance-4 tuple $(u, v)$

(b) When running 3-DRFWL(2) in a 10-cycle (*or any cycle with length* $\geqslant 10$), there are only 4 (marked as colored) nodes contributing to the update of any distance-3 tuple $(u, v)$

Figure 3: Illustration on the counterexample used in the proof of the separation result in Theorem 3.2. Here we take $d = 3$ and a $(3d + 1)$-cycle, or 10-cycle, is shown. It is clear that for a distance-$(d + 1)$ tuple there are nodes on both the inferior arc and the superior arc that contribute to its update. Contrarily, for a distance-$d$ tuple only nodes on the inferior arc contribute.

**$d$-DRFWL(2) Update.** For the $(n + t)$-th iteration with $t \geqslant 1$, the FWL(2) test uses the following $\text{POOL}^{(n+t)}$ and $\text{HASH}^{(n+t)}$ functions to simulate the $t$-th iteration of $d$-DRFWL(2):

- $\text{HASH}^{(n+t)}$ reads the first field of $W^{(n+t-1)}(u, v)$ to get $d(u, v) = k$, and decides whether to update the color for $(u, v)$ (if $0 \leqslant k \leqslant d$) or not (otherwise).

- For every $w \in \mathcal{V}_G$, $\text{POOL}^{(n+t)}$ reads the first field of $W^{(n+t-1)}(w, v)$ and $W^{(n+t-1)}(u, w)$ to find that $d(w, v) = i$ and $d(u, w) = j$, and then either selects $W^{(t-1)}_{d\text{-DRFWL(2)}}(w, v)$ and $W^{(t-1)}_{d\text{-DRFWL(2)}}(u, w)$ from the second field of $W^{(n+t-1)}(w, v)$ and $W^{(n+t-1)}(u, w)$, and applies (7) to get $M^{k(t)}_{ij}(u, v)$, if it finds $0 \leqslant i, j \leqslant d$, or ignores the contribution from $\left(W^{(n+t-1)}(w, v), W^{(n+t-1)}(u, w)\right)$ otherwise.

- $\text{HASH}^{(n+t)}$ calculates $W^{(t)}_{d\text{-DRFWL(2)}}(u, v)$ using (6), and assigns it to the second field of $W^{(n+t)}(u, v)$, if it has decided that the color for $(u, v)$ should be updated.

For the READOUT part, FWL(2) simply ignores the tuples $(u, v)$ with $d(u, v) > d$. It is easy to see the above construction does provide an implementation for $d$-DRFWL(2).

Similarly, one can prove that $(d + 1)$-DRFWL(2) also gives an implementation of $d$-DRFWL(2), for any $d \geqslant 1$, by only executing the "initial color generation" and "$d$-DRFWL(2) update" steps stated above. Since the colors for $(u, v)$ with $d(u, v) > d$ never update, it is sufficient only keeping track of pairs with mutual distance $\leqslant d + 1$, which $(d + 1)$-DRFWL(2) is capable of.

Now we will turn to prove the separation result: for any $d \geqslant 1$, there exist graphs $G$ and $H$ that FWL(2) (or $(d + 1)$-DRFWL(2)) can distinguish, but $d$-DRFWL(2) cannot. We ask $G$ **to be two** $(3d + 1)$**-cycles, and** $H$ **a single** $(6d + 2)$**-cycle**. FWL(2) can distinguish between $G$ and $H$ by (i) calculating distance between every pair of nodes in $G$ and $H$; (ii) check if there is $(u, v) \in \mathcal{V}^2$ such that $d(u, v) = \infty$ at the READOUT step. The above procedure will give $G$ a true label and $H$ a false label.

To see why $(d+1)$-DRFWL(2) can also distinguish between $G$ and $H$, we designate that $\text{POOL}^{d+1(1)}_{ij}$ be a multiset function that counts the elements in the multiset, for all $0 \leqslant i, j \leqslant d + 1$, and that $\text{HASH}^{(1)}_{d+1}$ be the sum of all items in $\left(M^{d+1(1)}_{ij}(u, v)\right)_{0 \leqslant i, j \leqslant d+1}$. Briefly speaking, **we are asking the first iteration of** $(d+1)$**-DRFWL(2) to count how many nodes** $w$ **contribute to the update of a distance-**$(d+1)$ **tuple** $(u, v)$, via the form $\left(W^{(0)}(w, v), W^{(0)}(u, w)\right)$. For any distance-$(d + 1)$ tuple in $G$, the count is 4 for $d = 1$ and $d + 5$ for $d > 1$ (actually when $d > 1$, a distance-$(d+1)$ tuple $(u, v)$ splits a $(3d + 1)$-cycle into an *inferior arc* of length $(d + 1)$ and a *superior arc* of length $2d$; $d$

nodes on the inferior arc and 3 nodes on the superior arc, along with $u$ and $v$, sum up to the number; Figure 3a illustrates the case of $d = 3$). For a distance-$(d + 1)$ tuple in $H$, the count becomes $d + 2$, which is always different from the count in $G$. The $(d+1)$-DRFWL(2) test then uses this discrepancy to render different colors for a distance-$(d + 1)$ tuple in $G$ and in $H$, thus telling $G$ from $H$.

The above proof also reveals why $d$-DRFWL(2) fails to distinguish between $G$ and $H$. Actually, let $(u, v)$ be a distance-$k$ tuple in $\mathcal{V}_G^2$ or $\mathcal{V}_H^2$ with $0 \leqslant k \leqslant d$. In the following we use $W^{(t)}(u, v)$ to denote the $d$-DRFWL(2) color of $(u, v)$ at the $t$-th iteration, both for $(u, v) \in \mathcal{V}_G^2$ and $(u, v) \in \mathcal{V}_H^2$. We will now use induction to prove that *for all $t$, $W^{(t)}(u, v)$ only depends on $d(u, v)$, and does not depend on whether $(u, v)$ is in $G$ or $H$.* This implies that for $(u, v) \in \mathcal{V}_G^2$ and $(u', v') \in \mathcal{V}_H^2$, $W^{(t)}(u, v) = W^{(t)}(u', v')$ holds for all iterations $t$ as long as $d(u, v) = d(u', v')$.

**Base case.** Since the initial $d$-DRFWL(2) color of a distance-$k$ tuple $(0 \leqslant k \leqslant d)$ only depends on $k$, the $t = 0$ case is trivial.

**Induction step.** Now we assume $W^{(t-1)}(u, v)$ only depends on $d(u, v)$ no matter which graph $(u, v)$ is in, for some $t \geqslant 1$. We will then prove that no matter what $\mathrm{HASH}_k^{(t)}$ and $\mathrm{POOL}_{ij}^{k(t)}$ functions we choose, $W^{(t)}(u, v)$ only depends on $d(u, v)$ no matter which graph $(u, v)$ is in. Actually, it is sufficient to prove that the multisets

$$\{\!\{(i, j) : w \in \mathcal{N}_i(u) \cap \mathcal{N}_j(v), 0 \leqslant i, j \leqslant d\}\!\} \tag{25}$$

are equal, *for any $(u, v) \in \mathcal{V}_G^2 \cup \mathcal{V}_H^2$ with $d(u, v) = k$ and $0 \leqslant k \leqslant d$.* This is because the inductive hypothesis leads us to the fact that

$$M_{ij}^{k(t)}(u, v) = \mathrm{POOL}_{ij}^{k(t)}\left(\{\!\{\left(W^{(t-1)}(w, v), W^{(t-1)}(u, w)\right) : w \in \mathcal{N}_i(u) \cap \mathcal{N}_j(v)\}\!\}\right).$$

can only depend on the *number of elements* of the multiset on the RHS, because *all elements in the RHS multiset must be equal*. Therefore, the aforementioned condition guarantees that $M_{ij}^{k(t)}(u, v)$ are all the same for any distance-$k$ tuple $(u, v) \in \mathcal{V}_G^2 \cup \mathcal{V}_H^2$ with any fixed $i, j, k$ and any $t$. This, along with the update rule (6), makes the induction step.

Now we prove that multisets defined by (25) are equal for all $(u, v) \in \mathcal{V}_G^2 \cup \mathcal{V}_H^2$ with any given $d(u, v) = k, 0 \leqslant k \leqslant d$. The intuition of the proof can be obtained from Figure 3b, which shows that for any distance-$d$ tuple $(u, v)$ in a cycle not shorter than $(3d + 1)$, the nodes that contribute to (25) are exactly those nodes on the *inferior arc* cut out by $(u, v)$, plus $u$ and $v$. This means that **the distance-$d$ tuple is completely agnostic about the length of the cycle in which it lies**, as long as the cycle length is $\geqslant 3d + 1$. Therefore, any distance-$d$ tuple cannot tell whether it is in $G$ or in $H$, resulting in the conclusion that the multisets in (25) are equal for all distance-$d$ tuples $(u, v)$. Similar arguments apply to all distance-$k$ tuples in $G$ and $H$, as long as $0 \leqslant k \leqslant d$. Therefore, we assert that $W^{(t)}(u, v)$ does only depend on $d(u, v)$, no matter which graph $(u, v)$ is in, at the $t$-th iteration. This finishes the inductive proof.

Notice that there are $(6d + 2)$ distance-$k$ tuples in either $G$ or $H$, for any $0 \leqslant k \leqslant d$; moreover, each of those distance-$k$ tuples have identical colors. We then assert that $G$ and $H$ must get identical representations after running $d$-DRFWL(2) on them. Therefore, $d$-DRFWL(2) fails to distinguish between $G$ and $H$. □

We remark that for any $d \geqslant 1$, $d$-DRFWL(2) tests cannot distinguish between two $k$-cycles and a $2k$-cycle, as long as $k \geqslant 3d + 1$. The proof for this fact is similar to the one elaborated above.

### B.3 Proof of the equivalence in representation power between $d$-DRFWL(2) tests and $d$-DRFWL(2) GNNs

At the end of 3.2, we informally state the fact that $d$-DRFWL(2) GNNs have equal representation power to $d$-DRFWL(2) tests under certain assumptions. We now restate the fact as the following

**Proposition B.3.** *Let $q : \mathcal{G} \to$ colors be a d-DRFWL(2) test whose $\mathrm{HASH}_k^{(t)}$, $\mathrm{POOL}_{ij}^{k(t)}$ and READOUT functions are injective, $\forall 0 \leqslant i, j, k \leqslant d$ and $\forall t \geqslant 1$; in addition, $q$ assigns different initial colors to 2-tuples $(u, v)$ with different $u, v$ distances. Let $f : \mathcal{G} \to \mathbb{R}^p$ be a d-DRFWL(2) GNN*

with $T$ d-DRFWL(2) GNN layers; the initial representations $h_{uv}^{(0)}$ are assumed to depend only on $d(u, v)$.

*If two graphs $G$ and $H$ get different representations under $f$, i.e. $f(G) \neq f(H)$, then $q$ assigns different colors for $G$ and $H$. Moreover, if $G$ and $H$ are graphs such that $q$ assigns different colors for $G$ and $H$, there exists a d-DRFWL(2) GNN $f$ such that $f(G) \neq f(H)$.*

*Proof.* For simplicity, we denote

$$\mathcal{V}_{G, \leqslant d}^2 = \{(u, v) \in \mathcal{V}_G^2 : 0 \leqslant d(u, v) \leqslant d\}. \tag{26}$$

We prove the first part by proving that, if two tuples $(u, v)$ and $(u', v')$ get different representations $h_{uv}^{(T)}$ and $h_{u'v'}^{(T)}$ after applying $L_T \circ \sigma_{T-1} \circ \cdots \circ \sigma_1 \circ L_1$, then

$$W^{(T)}(u, v) \neq W^{(T)}(u', v'), \tag{27}$$

after we apply $q$ for $T$ iterations. Since $W^{(\infty)}(u, v)$ is a refinement of $W^{(T)}(u, v)$, (27) implies that $W^{(\infty)}(u, v) \neq W^{(\infty)}(u', v')$. Now, $f(G) \neq f(H)$ means for all bijections $b$ from $\mathcal{V}_{G, \leqslant d}^2$ to $\mathcal{V}_{H, \leqslant d}^2$, there exists a pair $(u, v) \in \mathcal{V}_{G, \leqslant d}^2$ such that $h_{uv}^{(T)} \neq h_{u'v'}^{(T)}$ with $(u', v') = b(u, v)$. If the above statement holds true, this means for $(u, v)$ and $(u', v')$ we have $W^{(\infty)}(u, v) \neq W^{(\infty)}(u', v')$. Since this is the case for any bijection $b$, we assert

$$\text{READOUT} \left( \{\!\!\{ W^{(\infty)}(u, v) : (u, v) \in \mathcal{V}_{G, \leqslant d}^2 \}\!\!\} \right)$$
$$\neq \text{READOUT} \left( \{\!\!\{ W^{(\infty)}(u, v) : (u, v) \in \mathcal{V}_{H, \leqslant d}^2 \}\!\!\} \right),$$

or simply $W(G) \neq W(H)$, and the first part is proved.

Proof for the above statement can be conducted inductively. When $T = 0$, $h_{uv}^{(0)} \neq h_{u'v'}^{(0)}$ means $d(u, v) \neq d(u', v')$, since the initial representation $h_{uv}^{(0)}$ of $(u, v)$ only depends on the distance $d(u, v)$. By the second condition of $q$, the above fact further implies $W^{(0)}(u, v) \neq W^{(0)}(u', v')$, and the base case is proved.

Now, assuming that the above statement is true for $T = \ell - 1$ with $\ell \geqslant 1$, we now prove the $T = \ell$ case. Given $h_{uv}^{(\ell)} \neq h_{u'v'}^{(\ell)}$, there are two possibilities: (a) $h_{uv}^{(\ell-1)} \neq h_{u'v'}^{(\ell-1)}$; (b) for some $i, j$, $a_{uv}^{ijk(\ell)} \neq a_{u'v'}^{ijk(\ell)}$, where $k = d(u, v) = d(u', v')$ (we can safely assume $d(u, v) = d(u', v')$, since otherwise (27) trivially holds).

For possibility (a), the inductive hypothesis tells us $W^{(\ell-1)}(u, v) \neq W^{(\ell-1)}(u', v')$, thus $W^{(\ell)}(u, v) \neq W^{(\ell)}(u', v')$. For possibility (b), notice that $a_{uv}^{ijk(\ell)}$ is a function of the multiset

$$\{\!\!\{ (h_{wv}^{(\ell-1)}, h_{uw}^{(\ell-1)}) : w \in \mathcal{N}_i(u) \cap \mathcal{N}_j(v) \}\!\!\}.$$

Therefore, $a_{uv}^{ijk(\ell)} \neq a_{u'v'}^{ijk(\ell)}$ means that for all bijection $b'$ from $\mathcal{N}_i(u) \cap \mathcal{N}_j(v)$ to $\mathcal{N}_i(u') \cap \mathcal{N}_j(v')$, there exists $w \in \mathcal{N}_i(u) \cap \mathcal{N}_j(v)$ such that $(h_{wv}^{(\ell-1)}, h_{uw}^{(\ell-1)}) \neq (h_{w'v'}^{(\ell-1)}, h_{u'w'}^{(\ell-1)})$, where $w' = b'(w)$. Without loss of generality, let us discuss the case where $h_{wv}^{(\ell-1)} \neq h_{w'v'}^{(\ell-1)}$. In this case, the inductive hypothesis tells us $W^{(\ell-1)}(w, v) \neq W^{(\ell-1)}(w', v')$, thus $(W^{(\ell-1)}(w, v), W^{(\ell-1)}(u, w)) \neq (W^{(\ell-1)}(w', v'), W^{(\ell-1)}(u', w'))$. The other case $h_{uw}^{(\ell-1)} \neq h_{u'w'}^{(\ell-1)}$ also leads to this result. Since the bijection $b'$ is arbitrary, we assert that the multisets

$$\{\!\!\{ (W^{(\ell-1)}(w, v), W^{(\ell-1)}(u, w)) : w \in \mathcal{N}_i(u) \cap \mathcal{N}_j(v) \}\!\!\}$$

and

$$\{\!\!\{ (W^{(\ell-1)}(w', v'), W^{(\ell-1)}(u', w')) : w' \in \mathcal{N}_i(u') \cap \mathcal{N}_j(v') \}\!\!\}$$

are not equal. Since the functions $\text{POOL}_{ij}^{k(\ell)}$ and $\text{HASH}_k^{(\ell)}$ are injective, the above result implies $W^{(\ell)}(u, v) \neq W^{(\ell)}(u', v')$. So far, we have finished the induction step. Therefore, the first part of the theorem is true.

For the second part of the theorem, we quote the well-known result of Xu et al. [55] (Lemma 5 of the paper), that there exists a function $f : \mathcal{X} \to \mathbb{R}^n$ on a countable input space $\mathcal{X}$ such that *any multiset function $g$* can be written as

$$g(X) = \phi \left( \sum_{x \in X} f(x) \right) \tag{28}$$

for some function $\phi$, where $X$ is an arbitrary multiset whose elements are in $\mathcal{X}$.

Now, let $G$ and $H$ be graphs that both obtain their $d$-DRFWL(2) stable colorings under $q$ after $T$ iterations, and that $q$ assigns different colors for them. We will now prove that there exists a $d$-DRFWL(2) GNN with $T$ $d$-DRFWL(2) GNN layers that gives $G$ and $H$ different representations.

Since the set of $d$-DRFWL(2) colors generated by $q$ is countable, we can designate a mapping $\nu : \text{colors} \to \mathbb{N}$ that assigns a unique integer for every kind of $d$-DRFWL(2) color. Under mapping $\nu$, all $\text{HASH}_k^{(t)}, \text{POOL}_{ij}^{k(t)}$ and READOUT functions can be seen as functions with codomain $\mathbb{N}$. Then, the initial representation for $(u, v)$ with $0 \leqslant d(u, v) \leqslant d$ is $h_{uv}^{(0)} = \nu \left( W^{(0)}(u, v) \right)$. For the update rules, the lemma in [55] tells us that by choosing proper $m_{ijk}^{(t)}$ and $\phi_{ijk}^{(t)}$ functions, the following equality can hold,

$$\text{POOL}_{ij}^{k(t)}(X) = \phi_{ijk}^{(t)} \left( \sum_{x \in X} m_{ijk}^{(t)}(\nu(x)) \right),$$

with $X$ being any *multiset of 2-tuples of $d$-DRFWL(2) colors*. Therefore, we can choose $m_{ijk}^{(t)}$ in (10) following the instructions of the above lemma, and choose $\bigoplus$ as $\sum$. We then choose the functions $f_k^{(t)}$ in (11) as

$$f_k^{(t)} \left( h_{uv}^{(t-1)}, \left( a_{uv}^{ijk(t)} \right)_{0 \leqslant i,j \leqslant d} \right) = \text{HASH}_k^{(t)} \left( \nu^{-1} \left( h_{uv}^{(t-1)} \right), \left( \phi_{ijk}^{(t)} \left( a_{uv}^{ijk(t)} \right) \right)_{0 \leqslant i,j \leqslant d} \right). \tag{29}$$

It is now easy to iteratively prove that $h_{uv}^{(t)}$ is exactly $\nu \left( W^{(t)}(u, v) \right)$. For the pooling layer, we again leverage the lemma in [55] to assert that there exist functions $r$ and $M'$ such that

$$\text{READOUT}(X) = M' \left( \sum_{x \in X} r(\nu(x)) \right), \tag{30}$$

where $X$ is any multiset of $d$-DRFWL(2) colors. We then choose

$$R(X) = \sum_{x \in X} r(\nu(x)), \tag{31}$$

and choose $M = \nu \circ M'$. Now the $d$-DRFWL(2) GNN constructed above with $T$ $d$-DRFWL(2) GNN layers produces exactly the same output as $q$ (except that the output is converted to integers via $\nu$). Because $q$ assigns $G$ and $H$ different colors, the $d$-DRFWL(2) GNN gives different representations for $G$ and $H$. $\qquad \square$

## C  Proofs of theorems in Section 4

We first give a few definitions that will be useful in the proofs.

**Definition C.1** (Pair-wise cycle counts). Let $u, v \in \mathcal{V}_G$, we denote $C_{k,l}(u, v)$ as the number of $(k + l)$-cycles $S$ that satisfy:

- $S$ passes $u$ and $v$.

- There exists a $k$-path and an $l$-path (distinct from one another) from $u$ to $v$, such that every edge in either path is an edge included in $S$.

For instance, the substructures counted by $C_{2,3}(u, v)$ and $C_{3,4}(u, v)$ are depicted in Figures 4a and 4b respectively.

**Definition C.2** (Tailed triangle counts). Let $u, v \in \mathcal{V}_G$, $T(u, v)$ is defined as the number of tailed triangles in which $u$ and $v$ are at positions shown in Figure 4c.

**Definition C.3** (Chordal cycle counts). Let $u, v \in \mathcal{V}_G$, $CC_1(u, v)$ and $CC_2(u, v)$ are defined as the numbers of chordal cycles in which $u$ and $v$ are at positions shown in Figures 4d and 4e, respectively. Moreover, we use $CC_1(u)$ and $CC_2(u)$ to denote the node-level chordal cycle counts with the node $u$ located at positions shown in Figures 4f and 4g, respectively.

The notations $CC_1$ and $CC_2$ are overloaded in the above definition, which shall not cause ambiguities with a check on the number of arguments.

**Definition C.4** (Triangle-rectangle counts). Let $u, v \in \mathcal{V}_G$, $TR_1(u, v)$ and $TR_2(u, v)$ are defined as the numbers of triangle-rectangles in which $u$ and $v$ are at positions shown in Figures 4h and 4i, respectively. We also define three types of node-level triangle-rectangle counts, namely $TR_1(u)$, $TR_2(u)$ and $TR_3(u)$. They are the numbers of triangle-rectangles where node $u$ is located at positions shown in Figures 4j, 4k and 4l respectively.

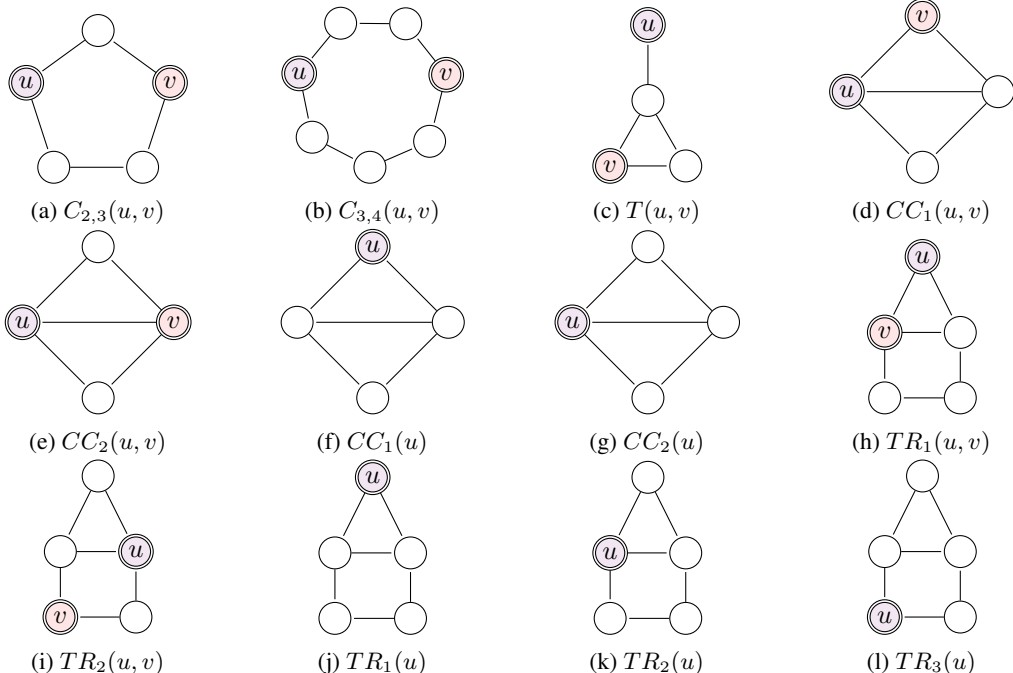

Figure 4: Illustrations of certain substructure counts.

There are relations between the above-defined counts. For chordal cycle counts we have

$$CC_1(u) = \frac{1}{2} \sum_{v \in \mathcal{N}_1(u)} CC_1(v, u), \quad CC_2(u) = \frac{1}{2} \sum_{v \in \mathcal{N}_1(u)} CC_1(u, v). \tag{32}$$

and

$$CC_2(u) = \sum_{v \in \mathcal{N}_1(u)} CC_2(u, v). \tag{33}$$

For triangle-rectangle counts we have

$$TR_1(u) = \frac{1}{2} \sum_{v \in \mathcal{N}_1(u)} TR_1(u, v), \quad TR_2(u) = \sum_{v \in \mathcal{N}_1(u)} TR_1(v, u), \tag{34}$$

and

$$TR_2(u) = \sum_{v \in \mathcal{N}_1(u) \cup \mathcal{N}_2(u)} TR_2(u, v), \quad TR_3(u) = \sum_{v \in \mathcal{N}_1(u) \cup \mathcal{N}_2(u)} TR_2(v, u). \tag{35}$$

We also denote $P_k(u, v)$ as the number of $k$-paths starting at node $u$ and ending at node $v$, $W_k(u, v)$ as the number of $k$-walks from $u$ to $v$, and $C_k(u)$ as the number of $k$-cycles that pass a node $u$.

The following lemma will be used throughout this section.

**Lemma C.5.** *Let $S$ be a graph substructure. If there exists a $d$-DRFWL(2) test $q$ such that for any graph $G \in \mathcal{G}$,*

$$W^{(T)}(u, u) = C(S, u, G), \quad \forall u \in \mathcal{V}_G \tag{36}$$

*after running $q$ for $T$ iterations on $G$, then $d$-DRFWL(2) GNNs can node-level count $S$.*

*Proof.* Let $(G_1, u_1), (G_2, u_2) \in \mathcal{G} \times \mathcal{V}$, and $C(S, u_1, G_1) \neq C(S, u_2, G_2)$. Then by assumption,

$$W^{(T)}(u_1, u_1) \neq W^{(T)}(u_2, u_2) \tag{37}$$

after running $q$ for $T$ iterations on $G_1$ and $G_2$. From the proof of Proposition B.3 (which is in Appendix B.3), we see that there exists a $d$-DRFWL(2) GNN $f$ with $T$ $d$-DRFWL(2) GNN layers such that a tuple $(u, v)$ with $0 \leqslant d(u, v) \leqslant d$ gets its representation $h_{uv}^{(T)} = \nu\left(W^{(T)}(u, v)\right)$, where $\nu$ is an injective mapping from the color space of $q$ to $\mathbb{N}$. Therefore, (37) implies $h_{u_1 u_1}^{(T)} \neq h_{u_2 u_2}^{(T)}$, or $f(G_1, u_1) \neq f(G_2, u_2)$. This means that $d$-DRFWL(2) GNNs can node-level count $S$. $\qquad\square$

## C.1 Proof of Theorem 4.3

We restate Theorem 4.3 as following,

**Theorem C.6.** *1-DRFWL(2) GNNs can node-level count 3-cycles, but cannot graph-level count any longer cycles.*

*Proof.* To prove that 1-DRFWL(2) GNNs can node-level count 3-cycles, it suffices to find a 1-DRFWL(2) test $q$ such that after running $q$ for $T$ iterations on a graph $G$,

$$W^{(T)}(u, u) = C_3(u), \quad \forall u \in \mathcal{V}_G.$$

Below we explicitly construct $q$. Notice that if $d(u, v) = 1$, $M_{11}^{1(1)}(u, v)$ can know the count $C(u, v) = |\mathcal{N}_1(u) \cap \mathcal{N}_1(v)|$ in the first 1-DRFWL(2) iteration. This is essentially the number of triangles in which $\{u, v\} \in \mathcal{E}_G$ is included as an edge. In the second iteration, we can ask

$$W^{(2)}(u, u) = \frac{1}{2} \sum_{v \in \mathcal{N}_1(u)} C(u, v), \tag{38}$$

and we have $W^{(2)}(u, u) = C_3(u)$. Therefore, the positive result is proved.

To prove that 1-DRFWL(2) GNNs cannot graph-level count $k$-cycles with $k \geqslant 4$, we make use of the fact mentioned at the end of Appendix B.2, that 1-DRFWL(2) tests (and thus GNNs) cannot distinguish between two $k$-cycles and one $2k$-cycle, as long as $k \geqslant 4$. This set of counterexamples leads to the negative result. $\qquad\square$

## C.2 Proofs of Theorems 4.4–4.7

Before proving the theorems, we state and prove a series of lemmas.

**Lemma C.7.** *There exists a 2-DRFWL(2) test $q$ such that for any graph $G \in \mathcal{G}$ and for any 2-tuple $(u, v) \in \mathcal{V}_G^2$ with $d(u, v) = 1$ or $d(u, v) = 2$, $q$ assigns*

$$W^{(T)}(u, v) = P_2(u, v), \tag{39}$$

*for some integer $T \geqslant 1$.*

*Proof.* Let $q$ be a 2-DRFWL(2) test that satisfies: at the first iteration, $M_{11}^{1(1)}(u, v)$ and $M_{11}^{2(1)}(u, v)$ both count the number of nodes in $\mathcal{N}_1(u) \cap \mathcal{N}_1(v)$, and $\mathrm{HASH}_k^{(1)}$ of $q$ simply selects the $M_{11}^{k(1)}$ component, for $k = 1, 2$; $q$ stops updating colors for any tuple from the second iteration. It is easy to see that $q$ satisfies the condition stated in the lemma. $\qquad\square$

**Lemma C.8.** *There exists a 2-DRFWL(2) test $q$ such that for any graph $G \in \mathcal{G}$ and for any node $u \in \mathcal{V}_G$, $q$ assigns*

$$W^{(T)}(u, u) = \deg(u), \tag{40}$$

*for some integer $T \geqslant 1$. Here $\deg(u)$ is the degree of node $u \in \mathcal{V}_G$.*

*Proof.* Let $q$ be a 2-DRFWL(2) test that satisfies: at the first iteration, $M_{11}^{0(1)}(u, u)$ counts $|\mathcal{N}_1(u)| = \deg(u)$, and $\mathrm{HASH}_0^{(1)}$ of $q$ selects the $M_{11}^{0(1)}$ component, then $W^{(1)}(u, u) = \deg(u)$. $q$ then stops updating colors for any tuple from the second iteration. It is obvious that $q$ satisfies the condition stated in the lemma. $\qquad\square$

**Lemma C.9.** *There exists a 2-DRFWL(2) test $q$ such that for any graph $G \in \mathcal{G}$ and for any 2-tuple $(u, v) \in \mathcal{V}_G^2$ with $d(u, v) = 1$ or $d(u, v) = 2$, $q$ assigns*

$$W^{(T)}(u, v) = \deg(u) + \deg(v), \tag{41}$$

*for some integer $T \geqslant 2$.*

*Proof.* By Lemma C.8, there exists a 2-DRFWL(2) test $q$ that assigns $W^{(T)}(u, u) = \deg(u)$ for distance-0 tuples $(u, u)$ with $u \in \mathcal{V}_G$, where $T \geqslant 1$. Now, for the $(T+1)$-th iteration of $q$, we ask

$$M_{0k}^{k(T+1)}(u, v) = W^{(T)}(u, u), \tag{42}$$

$$M_{k0}^{k(T+1)}(u, v) = W^{(T)}(v, v), \tag{43}$$

where $k = 1$ or $2$; $q$ then assigns

$$W^{(T+1)}(u, v) = M_{0k}^{k(T+1)}(u, v) + M_{k0}^{k(T+1)}(u, v), \tag{44}$$

for $d(u, v) = k$ and $k = 1, 2$. Now $W^{(T+1)}(u, v) = \deg(u) + \deg(v)$ for $T + 1 \geqslant 2$. $\qquad\square$

**Lemma C.10.** *There exists a 2-DRFWL(2) test $q$ such that for any graph $G \in \mathcal{G}$ and for any 2-tuple $(u, v) \in \mathcal{V}_G^2$ with $d(u, v) = 1$ or $d(u, v) = 2$, $q$ assigns*

$$W^{(T)}(u, v) = P_3(u, v), \tag{45}$$

*for some integer $T \geqslant 2$.*

*Proof.* A 3-path starting at $u$ and ending at $v$ is a 3-walk $u \to x \to y \to v$ with $u \neq y$ and $x \neq v$. Therefore,

$$P_3(u, v) = W_3(u, v) - \#(u \to x \to u \to v) - \#(u \to v \to y \to v) + 1_{(u,v)\in\mathcal{E}_G}. \tag{46}$$

Here $\#(u \to x \to u \to v)$ is the number of different ways to walk from $u$ to a neighboring node $x$, then back to $u$, and finally to $v$, which is also a neighbor of $u$ ($x$ can coincide with $v$). The term $\#(u \to v \to y \to v)$ can be interpreted analogously. The $1_{(u,v)\in\mathcal{E}_G}$ takes value 1 if $(u, v) \in \mathcal{E}_G$, and 0 otherwise. This term accounts for the count $\#(u \to v \to u \to v)$ which is subtracted twice.

It is easy to see that

$$\#(u \to x \to u \to v) = 1_{(u,v)\in\mathcal{E}_G}\deg(u), \tag{47}$$

$$\#(u \to v \to y \to v) = 1_{(u,v)\in\mathcal{E}_G}\deg(v). \tag{48}$$

Now we construct $q$ as following: at the first iteration, $q$ chooses proper $\mathrm{POOL}_{ij}^{k(1)}$ and $\mathrm{HASH}_k^{(1)}$ functions ($0 \leqslant i, j, k \leqslant 2$) such that

$$W^{(1)}(u, v) = \begin{cases} \text{arbitrary,} & \text{if } d(u, v) = 0, \\ (P_2(u, v), \deg(u) + \deg(v)), & \text{if } d(u, v) = 1, \\ P_2(u, v), & \text{if } d(u, v) = 2. \end{cases} \tag{49}$$

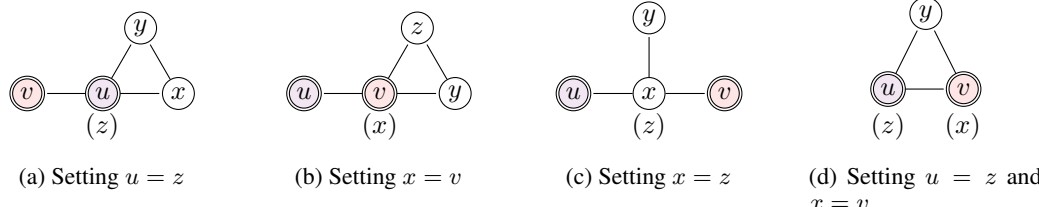

| (a) Setting $u = z$ | (b) Setting $x = v$ | (c) Setting $x = z$ | (d) Setting $u = z$ and $x = v$ |

Figure 5: The four types of redundant 4-walks requiring subtraction in the proof of Lemma C.12. Each of the 4 situations is obtained by coalescing one or more node pairs from $(u, z)$, $(x, z)$ and $(x, v)$ in a 4-walk $u \to x \to y \to z \to v$, while subject to constraints $u \neq y$ and $y \neq v$.

From Lemmas C.7 and C.9, this is possible. At the second iteration, we ask $M_{11}^{k(2)}$, $M_{12}^{k(2)}$ and $M_{21}^{k(2)}$ to be

$$M_{11}^{k(2)}(u, v) = \sum_{w \in \mathcal{N}_1(u) \cap \mathcal{N}_1(v)} \left( P_2(u, w) + P_2(w, v) \right), \tag{50}$$

$$M_{12}^{k(2)}(u, v) = \sum_{w \in \mathcal{N}_1(u) \cap \mathcal{N}_2(v)} P_2(w, v), \tag{51}$$

$$M_{21}^{k(2)}(u, v) = \sum_{w \in \mathcal{N}_2(u) \cap \mathcal{N}_1(v)} P_2(u, w), \tag{52}$$

respectively. Here $k$ takes 1 or 2. We notice that if $d(u, v) = k$ with $k = 1, 2$,

$$W_3(u, v) = \frac{1}{2} \left( M_{11}^{k(2)}(u, v) + M_{12}^{k(2)}(u, v) + M_{21}^{k(2)}(u, v) + 1_{d(u,v)=1}(\deg(u) + \deg(v)) \right) \tag{53}$$

because the RHS of (53) is exactly $\frac{1}{2} \left( \sum_{w \in \mathcal{N}_1(v)} W_2(u, w) + \sum_{w \in \mathcal{N}_1(u)} W_2(w, v) \right)$. Therefore, for a distance-$k$ tuple $(u, v)$ where $k = 1$ or 2,

$$P_3(u, v) = W_3(u, v) - 1_{d(u,v)=1}(\deg(u) + \deg(v) - 1) \tag{54}$$

is a function of $M_{11}^{k(2)}$, $M_{12}^{k(2)}$, $M_{21}^{k(2)}$ and $W^{(1)}(u, v)$, and thus can be assigned to $W^{(2)}(u, v)$ by choosing proper $\text{HASH}_k^{(2)}$ functions. After $q$ assigns $W^{(2)}(u, v) = P_3(u, v)$ for those $(u, v)$ with $d(u, v) = 1$ or $d(u, v) = 2$, it then stops updating colors for any tuple from the third iteration. It is now clear that $q$ satisfies the condition stated in the lemma. $\qquad \square$

**Lemma C.11.** *There exists a 2-DRFWL(2) test $q$ such that for any graph $G \in \mathcal{G}$ and for any node $u \in \mathcal{V}_G$, $q$ assigns*

$$W^{(T)}(u, u) = C_3(u), \tag{55}$$

*for some integer $T \geqslant 2$.*

*Proof.* This lemma follows from Theorem 4.3 and the fact that the update rule of 2-DRFWL(2) tests encompasses that of 1-DRFWL(2) tests. $\qquad \square$

**Lemma C.12.** *There exists a 2-DRFWL(2) test $q$ such that for any graph $G \in \mathcal{G}$ and for any 2-tuple $(u, v) \in \mathcal{V}_G^2$ with $d(u, v) = 1$ or $d(u, v) = 2$, $q$ assigns*

$$W^{(T)}(u, v) = P_4(u, v), \tag{56}$$

*for some integer $T \geqslant 3$.*

*Proof.* A 4-path starting at $u$ and ending at $v$ is a 4-walk $u \to x \to y \to z \to v$ with $u \neq y, u \neq z, x \neq z, x \neq v$ and $y \neq v$. We will calculate $P_4(u, v)$ as following: first define

$$P_{2,2}(u, v) = \sum_{y \neq u, y \neq v} P_2(u, y) P_2(y, v). \tag{57}$$

It is easy to see that $P_{2,2}(u,v)$ gives the number of 4-walks $u \to x \to y \to z \to v$ with only constraints $u \neq y$ and $y \neq v$. Therefore,

$$P_4(u,v) = P_{2,2}(u,v) - \#(\mathrm{a}) - \#(\mathrm{b}) - \#(\mathrm{c}) + \#(\mathrm{d}), \tag{58}$$

where $\#(\mathrm{a})$, $\#(\mathrm{b})$, $\#(\mathrm{c})$ and $\#(\mathrm{d})$ are the numbers of four types of 4-walks illustrated in Figure 5a, 5b, 5c and 5d, respectively. They correspond to setting $u = z$, $x = v$, $x = z$ and both setting $u = z$ and $x = v$ in the 4-walk $u \to x \to y \to z \to v$, respectively, while keeping $u \neq y$ and $y \neq v$.

Now, it is not hard to get the following results,

$$\#(\mathrm{a}) = 1_{(u,v) \in \mathcal{E}_G}(2C_3(u) - P_2(u,v)), \tag{59}$$

$$\#(\mathrm{b}) = 1_{(u,v) \in \mathcal{E}_G}(2C_3(v) - P_2(u,v)), \tag{60}$$

$$\#(\mathrm{c}) = \sum_{x \in \mathcal{N}_1(u) \cap \mathcal{N}_1(v)} (\deg(x) - 2), \tag{61}$$

$$\#(\mathrm{d}) = 1_{(u,v) \in \mathcal{E}_G} P_2(u,v). \tag{62}$$

We will briefly explain (59)–(62) in the following. The contribution from (59), (60) and (62) only exists when $u$ and $v$ are neighbors, which accounts for the $1_{(u,v) \in \mathcal{E}_G}$ factor. For $\#(\mathrm{a})$, if we allow $v = y$, then the count is simply $C_3(u)$ times 2 (because two directions of a 3-cycle that passes $u$ are counted as two different walks); the contribution of counts from the additional $v = y$ case is exactly $\#(\mathrm{d})$, and is $P_2(u,v)$, as long as $(u,v) \in \mathcal{E}_G$. A similar argument applies to $\#(\mathrm{b})$. Determining $\#(\mathrm{c})$ and $\#(\mathrm{d})$ is easier and the calculation will not be elaborated here.

Combining the above results, we can give a formula for $P_4(u,v)$,

$$P_4(u,v) = P_{2,2}(u,v) - \sum_{x \in \mathcal{N}_1(u) \cap \mathcal{N}_1(v)} (\deg(x) - 2) - 1_{(u,v) \in \mathcal{E}_G}(2C_3(u) + 2C_3(v) - 3P_2(u,v)). \tag{63}$$

Given the explicit formula (63), we now construct $q$ as follows: at the first iteration, $q$ chooses proper $\mathrm{POOL}_{ij}^{k(1)}$ and $\mathrm{HASH}_k^{(1)}$ functions ($0 \leqslant i,j,k \leqslant 2$) such that

$$W^{(1)}(u,v) = \begin{cases} \text{arbitrary,} & \text{if } d(u,v) = 0, \\ (1, P_2(u,v), \deg(u) + \deg(v)), & \text{if } d(u,v) = 1, \\ (2, P_2(u,v), \deg(u) + \deg(v)), & \text{if } d(u,v) = 2. \end{cases} \tag{64}$$

From Lemmas C.7 and C.9, this is possible. At the second iteration, we let

$$P_{2,2}^{ij}(u,v) = \sum_{w \in \mathcal{N}_i(u) \cap \mathcal{N}_j(v)} P_2(u,w) P_2(w,v), \tag{65}$$

where $1 \leqslant i,j \leqslant 2$, and

$$D_{11}(u,v) = \sum_{w \in \mathcal{N}_1(u) \cap \mathcal{N}_1(v)} [(\deg(u) + \deg(w)) + (\deg(w) + \deg(v))], \tag{66}$$

$$N_{11}(u,v) = |\mathcal{N}_1(u) \cap \mathcal{N}_1(v)|, \tag{67}$$

for every $(u,v)$ with $d(u,v) = 1$ or $d(u,v) = 2$. Notice that (65), (66) and (67) can be calculated by an iteration of 2-DRFWL(2) with proper $\mathrm{POOL}_{ij}^{k(2)}$ functions, where $1 \leqslant i,j,k \leqslant 2$. Therefore, we assert that

$$\sum_{x \in \mathcal{N}_1(u) \cap \mathcal{N}_1(v)} (\deg(x) - 2) = \frac{1}{2} D_{11}(u,v) - \left(2 + \frac{\deg(u) + \deg(v)}{2}\right) N_{11}(u,v) \tag{68}$$

can be calculated by an iteration of 2-DRFWL(2), with proper $\mathrm{POOL}_{ij}^{k(2)}$ and $\mathrm{HASH}_k^{(2)}$ functions, $1 \leqslant i,j,k \leqslant 2$. We now ask $q$ to assign

$$W^{(2)}(u,v) = \begin{cases} C_3(u), & \text{if } d(u,v) = 0, \\ \left(P_2(u,v), \sum_{x \in \mathcal{N}_1(u) \cap \mathcal{N}_1(v)}(\deg(x) - 2), \sum_{1 \leqslant i,j \leqslant 2} P_{2,2}^{ij}(u,v)\right), & \text{if } d(u,v) = 1, \\ \left(\sum_{x \in \mathcal{N}_1(u) \cap \mathcal{N}_1(v)}(\deg(x) - 2), \sum_{1 \leqslant i,j \leqslant 2} P_{2,2}^{ij}(u,v)\right), & \text{if } d(u,v) = 2, \end{cases} \tag{69}$$

at the second iteration. From the above discussion and Lemma C.11, this is possible. At the third iteration, $q$ gathers information from (69) and assigns $W^{(3)}(u,v) = P_4(u,v)$ for those $(u,v)$ with $d(u,v) = 1$ or $d(u,v) = 2$, using formula (63); since we already have all the terms of (63) prepared in (69) (notice that $\sum_{1 \leqslant i,j \leqslant 2} P_{2,2}^{ij}(u,v) = P_{2,2}(u,v)$), it is easy to see that $P_4(u,v)$ can be calculated by an iteration of 2-DRFWL(2) with proper $\mathrm{POOL}_{ij}^{k(3)}$ and $\mathrm{HASH}_k^{(3)}$ functions, $1 \leqslant i,j,k \leqslant 2$. We then ask $q$ to stop updating colors for any tuple from the fourth iteration. Now $q$ satisfies the condition stated in the lemma. $\qquad\square$

**Lemma C.13.** *There exists a 2-DRFWL(2) test $q$ such that for any graph $G \in \mathcal{G}$ and for any 2-tuple $(u,v) \in \mathcal{V}_G^2$ with $d(u,v) = 1$ or $d(u,v) = 2$, $q$ assigns*

$$W^{(T)}(u,v) = W_4(u,v), \tag{70}$$

*for some integer $T \geqslant 3$.*

*Proof.* Following the notations in the proof of Lemma C.12, we have

$$W_4(u,v) = W_{2,2}(u,v) + \#(u \to x \to u \to z \to v) + \#(u \to x \to v \to z \to v), \tag{71}$$

where $\#(u \to x \to u \to z \to v)$ is the number of different ways to walk from $u$ to a neighboring node $x$, then back to $u$, and, through another neighboring node $z$, finally to $v$ ($x, z, v$ can coincide with each other). The number $\#(u \to x \to v \to z \to v)$ can be interpreted analogously. It is easy to see that

$$\#(u \to x \to u \to z \to v) = \deg(u)P_2(u,v), \tag{72}$$
$$\#(u \to x \to v \to z \to v) = \deg(v)P_2(u,v), \tag{73}$$

for $d(u,v) = 1$ or $d(u,v) = 2$. Therefore,

$$W_4(u,v) = W_{2,2}(u,v) + (\deg(u) + \deg(v))P_2(u,v). \tag{74}$$

From Lemma C.12, there exists a 2-DRFWL(2) test $q$ such that $q$ assigns $W^{(2)}(u,v) = P_{2,2}(u,v)$ for $(u,v)$ with $d(u,v) = 1$ or $d(u,v) = 2$. Moreover, from Lemmas C.7 and C.9, there exists another 2-DRFWL(2) test $q'$ such that $q'$ assigns $W^{(2)}(u,v) = (\deg(u) + \deg(v))P_2(u,v)$, for $d(u,v) = 1$ or $d(u,v) = 2$. Synthesizing the colors that $q$ and $q'$ assign to $(u,v)$ respectively, it is easy to construct a 2-DRFWL(2) test that assigns $W^{(3)}(u,v) = W_4(u,v)$, where $d(u,v) = 1$ or $d(u,v) = 2$. We then let it stop updating colors for any tuple from the fourth iteration to make it satisfy the condition stated in the lemma. $\qquad\square$

In the following, we denote $P_k(u)$ as the number of $k$-paths that starts at a node $u$, and $W_k(u)$ as the number of $k$-walks that starts at a node $u$.

**Lemma C.14.** *There exists a 2-DRFWL(2) test $q$ for any $k \geqslant 1$ such that for any graph $G \in \mathcal{G}$ and for any node $u \in \mathcal{V}_G$, $q$ assigns*

$$W^{(T)}(u,u) = W_k(u), \tag{75}$$

*for some integer $T$.*

*Proof.* We prove by induction. When $k = 1$, $W_1(u) = \deg(u)$, and by Lemma C.8, the conclusion is obvious. Now we assume that for $k = \ell$, there exists a 2-DRFWL(2) test $q$ such that $q$ assigns

$$W^{(T)}(u,u) = W_\ell(u), \quad \forall u \in \mathcal{V}_G, \tag{76}$$

for some integer $T$. Then, at the $(T+1)$-th iteration, we ask

$$W^{(T+1)}(u,v) = \begin{cases} (\deg(u), W^{(T)}(u,u)), & \text{if } d(u,v) = 0, \\ W^{(T)}(u,u) + W^{(T)}(v,v), & \text{if } d(u,v) = 1, \\ \text{arbitrary}, & \text{if } d(u,v) = 2. \end{cases} \tag{77}$$

At the $(T+2)$-th iteration, we ask

$$W^{(T+2)}(u,u) = \sum_{v \in \mathcal{N}_1(u)} W^{(T+1)}(u,v) - \deg(u)W^{(T)}(u,u), \tag{78}$$

for every distance-0 tuple $(u, u) \in \mathcal{V}_G^2$. It is easy to see that

$$W^{(T+2)}(u, u) = \sum_{v \in \mathcal{N}_1(u)} W_\ell(v). \tag{79}$$

Therefore, $W^{(T+2)}(u, u) = W_{\ell+1}(u)$, and the induction step is finished. We then assert that the lemma is true. $\qquad \square$

**Lemma C.15.** *There exists a 2-DRFWL(2) test $q$ such that for any graph $G \in \mathcal{G}$ and for any node $u \in \mathcal{V}_G$, $q$ assigns*

$$W^{(T)}(u, u) = C_4(u), \tag{80}$$

*for some integer $T \geqslant 3$.*

*Proof.* From Lemma C.10, there exists a 2-DRFWL(2) test $q$ such that $q$ assigns $W^{(T)}(u, v) = C_{1,3}(u, v)$ for some integer $T \geqslant 2$, for any graph $G \in \mathcal{G}$ and any 2-tuple $(u, v) \in \mathcal{V}_G^2$ with $d(u, v) = 1$. $C_4(u)$ can be calculated from $C_{1,3}(u, v)$ in another iteration. $\qquad \square$

**Lemma C.16.** *There exists a 2-DRFWL(2) test $q$ such that for any graph $G \in \mathcal{G}$ and for any node $u \in \mathcal{V}_G$, $q$ assigns*

$$W^{(T)}(u, u) = C_5(u), \tag{81}$$

*for some integer $T \geqslant 4$.*

*Proof.* From Lemma C.12, there exists a 2-DRFWL(2) test $q$ such that $q$ assigns $W^{(T)}(u, v) = C_{1,4}(u, v)$ for some integer $T \geqslant 3$, for any graph $G \in \mathcal{G}$ and any 2-tuple $(u, v) \in \mathcal{V}_G^2$ with $d(u, v) = 1$. $C_5(u)$ can be calculated from $C_{1,4}(u, v)$ in another iteration. $\qquad \square$

**Lemma C.17.** *There exists a 2-DRFWL(2) test $q$ such that for any graph $G \in \mathcal{G}$ and for any node $u \in \mathcal{V}_G$, $q$ assigns*

$$W^{(T)}(u, u) = CC_1(u), \tag{82}$$

*for some integer $T \geqslant 3$.*

*Proof.* By the first equation of (32),

$$
\begin{aligned}
CC_1(u) &= \frac{1}{2} \sum_{v \in \mathcal{N}_1(u)} CC_1(v, u) \\
&= \frac{1}{2} \sum_{v \in \mathcal{N}_1(u)} \sum_{x \in \mathcal{N}_1(u) \cap \mathcal{N}_1(v)} (P_2(x, v) - 1) \\
&= \frac{1}{2} \sum_{v \in \mathcal{N}_1(u)} \sum_{x \in \mathcal{N}_1(u) \cap \mathcal{N}_1(v)} (P_2(x, v) + P_2(u, x) - 1) \\
&\quad - \frac{1}{2} \sum_{v \in \mathcal{N}_1(u)} \sum_{x \in \mathcal{N}_1(u) \cap \mathcal{N}_1(v)} P_2(u, x) \\
&= \frac{1}{2} \sum_{v \in \mathcal{N}_1(u)} \sum_{x \in \mathcal{N}_1(u) \cap \mathcal{N}_1(v)} (P_2(x, v) + P_2(u, x) - 1) \\
&\quad - \frac{1}{2} \sum_{x \in \mathcal{N}_1(u)} P_2(u, x) \left( \sum_{v \in \mathcal{N}_1(u) \cap \mathcal{N}_1(x)} 1 \right).
\end{aligned}
\tag{83}
$$

Now, using Lemma C.7, $CC_1(u)$ can be assigned by a 2-DRFWL(2) test to $W^{(T)}(u, u)$, for some integer $T \geqslant 3$ and for all nodes $u \in \mathcal{V}_G$. $\qquad \square$

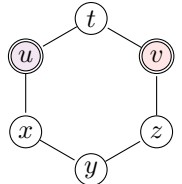 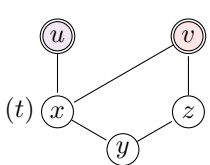 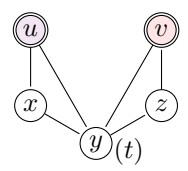 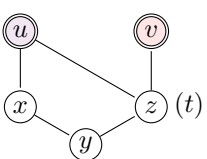

(a) A 6-cycle with $u$ and $v$ on meta-positions. The number of such 6-cycles is exactly $C_{2,4}(u,v)$. We require $t \neq x, t \neq y, t \neq z$

(b) Setting $t = x$, the count of the substructure is #(b)

(c) Setting $t = y$, the count of the substructure is #(c)

(d) Setting $t = z$, the count of the substructure is #(d)

Figure 6: Illustrations accompanying the proof of Lemma C.19. In all subfigures, it is assumed that $u \to x \to y \to z \to v$ is a 4-path and $u \to t \to v$ is a 2-path.

**Lemma C.18.** *There exists a 2-DRFWL(2) test $q$ such that for any graph $G \in \mathcal{G}$ and for any node $u \in \mathcal{V}_G$, $q$ assigns*

$$W^{(T)}(u, u) = TR_1(u), \tag{84}$$

*for some integer $T \geqslant 4$.*

*Proof.* We first calculate $TR_1(u, v)$ as following,

$$TR_1(u, v) = \sum_{z \in \mathcal{N}_1(u) \cap \mathcal{N}_1(v)} P_3(z, v) - \sum_{z \in \mathcal{N}_1(u) \cap \mathcal{N}_1(v)} (P_2(u, z) - 1) - P_2(u, v)(P_2(u, v) - 1). \tag{85}$$

Therefore, by the first equation of (34),

$$\begin{aligned} TR_1(u) &= \frac{1}{2} \sum_{v \in \mathcal{N}_1(u)} TR_1(u, v) \\ &= \frac{1}{2} \sum_{v \in \mathcal{N}_1(u)} \sum_{z \in \mathcal{N}_1(u) \cap \mathcal{N}_1(v)} (P_3(u, z) + P_3(z, v)) \\ &\quad - \frac{1}{2} \sum_{z \in \mathcal{N}_1(u)} P_3(u, z) \left( \sum_{v \in \mathcal{N}_1(u) \cap \mathcal{N}_1(z)} 1 \right) \\ &\quad - \frac{1}{2} \sum_{z \in \mathcal{N}_1(u)} (P_2(u, z) - 1) \left( \sum_{v \in \mathcal{N}_1(u) \cap \mathcal{N}_1(z)} 1 \right) \\ &\quad - \frac{1}{2} \sum_{v \in \mathcal{N}_1(u)} P_2(u, v)(P_2(u, v) - 1). \end{aligned} \tag{86}$$

By Lemmas C.7 and C.10, $TR_1(u)$ can be assigned by a 2-DRFWL(2) test to $W^{(T)}(u, u)$, for some integer $T \geqslant 4$ and for all nodes $u \in \mathcal{V}_G$. $\qquad\square$

**Lemma C.19.** *There exists a 2-DRFWL(2) test $q$ such that for any graph $G \in \mathcal{G}$ and for any node $u \in \mathcal{V}_G$, $q$ assigns*

$$W^{(T)}(u, u) = C_6(u), \tag{87}$$

*for some integer $T \geqslant 5$.*

*Proof.* We first try to calculate $C_{2,4}(u, v)$, which is shown in Figure 6a. (If the 6-cycle is thought to be a benzene ring in organic chemistry, then $u$ and $v$ are on *meta-positions*.) It is easy to see that

$$C_6(u) = \frac{1}{2} \left( \sum_{v \in \mathcal{N}_1(u)} C_{2,4}(u, v) + \sum_{v \in \mathcal{N}_2(u)} C_{2,4}(u, v) \right). \tag{88}$$

We calculate $C_{2,4}(u, v)$ as following,

$$C_{2,4}(u, v) = P_2(u, v)P_4(u, v) - \#(\mathrm{b}) - \#(\mathrm{c}) - \#(\mathrm{d}), \tag{89}$$

where $\#(\mathrm{b})$, $\#(\mathrm{c})$ and $\#(\mathrm{d})$ are the numbers of substructures illustrated in Figure 6b, 6c and 6d, respectively. In those figures, we always assume that $u \to x \to y \to z \to v$ is a 4-path and $u \to t \to v$ is a 2-path. Since every 4-path $u \to x \to y \to z \to v$ and every 2-path $u \to t \to v$ with $t \neq x, t \neq y$ and $t \neq z$ contribute one count to $C_{2,4}(u, v)$, it is clear that (89) gives the correct number $C_{2,4}(u, v)$.

We now calculate $\#(\mathrm{b})$, $\#(\mathrm{c})$ and $\#(\mathrm{d})$. For $\#(\mathrm{b})$, we have

$$\#(\mathrm{b}) = \sum_{x \in \mathcal{N}_1(u) \cap \mathcal{N}_1(v)} P_3(x, v) - \#(\mathrm{b}, u = y) - \#(\mathrm{b}, u = z). \tag{90}$$

Here, $\#(\mathrm{b}, u = y)$ or $\#(\mathrm{b}, u = z)$ is the count of substructure obtained by coalescing nodes $u, y$ or $u, z$ in Figure 6b, while keeping $y \neq v$ and $x \neq z$. It is easy to see that

$$\#(\mathrm{b}, u = y) = P_2(u, v)(P_2(u, v) - 1), \tag{91}$$

$$\#(\mathrm{b}, u = z) = \mathbb{1}_{d(u,v)=1} \sum_{x \in \mathcal{N}_1(u) \cap \mathcal{N}_1(v)} (P_2(u, x) - 1). \tag{92}$$

The count $\#(\mathrm{d})$ can be calculated analogously, and we summarize the results below.

$$\#(\mathrm{b}) + \#(\mathrm{d}) = \sum_{x \in \mathcal{N}_1(u) \cap \mathcal{N}_1(v)} (P_3(x, v) + P_3(u, x)) - 2P_2(u, v)(P_2(u, v) - 1)$$

$$- \mathbb{1}_{d(u,v)=1} \sum_{x \in \mathcal{N}_1(u) \cap \mathcal{N}_1(v)} (P_2(u, x) + P_2(x, v) - 2). \tag{93}$$

By Lemmas C.7, C.10 and C.12, there exist 2-DRFWL(2) tests that assign the numbers $P_2(u, v)P_4(u, v)$ and $\#(\mathrm{b}) + \#(\mathrm{d})$ to $W^{(T)}(u, v)$ respectively, for some integer $T \geqslant 4$ and for all pairs $(u, v)$ with $d(u, v) = 1$ or $d(u, v) = 2$. Therefore, both

$$\sum_{v \in \mathcal{N}_1(u)} P_2(u, v)P_4(u, v) + \sum_{v \in \mathcal{N}_2(u)} P_2(u, v)P_4(u, v)$$

and

$$\sum_{v \in \mathcal{N}_1(u)} (\#(\mathrm{b}) + \#(\mathrm{d})) + \sum_{v \in \mathcal{N}_2(u)} (\#(\mathrm{b}) + \#(\mathrm{d}))$$

can be assigned by a specific 2-DRFWL(2) test $q$ to $W^{(T)}(u, u)$ for some $T \geqslant 5$ and for all nodes $u$. To prove the lemma it now suffices to show that

$$\sum_{v \in \mathcal{N}_1(u)} \#(\mathrm{c}) + \sum_{v \in \mathcal{N}_2(u)} \#(\mathrm{c})$$

can be assigned by a 2-DRFWL(2) test to $W^{(T)}(u, u)$ for some $T \geqslant 5$ and for all nodes $u$. We calculate $\#(\mathrm{c})$ as following,

$$\#(\mathrm{c}) = \sum_{x \in \mathcal{N}_1(u) \cap \mathcal{N}_1(v)} P_2(u, x)P_2(x, v) - \#(\mathrm{c}, u = z) - \#(\mathrm{c}, x = v) - \#(\mathrm{c}, x = z)$$

$$- \#(\mathrm{c}, u = z, x = v). \tag{94}$$

Here $\#(\mathrm{c}, u = z)$ is the count of substructure obtained by coalescing nodes $u, z$ in Figure 6c, while keeping $x \neq v$. The counts $\#(\mathrm{c}, x = v)$, $\#(\mathrm{c}, x = z)$ and $\#(\mathrm{c}, u = z, x = v)$ can be understood analogously, but one needs to notice that when counting $\#(\mathrm{c}, x = v)$ we keep $u \neq z$. We have

$$\#(\mathrm{c}, u = z) = \mathbb{1}_{d(u,v)=1} \sum_{z \in \mathcal{N}_1(u) \cap \mathcal{N}_1(v)} (P_2(u, z) - 1), \tag{95}$$

$$\#(\mathrm{c}, x = v) = \mathbb{1}_{d(u,v)=1} CC_1(v, u), \tag{96}$$

$$\#(\mathrm{c}, u = z, x = v) = \mathbb{1}_{d(u,v)=1} P_2(u, v), \tag{97}$$

and

$$\sum_{v \in \mathcal{N}_1(u)} \#(\mathbf{c}, x = z) + \sum_{v \in \mathcal{N}_2(u)} \#(\mathbf{c}, x = z) = CC_1(u). \tag{98}$$

By Lemmas C.7, C.17 and C.18, after summing over all nodes $v \in \mathcal{N}_1(u) \cup \mathcal{N}_2(u)$, each of the terms in (94) can be assigned by a 2-DRFWL(2) test to $W^{(T)}(u, u)$, for some $T \geqslant 5$ and for all nodes $u$. Therefore, we have finally reached the end of the proof. $\qquad\square$

We are now in a position to give the proofs for Theorems 4.4, 4.5, 4.6 and 4.7.

We restate Theorem 4.4 as following.

**Theorem C.20.** *2-DRFWL(2) GNNs can node-level count 2, 3, 4-paths.*

*Proof.* By Lemma C.5, it suffices to prove that for each $k = 2, 3, 4$, there exists a 2-DRFWL(2) test $q$ such that for any graph $G \in \mathcal{G}$ and for any node $u \in \mathcal{V}_G$, $q$ assigns

$$W^{(T)}(u, u) = P_k(u), \tag{99}$$

for some integer $T$.

In the following proofs, we no longer care the exact value of $T$; instead, we are only interested in whether a graph property *can be calculated* by a 2-DRFWL(2) test, i.e. there exists a finite integer $T$ and a subset of 2-DRFWL(2) colors (for example, $W^{(T)}(u, u), u \in \mathcal{V}_G$ for node-level properties, or $W^{(T)}(u, v), d(u, v) \leqslant 2$ for pair-level properties) that the subset of 2-DRFWL(2) colors gives exactly the values of the graph property, at the $T$-th iteration.

For $k = 2$,

$$P_2(u) = \sum_{v \in \mathcal{N}_1(u)} P_2(u, v) + \sum_{v \in \mathcal{N}_2(u)} P_2(u, v). \tag{100}$$

But by Lemma C.7, there exists a 2-DRFWL(2) test $q$ and an integer $T$ such that (39) holds for all pairs $(u, v)$ with $d(u, v) = 1$ or $d(u, v) = 2$. Therefore, at the $(T + 1)$-th iteration, $q$ can assign $W^{(T+1)}(u, u) = P_2(u)$ using (100).

For $k = 3$,

$$P_3(u) = \sum_{v \in \mathcal{N}_1(u)} P_3(u, v) + \sum_{v \in \mathcal{N}_2(u)} P_3(u, v) + \sum_{v \in \mathcal{N}_3(u)} P_3(u, v). \tag{101}$$

Using Lemma C.10 and a similar argument to above, the first two terms of (101) can be calculated by a 2-DRFWL(2) test. For the last term of (101), we notice that

$$\sum_{v \in \mathcal{N}_3(u)} P_3(u, v) = \sum_{v \in \mathcal{N}_3(u)} W_3(u, v)$$

$$= W_3(u) - W_3(u, u) - \sum_{v \in \mathcal{N}_1(u)} W_3(u, v) - \sum_{v \in \mathcal{N}_2(u)} W_3(u, v). \tag{102}$$

By Lemma C.14, $W_3(u)$ can be calculated by a 2-DRFWL(2) test. From the proof of Lemma C.10, one can see that $W_3(u, v)$ can be calculated by a 2-DRFWL(2) test, for $d(u, v) = 1$ or $d(u, v) = 2$. Therefore, the last two terms of (102) can also be calculated by a 2-DRFWL(2) test. Since $W_3(u, u) = 2C_3(u)$, it can be seen from Lemma C.11 that this term can also be calculated by a 2-DRFWL(2) test. Therefore, we conclude that $\sum_{v \in \mathcal{N}_3(u)} P_3(u, v)$, thus $P_3(u)$, can be calculated by a 2-DRFWL(2) test.

For $k = 4$,

$$P_4(u) = \sum_{v \in \mathcal{N}_1(u)} P_4(u, v) + \sum_{v \in \mathcal{N}_2(u)} P_4(u, v) + \sum_{v \in \mathcal{N}_3(u)} P_4(u, v) + \sum_{v \in \mathcal{N}_4(u)} P_4(u, v). \tag{103}$$

Using Lemma C.12 and a similar argument to above, the first two terms of (103) can be calculated by a 2-DRFWL(2) test. Like in the $k = 3$ case, we treat the last two terms of (103) as following,

$$\sum_{v \in \mathcal{N}_3(u)} P_4(u, v) + \sum_{v \in \mathcal{N}_4(u)} P_4(u, v) = \sum_{v \in \mathcal{N}_3(u)} W_4(u, v) + \sum_{v \in \mathcal{N}_4(u)} W_4(u, v)$$
$$= W_4(u) - W_4(u, u)$$
$$- \sum_{v \in \mathcal{N}_1(u)} W_4(u, v) - \sum_{v \in \mathcal{N}_2(u)} W_4(u, v). \qquad (104)$$

By Lemmas C.13 and C.14, the terms $W_4(u)$, $\sum_{v \in \mathcal{N}_1(u)} W_4(u, v)$ and $\sum_{v \in \mathcal{N}_2(u)} W_4(u, v)$ can be calculated by 2-DRFWL(2) tests. Moreover, it is easy to verify that

$$W_4(u, u) = 2C_4(u) + (\deg(u))^2 + P_2(u). \qquad (105)$$

Therefore, by Lemmas C.7, C.8 and C.15, $W_4(u, u)$ can also be calculated by a 2-DRFWL(2) test. Summarizing the results above, we have proved that $P_4(u)$ can be calculated by a 2-DRFWL(2) test. $\qquad \square$

We restate Theorem 4.5 as following.

**Theorem C.21.** *2-DRFWL(2) GNNs can node-level count 3, 4, 5, 6-cycles.*

*Proof.* By Lemma C.5, it suffices to prove that for each $k = 3, 4, 5, 6$, $C_k(u)$ can be calculated by a 2-DRFWL(2) test. From Lemmas C.11, C.15, C.16 and C.19, the above statement is true. $\qquad \square$

We restate Theorem 4.6 as following.

**Theorem C.22.** *2-DRFWL(2) GNNs can node-level count tailed triangles, chordal cycles and triangle-rectangles.*

*Proof.* By Lemmas C.17 and C.18, and using a similar argument to above, it is easy to prove that 2-DRFWL(2) GNNs can node-level count chordal cycles and triangle-rectangles. To see why 2-DRFWL(2) GNNs can node-level count tailed triangles, we only need to give a 2-DRFWL(2) test $q$ that assigns

$$W^{(T)}(u, u) = C(\text{tailed triangle}, u, G), \quad \forall u \in \mathcal{V}_G, \qquad (106)$$

for some integer $T$. Actually, we have

$$C(\text{tailed triangle}, u, G) = \sum_{v \in \mathcal{N}_1(u)} (C_3(v) - P_2(u, v)), \qquad (107)$$

or in a symmetric form,

$$C(\text{tailed triangle}, u, G) = \sum_{v \in \mathcal{N}_1(u)} (C_3(u) + C_3(v) - P_2(u, v)) - C_3(u)\deg(u). \qquad (108)$$

Using this formula, along with Lemmas C.7, C.8 and C.11, it is straightforward to construct the 2-DRFWL(2) test $q$ that satisfies (106), as long as $T \geqslant 3$. Therefore, 2-DRFWL(2) GNNs can also node-level count tailed triangles. $\qquad \square$

We restate Theorem 4.7 as following.

**Theorem C.23.** *2-DRFWL(2) GNNs cannot graph-level count more than 7-cycles or more than 4-cliques.*

*Proof.* The theorem is a direct corollary of Theorem 3.2. In the remark at the end of Appendix B.2, we have shown that no $d$-DRFWL(2) test can distinguish between two $k$-cycles and a single $2k$-cycle, as long as $k \geqslant 3d + 1$. For $d = 2$, we assert that 2-DRFWL(2) test (thus no 2-DRFWL(2) GNN) can distinguish between two $k$-cycles and a $2k$-cycle, as long as $k \geqslant 7$. Therefore, 2-DRFWL(2) GNNs cannot graph-level count more than 7-cycles.

Moreover, Theorem 4.2 of [56] states that for any $k \geqslant 2$, there exists a pair of graphs with different numbers of $(k+1)$-cliques that FWL$(k-1)$ fails to distinguish between. Therefore, we assert that FWL(2) cannot graph-level count more than 4-cliques. By Theorem 3.2, the 2-DRFWL(2) tests (thus 2-DRFWL(2) GNNs) are strictly less powerful than FWL(2) in terms of the ability to distinguish between non-isomorphic graphs. Therefore, 2-DRFWL(2) GNNs cannot graph-level count more than 4-cliques. $\qquad\square$

## C.3 Proof of Theorem 4.8

Before proving the theorem, we state and prove a series of lemmas.

**Lemma C.24.** *There exists a 3-DRFWL(2) test $q$ such that for any graph $G \in \mathcal{G}$ and for any 2-tuple $(u, v) \in \mathcal{V}_G^2$ with $1 \leqslant d(u, v) \leqslant 3$, $q$ assigns*

$$W^{(T)}(u, v) = P_3(u, v), \tag{109}$$

*for some integer $T$.*

*Proof.* If $1 \leqslant d(u, v) \leqslant 2$, the result follows from Lemma C.10 since 2-DRFWL(2) tests constitute a subset of 3-DRFWL(2) tests. If $d(u, v) = 3$, then equations (50)–(54) still hold (with the value of $k$ changed to 3), giving rise to

$$P_3(u, v) = W_3(u, v) = \frac{1}{2} \left( \sum_{w \in \mathcal{N}_1(u) \cap \mathcal{N}_2(v)} P_2(w, v) + \sum_{w \in \mathcal{N}_2(u) \cap \mathcal{N}_1(v)} P_2(u, w) \right), \quad d(u, v) = 3. \tag{110}$$

The RHS of (110) is obviously calculable by 3-DRFWL(2) tests. Therefore, the lemma holds. $\qquad\square$

**Lemma C.25.** *There exists a 3-DRFWL(2) test $q$ such that for any graph $G \in \mathcal{G}$ and for any 2-tuple $(u, v) \in \mathcal{V}_G^2$ with $1 \leqslant d(u, v) \leqslant 3$, $q$ assigns*

$$W^{(T)}(u, v) = P_4(u, v), \tag{111}$$

*for some integer $T$.*

*Proof.* If $1 \leqslant d(u, v) \leqslant 2$, the result follows from Lemma C.12. If $d(u, v) = 3$, then equation (63) still holds, giving rise to

$$P_4(u, v) = P_{2,2}(u, v) = \sum_{1 \leqslant i,j \leqslant 2} P_{2,2}^{ij}(u, v), \quad d(u, v) = 3. \tag{112}$$

The definitions of $P_{2,2}(u, v)$ and $P_{2,2}^{ij}(u, v)$ are given in (57) and (65), respectively. In particular, we point out that

$$P_{2,2}^{ij}(u, v) = \sum_{w \in \mathcal{N}_i(u) \cap \mathcal{N}_j(v)} P_2(u, w) P_2(w, v)$$

is calculable by 3-DRFWL(2) tests, for any $1 \leqslant i, j \leqslant 2$ and for $d(u, v) = 3$. Therefore, the lemma holds. $\qquad\square$

**Lemma C.26.** *There exists a 2-DRFWL(2) test $q$ such that for any graph $G \in \mathcal{G}$ and for any 2-tuple $(u, v) \in \mathcal{V}_G^2$ with $1 \leqslant d(u, v) \leqslant 2$, $q$ assigns*

$$W^{(T_0)}(u, v) = T(u, v), \tag{113}$$

*for some integer $T_0$.*

*Proof.* The lemma follows from

$$T(u, v) = \sum_{w \in \mathcal{N}_1(u) \cap \mathcal{N}_1(v)} P_2(w, v) - 1_{d(u,v)=1} P_2(u, v), \tag{114}$$

and Lemma C.7. $\qquad\square$

**Lemma C.27.** *There exist two 2-DRFWL(2) tests $q_1, q_2$ such that for any graph $G \in \mathcal{G}$ and for any 2-tuple $(u, v) \in \mathcal{V}_G^2$ with $1 \leqslant d(u, v) \leqslant 2$, $q_1, q_2$ assign*

$$W_1^{(T_1)}(u, v) = CC_1(u, v), \quad W_2^{(T_2)}(u, v) = CC_2(u, v), \tag{115}$$

*respectively, for some integers $T_1$ and $T_2$.*

*Proof.* The existence of $q_1$ follows from the proof of Lemma C.17. To see why 2-DRFWL(2) tests can count $CC_2(u, v)$, notice that

$$CC_2(u, v) = \frac{1}{2} P_2(u, v)(P_2(u, v) - 1), \tag{116}$$

and the result follows from Lemma C.7.

Due to the relations (32) and (33), we assert that 2-DRFWL(2) tests can also node-level count $CC_1(u)$ and $CC_2(u)$. □

**Lemma C.28.** *There exist three 2-DRFWL(2) tests $q_1, q_2$ and $q_3$ such that for any graph $G \in \mathcal{G}$ and for any node $u \in \mathcal{V}_G$, $q_1, q_2$ and $q_3$ assign*

$$W_1^{(T_1)}(u, u) = TR_1(u), \quad W_2^{(T_2)}(u, u) = TR_2(u), \quad W_3^{(T_3)}(u, u) = TR_3(u), \tag{117}$$

*respectively, for some integers $T_1, T_2$ and $T_3$.*

*Proof.* From the proof of Lemma C.18 we see that 2-DRFWL(2) tests can count $TR_1(u, v)$. By (34), 2-DRFWL(2) tests can count $TR_1(u)$ and $TR_2(u)$. We only need to prove that 2-DRFWL(2) tests can count $TR_3(u)$.

Notice that

$$\begin{aligned} TR_3(u) &= \sum_{v \in \mathcal{N}_1(u) \cup \mathcal{N}_2(u)} TR_2(v, u) \\ &= \sum_{v \in \mathcal{N}_1(u) \cup \mathcal{N}_2(u)} (P_2(u, v) - 1) \, T(u, v) - 2CC_1(u). \end{aligned} \tag{118}$$

Therefore, by Lemmas C.7, C.26 and C.27, the lemma is true. □

**Lemma C.29.** *There exists a 2-DRFWL(2) test $q$ such that for any graph $G \in \mathcal{G}$ and for any 2-tuple $(u, v) \in \mathcal{V}_G^2$ with $1 \leqslant d(u, v) \leqslant 2$, $q$ assigns*

$$W^{(T)}(u, v) = C_{2,3}(u, v), \tag{119}$$

*for some integer $T$.*

*Proof.* Notice that

$$C_{2,3}(u, v) = P_2(u, v)P_3(u, v) - T(u, v) - T(v, u). \tag{120}$$

Therefore, the lemma follows from Lemmas C.7, C.10 and C.26. □

Now, we turn to proving Theorem 4.8. We restate the theorem as following.

**Theorem C.30.** *For any $d \geqslant 3$, $d$-DRFWL(2) GNNs can node-level count 3, 4, 5, 6, 7-cycles, but cannot graph-level count any longer cycles.*

*Proof.* We first prove the negative result: for any $d \geqslant 3$, $d$-DRFWL(2) GNNs cannot graph-level count $k$-cycles, with $k \geqslant 8$. This is because in terms of the ability to distinguish between non-isomorphic graphs, $d$-DRFWL(2) tests (and thus GNNs) are strictly less powerful than FWL(2), for any $d$, as shown in Theorem 3.2. Nevertheless, even the more powerful FWL(2) tests fail to graph-level count $k$-cycles with $k \geqslant 8$. [4]

Next, we will prove the positive result. Notice that the update rule of $(d + 1)$-DRFWL(2) GNNs always encompasses that of $d$-DRFWL(2) GNNs. Therefore, for any $d \geqslant 3$, the node-level cycle

counting power of $d$-DRFWL(2) GNNs is at least as strong as that of 2-DRFWL(2) GNNs. This implies that $d$-DRFWL(2) GNNs can node-level count 3, 4, 5, 6-cycles for any $d \geqslant 3$.

For the same reason, to prove that $d$-DRFWL(2) GNNs can node-level count 7-cycles for any $d \geqslant 3$, it suffices to prove that 3-DRFWL(2) GNNs can do so. By Lemma C.5, this reduces to proving the existence of a 3-DRFWL(2) test $q$, such that for any graph $G \in \mathcal{G}$ and for any node $u \in \mathcal{V}_G$, $q$ assigns

$$W^{(T)}(u, u) = C_7(u), \tag{121}$$

for some integer $T$.

We try to calculate $C_7(u)$ via

$$C_7(u) = \frac{1}{2} \sum_{v:1 \leqslant d(u,v) \leqslant 3} C_{3,4}(u, v). \tag{122}$$

We assert that

$$C_{3,4}(u, v) = P_3(u, v)P_4(u, v) - \#(\text{a}) - \#(\text{b}) - \cdots - \#(\text{l}), \tag{123}$$

where $\#(\text{a}), \#(\text{b}), \ldots, \#(\text{l})$ refer to numbers of the substructures depicted in (a), (b), ..., (l) of Figure 7, respectively. Since $P_3(u, v)$ and $P_4(u, v)$ are calculable by 3-DRFWL(2) tests (by Lemmas C.24 and C.25), it now suffices to show that 3-DRFWL(2) tests can calculate the counts $\#(\text{a}), \#(\text{b}), \ldots, \#(\text{l})$, summed over all $v$ with $1 \leqslant d(u, v) \leqslant 3$.

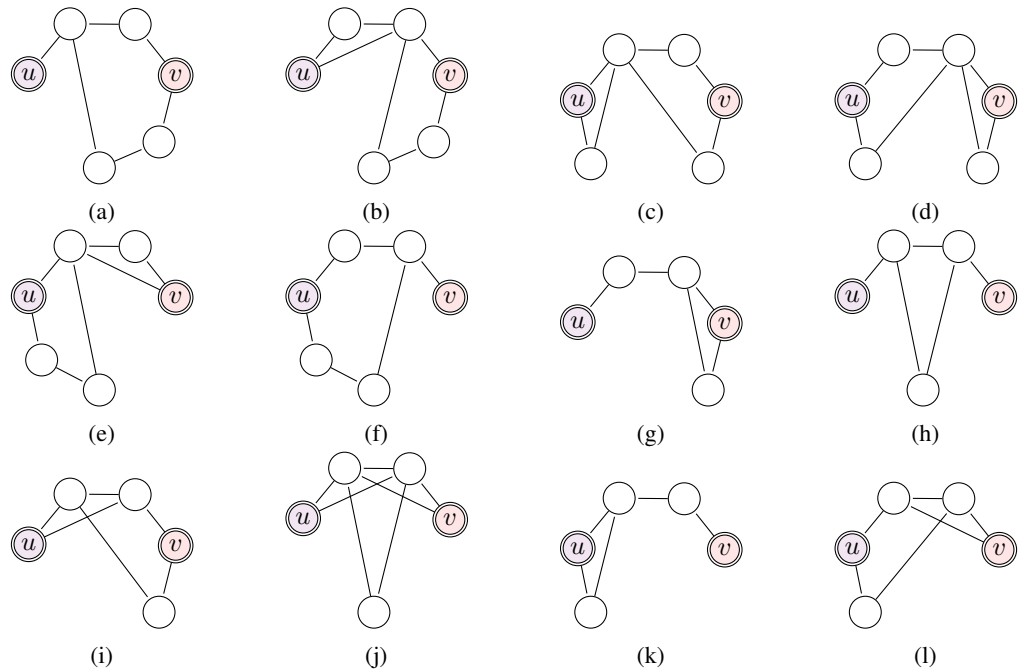

Figure 7: The twelve substructures whose counts should be subtracted from $P_3(u, v)P_4(u, v)$ to get $C_{3,4}(u, v)$.

For the rest of the proof, we express each of the counts $\#(\text{a}), \#(\text{b}), \ldots, \#(\text{l})$ (after summing over $\{v \in \mathcal{V}_G : 1 \leqslant d(u, v) \leqslant 3\}$) in terms of numbers already known calculable by 3-DRFWL(2) tests. Since the enumeration procedure is rather tedious, we directly provide the results below.

$$\sum_{v:1 \leqslant d(u,v) \leqslant 3} \#(\text{a}) = \sum_{v:1 \leqslant d(u,v) \leqslant 2} \sum_{w \in \mathcal{N}_1(u) \cap \mathcal{N}_1(v)} C_{2,3}(w, v)$$
$$+ \sum_{v:1 \leqslant d(u,v) \leqslant 3} \sum_{w \in \mathcal{N}_1(u) \cap \mathcal{N}_2(v)} C_{2,3}(w, v)$$

$$- 4C_5(u) - TR_2(u), \tag{124}$$

$$\sum_{v:1\leqslant d(u,v)\leqslant 3} \#(\mathrm{b}) = \sum_{v:1\leqslant d(u,v)\leqslant 2} \sum_{w\in\mathcal{N}_1(u)\cap\mathcal{N}_1(v)} P_2(u,w)P_3(w,v)$$

$$- \sum_{v\in\mathcal{N}_1(u)} \sum_{w\in\mathcal{N}_1(u)\cap\mathcal{N}_1(v)} P_2(u,w)(P_2(u,w)-1)$$

$$- \sum_{v\in\mathcal{N}_1(u)} \sum_{w\in\mathcal{N}_1(u)\cap\mathcal{N}_1(v)} P_2(w,v)(P_2(w,v)-1)$$

$$- CC_1(u) - 2CC_2(u) - 4TR_1(u) - TR_2(u), \tag{125}$$

$$\sum_{v:1\leqslant d(u,v)\leqslant 3} \#(\mathrm{c}) = \sum_{v:1\leqslant d(u,v)\leqslant 2} \sum_{w\in\mathcal{N}_1(u)\cap\mathcal{N}_1(v)} P_2(u,w)P_2(w,v)(P_2(w,v)-1)$$

$$+ \sum_{v:1\leqslant d(u,v)\leqslant 3} \sum_{w\in\mathcal{N}_1(u)\cap\mathcal{N}_2(v)} P_2(u,w)P_2(w,v)(P_2(w,v)-1)$$

$$- 2TR_2(u) - 4CC_2(u), \tag{126}$$

$$\sum_{v:1\leqslant d(u,v)\leqslant 3} \#(\mathrm{d}) = \sum_{v:1\leqslant d(u,v)\leqslant 2} \sum_{w\in\mathcal{N}_1(u)\cap\mathcal{N}_1(v)} P_2(u,w)P_2(w,v)(P_2(u,w)-1)$$

$$+ \sum_{v:1\leqslant d(u,v)\leqslant 3} \sum_{w\in\mathcal{N}_2(u)\cap\mathcal{N}_1(v)} P_2(u,w)P_2(w,v)(P_2(u,w)-1)$$

$$- \sum_{v\in\mathcal{N}_1(u)} \sum_{w\in\mathcal{N}_1(u)\cap\mathcal{N}_1(v)} P_2(u,w)(P_2(u,w)-1)$$

$$- 4CC_1(u) - 4TR_3(u), \tag{127}$$

$$\sum_{v:1\leqslant d(u,v)\leqslant 3} \#(\mathrm{e}) = \sum_{v:1\leqslant d(u,v)\leqslant 2} \sum_{w\in\mathcal{N}_1(u)\cap\mathcal{N}_1(v)} P_3(u,w)P_2(w,v)$$

$$- 2 \sum_{v\in\mathcal{N}_1(u)} \sum_{w\in\mathcal{N}_1(u)\cap\mathcal{N}_1(v)} P_2(w,v)(P_2(w,v)-1)$$

$$+ 2CC_1(u) - 2CC_2(u) - TR_2(u) - 2TR_3(u), \tag{128}$$

$$\sum_{v:1\leqslant d(u,v)\leqslant 3} \#(\mathrm{f}) = \sum_{v:1\leqslant d(u,v)\leqslant 2} \sum_{w\in\mathcal{N}_1(u)\cap\mathcal{N}_1(v)} C_{2,3}(u,w)$$

$$+ \sum_{v:1\leqslant d(u,v)\leqslant 3} \sum_{w\in\mathcal{N}_2(u)\cap\mathcal{N}_1(v)} C_{2,3}(u,w)$$

$$- 4C_5(u) - TR_3(u), \tag{129}$$

$$\sum_{v:1\leqslant d(u,v)\leqslant 3} \#(\mathrm{g}) = \sum_{v:1\leqslant d(u,v)\leqslant 2} \sum_{w\in\mathcal{N}_1(u)\cap\mathcal{N}_1(v)} P_2(u,w)P_2(w,v)$$

$$+ \sum_{v:1\leqslant d(u,v)\leqslant 3} \sum_{w\in\mathcal{N}_2(u)\cap\mathcal{N}_1(v)} P_2(u,w)P_2(w,v)$$

$$- 2CC_2(u) - C(\text{tailed triangle}, u, G), \tag{130}$$

$$\sum_{v:1\leqslant d(u,v)\leqslant 3} \#(\mathrm{h}) = \sum_{v:1\leqslant d(u,v)\leqslant 2} \sum_{w\in\mathcal{N}_1(u)\cap\mathcal{N}_1(v)} T(u,w)$$

$$+ \sum_{v:1\leqslant d(u,v)\leqslant 3} \sum_{w\in\mathcal{N}_2(u)\cap\mathcal{N}_1(v)} T(u,w)$$

$$- 4C(\text{tailed triangle}, u, G), \tag{131}$$

$$\sum_{v:1\leqslant d(u,v)\leqslant 3} \#(\mathrm{i}) = TR_1(u), \tag{132}$$

$$\sum_{v:1\leqslant d(u,v)\leqslant 3} \#(\mathrm{j}) = \sum_{v\in\mathcal{N}_1(u)} \sum_{w\in\mathcal{N}_1(u)\cap\mathcal{N}_1(v)} P_2(w,v)(P_2(w,v)-1) - 4CC_1(u), \tag{133}$$

$$\sum_{v:1\leqslant d(u,v)\leqslant 3} \#(\text{k}) = \sum_{v:1\leqslant d(u,v)\leqslant 2} \sum_{w\in\mathcal{N}_1(u)\cap\mathcal{N}_1(v)} P_2(u,w)P_2(w,v)$$

$$+ \sum_{v:1\leqslant d(u,v)\leqslant 3} \sum_{w\in\mathcal{N}_1(u)\cap\mathcal{N}_2(v)} P_2(u,w)P_2(w,v)$$

$$- 3\sum_{v:1\leqslant d(u,v)\leqslant 2} T(v,u) - 4C_3(u)(d(u)-2), \tag{134}$$

$$\sum_{v:1\leqslant d(u,v)\leqslant 3} \#(\text{l}) = TR_3(u). \tag{135}$$

Combining the above results, we conclude that the theorem holds. $\qquad\square$

## D Comparison between $d$-DRFWL(2) and localized FWL(2)

As mentioned in Section 5 of the main paper, $d$-DRFWL(2) can be seen as a sparse version of FWL(2). Another representative of this class of methods (adding sparsity to FWL(2)) is the **localized FWL(2)**, proposed by Zhang et al. [58]. In [58], two instances of localized FWL(2) are provided, i.e., LFWL(2) and SLFWL(2). Different from $d$-DRFWL(2), both LFWL(2) and SLFWL(2) assign a color $W(u,v)$ to *every* 2-tuple $(u,v) \in \mathcal{V}_G^2$, for any given graph $G$. Therefore, their space complexity is equal to that of FWL(2), namely $O(n^2)$. The major difference between LFWL(2)/SLFWL(2) and FWL(2) is in their update rules. Similar to FWL(2), both LFWL(2) and SLFWL(2) update $W(u,v)$ using a multiset of the form $\{\!\!\{(W(w,v),W(u,w))\}\!\!\}$. However, for LFWL(2), we restrict $w$ to be in $\mathcal{N}(v)$; for SLFWL(2), we restrict $w$ to be in $\mathcal{N}(u) \cup \mathcal{N}(v)$. Since restricting the range of $w$ reduces the multiset size from $O(n)$ (as in FWL(2)) to $O(\deg)$, the time complexity of LFWL(2) and SLFWL(2) is $O(n^2 \deg)$, which is usually much lower than the $O(n^3)$ time complexity of FWL(2).

We now compare $d$-DRFWL(2) with LFWL(2)/SLFWL(2) in terms of discriminative power. We have the following results:

- **No $d$-DRFWL(2) with a fixed $d$ value can be more powerful than LFWL(2) or SLFWL(2).** This is because for any finite $d$, we can construct graph pairs that are separable by LFWL(2) and SLFWL(2) but not separable by $d$-DRFWL(2). One of such graph pairs can be two $(3d + 1)$-cycles and one $(6d + 2)$-cycle, between which $d$-DRFWL(2) cannot discriminate (as shown in the proof of Theorem 3.2). However, since both LFWL(2) and SLFWL(2) can calculate distance between every pair of nodes (following a procedure constructed in the proof of Theorem 3.2), it is easy for LFWL(2) or SLFWL(2) to distinguish between the aforementioned pair of graphs, since they have different diameters.

- **With sufficiently large $d$, $d$-DRFWL(2) is not less powerful than LFWL(2) or SLFWL(2).** This is because for graphs with diameter $\leqslant d$, $d$-DRFWL(2) has equal power to FWL(2). Since it is shown in [58] that there exist graph pairs separable by FWL(2) but not separable by LFWL(2) or SLFWL(2), once the value of $d$ grows larger than the diameters of such graph pairs, those graph pairs can be separated by $d$-DRFWL(2) but not LFWL(2) or SLFWL(2).

Therefore, for sufficiently large $d$, the discriminative power of $d$-DRFWL(2) is neither stronger nor weaker than LFWL(2) or SLFWL(2). However, it remains unknown the relation in discriminative power between LFWL(2)/SLFWL(2) and $d$-DRFWL(2) with a *practical* $d$ value, such as $d = 2$ or $d = 3$. **Are $d$-DRFWL(2) with those smaller $d$ values less powerful than LFWL(2)/SLFWL(2), or are they incomparable?** We leave this question open for future research.

## E Experimental details

### E.1 Model implementation

Let $G$ be a graph with node features $f_u, u \in \mathcal{V}_G$ and edge features $e_{uv}, \{u, v\} \in \mathcal{E}_G$. In most of our experiments, we implement $d$-DRFWL(2) GNNs as following: we generate $h_{uv}^{(0)}$ as

$$h_{uv}^{(0)} = \begin{cases} \text{LIN}_0(f_u), & \text{if } d(u,v) = 0, \\ \text{LIN}_1(f_u + f_v, e_{uv}), & \text{if } d(u,v) = 1, \\ \text{LIN}_k(f_u + f_v), & \text{if } d(u,v) = k \geqslant 2, \end{cases} \tag{136}$$

where $\text{LIN}_i, i = 0, 1, \ldots, d$ are linear functions. When node and edge features are absent, we assign identical values to $h_{uv}^{(0)}$ with the same $d(u,v)$.

The $m_{ijk}^{(t)}$ and $f_k^{(t)}$ functions in (10) and (11) are chosen as

$$m_{ijk}^{(t)}\left(h_{wv}^{(t-1)}, h_{uw}^{(t-1)}\right) = \text{ReLU}\left(\text{LIN}^{(t)}\left(h_{wv}^{(t-1)} + h_{uw}^{(t-1)}\right)\right), \tag{137}$$

$$f_k^{(t)}\left(h_{uv}^{(t-1)}, \left(a_{uv}^{ijk(t)}\right)_{0\leqslant i,j\leqslant d}\right) = h_{uv}^{(t-1)} + \text{MLP}_k^{(t)}\Bigg((1+\epsilon)h_{uv}^{(t-1)}$$
$$+ \sum_{\substack{|i-j|\leqslant k\leqslant i+j \\ 0\leqslant i,j\leqslant d}} \text{LIN}_{\{\!\{i,j,k\}\!\}}^{(t)}\left(a_{uv}^{ijk(t)}\right)\Bigg), \tag{138}$$

where $\text{LIN}^{(t)}$ in (137) is a linear module **whose parameters are shared among all** $i, j, k$ **combinations**; $\text{LIN}_{\{\!\{i,j,k\}\!\}}^{(t)}$ in (138) is a linear module **whose parameters are shared among all** $i, j, k$ **that form the same multiset** $\{\!\{i, j, k\}\!\}$ (for example, the contributions $a_{uv}^{122}, a_{uv}^{212}$ and $a_{uv}^{221}$ are all linearly transformed by an identical $\text{LIN}_{\{\!\{1,2,2\}\!\}}^{(t)}$); $\text{MLP}_k^{(t)}$ is a multilayer perceptron, and can depend on $k$; $\epsilon$ can be a constant or a learnable parameter. After the $d$-DRFWL(2) GNN layers, we apply a sum-pooling layer

$$R\left(\{\!\{h_{uv}^{(T)} : (u,v) \in \mathcal{V}_G^2 \text{ and } 0 \leqslant d(u,v) \leqslant d\}\!\}\right) = \sum_{0\leqslant d(u,v)\leqslant d} h_{uv}^{(T)}, \tag{139}$$

and then a final MLP.

We point out that our implementation of $d$-DRFWL(2) GNNs has the property $h_{uv}^{(t)} = h_{vu}^{(t)}$, for any $u, v \in \mathcal{V}_G$ with $0 \leqslant d(u,v) \leqslant d$, and for any $t = 0, 1, \ldots, T$. We call an instance of $d$-DRFWL(2) GNN *symmetrized*, if it preserves the above property. It is obvious that symmetrized $d$-DRFWL(2) GNNs only constitute a subspace of the function space of all $d$-DRFWL(2) GNNs.

Nevertheless, we remark that for the case of $d = 2$, **symmetrized 2-DRFWL(2) GNNs have equal node-level cycle counting power to 2-DRFWL(2) GNNs**. Actually, in the proofs of Lemmas C.7–C.19 and Theorems 4.4–4.6 (see Appendix C for details), all the 2-DRFWL(2) tests we constructed have the property $W^{(t)}(u,v) = W^{(t)}(v,u)$, for any $u, v \in \mathcal{V}_G$ with $0 \leqslant d(u,v) \leqslant 2$ and any $t$; on the other hand, it is easy to establish the equivalence between such kind of 2-DRFWL(2) tests and symmetrized 2-DRFWL(2) GNNs. Similar facts can be verified for $d$-DRFWL(2) GNNs with $d = 1$ or $d = 3$. Therefore, at least for all our experiments (where $d \leqslant 3$), our symmetrized implementation of $d$-DRFWL(2) GNNs does no harm to their theoretical cycle counting power, although the model is greatly simplified.

### E.2 Experimental settings

#### E.2.1 Substructure counting

**Datasets.** The synthetic dataset is provided by open-source code of GNNAK on github. The node-level substructure counts are calculated by simple DFS algorithms, which we implement in C language. (This part of code is also available in our repository.)

**Models.** Implementations of all baseline methods (MPNN, ID-GNN, NGNN, GNNAK+, PPGN and I²-GNN) follow [33]. For $d$-DRFWL(2) GNN ($d = 1, 2, 3$), we use 5 $d$-DRFWL(2) GNN layers. In each layer (numbered by $t = 1, \ldots, 5$), $\text{MLP}_k^{(t)}$ is a 2-layer MLP, for $k = 0, 1, \ldots, d$. (See (138) for the definition of $\text{MLP}_k^{(t)}$.) The embedding size is 64.

**Training settings.** We use Adam optimizer with initial learning rate 0.001, and use plateau scheduler with patience 10, decay factor 0.9 and minimum learning rate $10^{-5}$. We train our model for 2,000 epochs. The batch size is 256.

### E.2.2 Molecular property prediction

**Datasets.** The QM9 and ZINC datasets are provided by PyTorch Geometric package [22]. The ogbg-molhiv and ogbg-molpcba datasets are provided by Open Graph Benchmark (OGB) [31]. We rewrite preprocessing code for all four datasets.

The training/validation/test splitting for QM9 is 0.8/0.1/0.1. The training/validation/test splittings for ZINC, ogbg-molhiv and ogbg-molpcba are provided in the original releases.

**Models.** For QM9, we adopt a 2-DRFWL(2) GNN with 5 2-DRFWL(2) GNN layers. In each layer (numbered by $t = 1, \ldots, 5$), $\mathrm{MLP}_k^{(t)}$ is a 2-layer MLP, for $k = 0, 1, 2$. The embedding size is 64.

For ZINC, we adopt a simplified version of 3-DRFWL(2) GNN, which only adds terms $a_{uv}^{313}$, $a_{uv}^{133}$ and $a_{uv}^{331}$ to the update rules (10) and (11) of a 2-DRFWL(2) GNN. Notice that the adopted model architecture is slightly different from the one described in Appendix E.1. This is because we observe that including other kinds of message passing in 3-DRFWL(2) GNN does not improve the performance on ZINC. The adopted 3-DRFWL(2) GNN has 6 3-DRFWL(2) GNN layers. In each layer (numbered by $t = 1, \ldots, 6$), $\mathrm{MLP}_k^{(t)}$ is a 2-layer MLP, for $k = 0, 1, 2, 3$. The embedding size is 64. We apply batch normalization between every two 3-DRFWL(2) GNN layers.

For ogbg-molhiv, we also adopt a simplified version of 2-DRFWL(2) GNN; namely, we remove the term $a_{uv}^{222}$ in the update rules of the original 2-DRFWL(2) GNN. The adopted 2-DRFWL(2) GNN has 5 2-DRFWL(2) GNN layers. In each layer (numbered by $t = 1, \ldots, 5$), $\mathrm{MLP}_k^{(t)}$ is a 2-layer MLP, for $k = 0, 1, 2$. The embedding size is 300. We apply a dropout layer with $p = 0.2$ after every 2-DRFWL(2) GNN layer.

For ogbg-molpcba, we adopt a 2-DRFWL(2) GNN with 3 2-DRFWL(2) GNN layers. In each layer (numbered by $t = 1, 2, 3$), $\mathrm{MLP}_k^{(t)}$ is a 2-layer MLP, with layer normalization applied after every MLP layer, for $k = 0, 1, 2$. The embedding size is 256. We apply layer normalization between every two 2-DRFWL(2) GNN layers, and also apply a dropout layer with $p = 0.2$ after every 2-DRFWL(2) GNN layer.

**Training settings.** For QM9, we use the Adam optimizer, and use plateau scheduler with patience 10, decay factor 0.9 and minimum learning rate $10^{-5}$. For each of the 12 targets on QM9, we search hyperparameters from the following space: (i) initial learning rate $\in \{0.001, 0.002, 0.005\}$; (ii) whether to apply layer normalization between every two 2-DRFWL(2) GNN layers (yes/no). We train our model for 400 epochs. The batch size is 64.

For both ZINC-12K and ZINC-250K, we use Adam optimizer with initial learning rate 0.001. For ZINC-12K, we use plateau scheduler with patience 20, decay factor 0.5 and minimum learning rate $10^{-5}$; for ZINC-250K, we use plateau scheduler with patience 25, decay factor 0.5 and minimum learning rate $5 \times 10^{-5}$. We train our model for 500 epochs on ZINC-12K and 800 epochs on ZINC-250K. The batch size is 128.

For both ogbg-molhiv and ogbg-molpcba, we use Adam optimizer with initial learning rate 0.0005. For ogbg-molhiv, we use step scheduler with step size 20 and decay factor 0.5. We train our model for 100 epochs and the batch size is 64. For ogbg-molpcba, we use step scheduler with step size 10 and decay factor 0.8. We train our model for 150 epochs and the batch size is 256.

## F  Additional experiments

In Appendix F, we present

- Experiments that answer **Q4** in Section 6 of the main paper;
- Ablation studies on the substructure counting power of 2-DRFWL(2) GNNs;

- Additional experiments that study the performance of $d$-DRFWL(2) GNNs on molecular datasets;

- Experiments that study the ability of $d$-DRFWL(2) GNNs to capture long-range interactions;

- Additional experiments that study the cycle counting power and empirical efficiency of 2-DRFWL(2) GNNs on larger graphs.

## F.1 Discriminative power

**Datasets.** To answer **Q4**, we evaluate the discriminative power on three synthetic datasets: (1) EXP [1], containing 600 pairs of non-isomorphic graphs that cannot be distinguished by the WL(1) test; (2) SR25 [6], containing 15 non-isomorphic strongly regular graphs that the FWL(2) test fails to distinguish; (3) BREC [51], containing 400 pairs of non-isomorphic graphs generated from a variety of sources.

For EXP, we follow the evaluation process in [59], use 10-fold cross validation, and report the average binary classification accuracy; for SR25, we follow [60], treat the task as a 15-way classification, and report the accuracy. Raw data for both EXP and SR25 datasets are provided by open-source code of GNNAK on github.

For BREC, we use Reliable Paired Comparison (RPC) as the evaluation method, following [51]. For every graph pair $(G, H)$, the RPC procedure consists of two stages: *major procedure* and *reliability check*.

- In the major procedure, we generate $q = 32$ copies $(G_i, H_i)$ of $(G, H)$, $i = 1, \ldots, q$. The copies $G_i$ and $H_i$ ($i = 1, \ldots, q$) are obtained by randomly relabeling nodes in $G$ and $H$, respectively. We then apply a Hotelling's $T^2$ test to check whether we should reject the null hypothesis $H_0 : \mathbb{E}_{\text{relabel}}[f(G) - f(H)] = 0$, meaning that the model $f$ cannot distinguish between $G$ and $H$.

- In the reliability check, we replace $H_i$ in each copy $(G_i, H_i)$ with $G_i^\pi$, which is obtained by randomly relabeling nodes in $G_i$, for $i = 1, \ldots, q$. We then apply a second $T^2$ test to check whether we should reject the null hypothesis $H_0' : \mathbb{E}_{\text{relabel}}[f(G) - f(G^\pi)] = 0$, meaning that the representations of two isomorphic graphs will not deviate too much from one another due to numerical errors.

Finally, $G$ and $H$ are considered separable by $f$ iff we should reject $H_0$ but should not reject $H_0'$. The detailed description of the RPC procedure is provided in [51].

Table 5: Accuracy on EXP/SR25.

| Method | EXP | SR25 |
|---|---|---|
| GIN | 50% | 6.67% |
| Nested GIN | 99.9% | 6.67% |
| GIN-AK+ | 100% | 6.67% |
| PPGN | 100% | 6.67% |
| 3-GCN | 99.7% | 6.67% |
| $I^2$-GNN | 100% | 100% |
| 2-DRFWL(2) GNN | 99.8% | 6.67% |
| 3-DRFWL(2) GNN | 100% | 6.67% |

**Baselines.** For EXP and SR25, baseline methods are chosen from: (1) Basic MPNNs. These methods have discriminative power upper-bounded by the WL(1) test, and we choose GIN [55], which can achieve WL(1) expressive power theoretically. (2) Subgraph MPNNs. These methods are strictly more powerful than WL(1) but strictly upper-bounded by the FWL(2) test [58]. We choose Nested GIN [59] and GIN-AK+ [60]. (3) Higher-order GNNs with expressive power equal to FWL(2). We choose PPGN [42] and 3-GCN [1]. (4) Methods with discriminative power partially stronger than FWL(2), e.g. $I^2$-GNN [33].

For BREC, the complete list of baseline results is given in Table 2 of [51]. We select 3-WL, NGNN, NGNN with distance encoding (DE), SUN [24], SSWL_P [58], GNN-AK, $I^2$-GNN and PPGN as our baselines.

**Models.** We evaluate both 2-DRFWL(2) GNN and 3-DRFWL(2) GNN on the three datasets. On all three datasets, we adopt a $d$-DRFWL(2) GNN with 5 $d$-DRFWL(2) GNN layers ($d = 2$ or $3$). In each layer (numbered by $t = 1, \ldots, 5$), $\text{MLP}_k^{(t)}$ is a 2-layer MLP, for $k = 0, \ldots, d$. The embedding size is 64 for EXP/SR25, and 32 for BREC.

**Training settings.** For both EXP and SR25 datasets, we use Adam optimizer with learning rate 0.001. We train our model for 10 epochs on EXP and 100 epochs on SR25. The batch size is 20 on EXP and 64 on SR25. For BREC, we use Adam optimizer with learning rate 0.0001. We train for 10 epochs and the batch size is 32.

**Results.** We can see from Table 5 that 2-DRFWL(2) GNN already achieves 99.8% accuracy on the EXP dataset, while 3-DRFWL(2) GNN achieves 100% accuracy. This complies with our theoretical result (Theorem 3.1) that $d$-DRFWL(2) GNNs are strictly more powerful than WL(1); moreover, 3-DRFWL(2) GNN achieves higher accuracy than 2-DRFWL(2) GNN, verifying the expressiveness hierarchy we obtain in Theorem 3.2. Table 5 also shows that both 2-DRFWL(2) GNN and 3-DRFWL(2) GNN fail on SR25 with a 1/15 accuracy. This verifies our assertion that $d$-DRFWL(2) GNNs are strictly less powerful than FWL(2).

Table 6: Numbers of distinguishable graph pairs on BREC.

| Method | Basic (60) | Regular (140) | Extension (100) | CFI (100) | Total (400) |
|---|---|---|---|---|---|
| 3-WL | 60 | 50 | 100 | 60 | 270 |
| NGNN | 59 | 48 | 59 | 0 | 166 |
| NGNN+DE | 60 | 50 | 100 | 21 | 231 |
| SUN | 60 | 50 | 100 | 13 | 223 |
| SSWL_P | 60 | 50 | 100 | 38 | 248 |
| GNN-AK | 60 | 50 | 97 | 15 | 222 |
| I²-GNN | 60 | 100 | 100 | 21 | 281 |
| PPGN | 60 | 50 | 100 | 23 | 233 |
| 2-DRFWL(2) GNN | 60 | 50 | 99 | 0 | 209 |
| 3-DRFWL(2) GNN | 60 | 50 | 100 | 13 | 223 |

The experimental results on BREC are shown in Table 6. Since the 400 graph pairs in BREC are categorized into four classes—basic graphs (60 pairs), regular graphs (140 pairs), extension graphs (100 pairs) and CFI graphs (100 pairs), we report the number of distinguishable graph pairs in each class, and also the total number. From Table 6, it is easy to observe the expressiveness gap between $d$-DRFWL(2) and 3-WL (which has equal discriminative power to FWL(2)), and the gap between 2-DRFWL(2) and 3-DRFWL(2). This again verifies Theorem 3.2.

### F.2 Ablation studies on substructure counting

We study the impact of different kinds of message passing in 2-DRFWL(2) GNNs on their substructure (especially, cycle) counting power, by comparing the full 2-DRFWL(2) GNN with modified versions in which some kinds of message passing are forbidden. The dataset, model hyperparameters and training settings are the same as those in Appendix E.2.1.

Table 7: Ablation studies of node-level counting substructures on synthetic dataset. The colored cell means an error less than 0.01.

| Method | Synthetic (norm. MAE) | | | | | | |
|---|---|---|---|---|---|---|---|
| | 3-Cyc. | 4-Cyc. | 5-Cyc. | 6-Cyc. | Tail. Tri. | Chor. Cyc. | Tri.-Rect. |
| 2-DRFWL(2) GNN | 0.0004 | 0.0015 | 0.0034 | 0.0087 | 0.0030 | 0.0026 | 0.0070 |
| 2-DRFWL(2) GNN (w/o distance-0 tuples) | 0.0006 | 0.0017 | 0.0035 | 0.0092 | 0.0030 | 0.0030 | 0.0070 |
| 2-DRFWL(2) GNN (w/o $a_{uv}^{222}$) | 0.0006 | 0.0013 | 0.0033 | 0.0448 | 0.0029 | 0.0029 | 0.0067 |
| 2-DRFWL(2) GNN (w/o $a_{uv}^{222}, a_{uv}^{122}, a_{uv}^{212}, a_{uv}^{221}$) | 0.0004 | 0.0009 | 0.0852 | 0.0735 | 0.0020 | 0.0020 | 0.0051 |
| 1-DRFWL(2) GNN | 0.0002 | 0.0379 | 0.1078 | 0.0948 | 0.0037 | 0.0013 | 0.0650 |

**Results.** From Table 7, we see that by removing the term $a_{uv}^{222}$ in (11) (i.e., by forbidding a distance-2 tuple $(u, v)$ to receive messages from other two distance-2 tuples $(u, w)$ and $(w, v)$), we make 2-DRFWL(2) GNN unable to node-level count 6-cycles. This complies with our intuitive discussion at the end of Section 4 that such kind of message passing is essential for identifying closed 6-walks, and thus for counting 6-cycles.

Similarly, if we further remove $a_{uv}^{122}$, $a_{uv}^{212}$ and $a_{uv}^{221}$, then 2-DRFWL(2) GNN will be unable to count 5-cycles. If $a_{uv}^{121}$, $a_{uv}^{211}$ and $a_{uv}^{112}$ are also removed, the 2-DRFWL(2) GNN actually degenerates into 1-DRFWL(2) GNN, and becomes unable to count 4-cycles and triangle-rectangles (the latter also containing a 4-cycle as its subgraph). Finally, we can see from Table 7 that even 1-DRFWL(2) GNN can count 3-cycles, tailed triangles and chordal cycles, since the numbers of these substructures only depend on the number of closed 3-walks.

We also study the effect of distance-0 tuples in 2-DRFWL(2) GNN. Removing embeddings for those tuples does not affect the theoretical counting power of 2-DRFWL(2) GNN, but we observe that the empirical performance on counting tasks slightly drops, possibly due to optimization issues.

### F.3    Additional experiments on molecular datasets

In the following, we present the results on ZINC, ogbg-molhiv and ogbg-molpcba datasets.

Table 8: Ten-runs MAE results on ZINC-12K (smaller the better), four-runs MAE results on ZINC-250K (smaller the better), ten-runs ROC-AUC results on ogbg-molhiv (larger the better), and four-runs AP results on ogbg-molpcba (larger the better). The * indicates the model uses virtual node on ogbg-molhiv and ogbg-molpcba.

| Method | ZINC-12K (MAE) | ZINC-250K (MAE) | ogbg-molhiv (AUC) | ogbg-molpcba (AP) |
|---|---|---|---|---|
| GIN* | 0.163±0.004 | 0.088±0.002 | 77.07±1.49 | 27.03±0.23 |
| PNA | 0.188±0.004 | – | 79.05±1.32 | 28.38±0.35 |
| DGN | 0.168±0.003 | – | 79.70±0.97 | 28.85±0.30 |
| HIMP | 0.151±0.006 | 0.036±0.002 | 78.80±0.82 | – |
| GSN | 0.115±0.012 | – | 80.39±0.90 | – |
| Deep LRP | – | – | 77.19±1.40 | – |
| CIN-small | 0.094±0.004 | 0.044±0.003 | 80.05±1.04 | – |
| CIN | 0.079±0.006 | **0.022**±0.002 | **80.94**±0.57 | – |
| Nested GIN* | 0.111±0.003 | 0.029±0.001 | 78.34±1.86 | 28.32±0.41 |
| GNNAK+ | 0.080±0.001 | – | 79.61±1.19 | **29.30**±0.44 |
| SUN (EGO) | 0.083±0.003 | – | 80.03±0.55 | – |
| I$^2$-GNN | 0.083±0.001 | 0.023±0.001 | 78.68±0.93 | – |
| $d$-DRFWL(2) GNN | **0.077**±0.002 | 0.025±0.003 | 78.18±2.19 | 25.38±0.19 |

### F.4    Long-range interactions

In 6.2 of the main paper, we mention that 2-DRFWL(2) GNN greatly outperforms subgraph GNNs like NGNN or I$^2$-GNN on the targets $U_0, U, H$ and $G$ of the QM9 dataset, a phenomenon we ascribe to the stronger ability of 2-DRFWL(2) GNN to capture long-range interactions. Actually, a bit knowledge from physical chemistry tells us that the targets $U_0, U, H$ and $G$ not only depend on the chemical bonds between atoms, but also the intermolecular forces (such as hydrogen bonds). Such intermolecular forces can exist between atoms that are distant from each other on the molecular graph (since the two atoms belong to different molecules). However, most subgraph GNNs process the input graphs by extracting $k$-hop subgraphs around each node, usually with a small $k$. This operation thus prevents information from propagating between nodes that are more than $k$-hop away from one another, and makes subgraph GNNs fail to learn the long-range interactions that are vital for achieving good performance on the targets $U_0, U, H$ and $G$. In contrast, 2-DRFWL(2) GNN directly performs message passing between distance-restricted 2-tuples on the *raw graph*, resulting in its capability to capture such long-range interactions.

So far, it remains an interesting question whether 2-DRFWL(2) GNNs' ability to capture long-range interactions is comparable with that of **globally expressive** models, such as $k$-GNNs [44] or GNNs based on SSWL, LFWL(2) or SLFWL(2) [58]. Those models keep embeddings for *all possible* $k$-tuples $\mathbf{v} \in \mathcal{V}_G^k$ (instead of only distance-restricted ones). In this sense, they treat the input graphs as fully-connected ones, and presumably have stronger capability to learn long-range dependencies than 2-DRFWL(2) GNNs. We verify this assertion by comparing the performance of 2-DRFWL(2)

Table 9: MAE results on QM9 (smaller the better).

| Target | 1-GNN | 1-2-GNN | 1-2-3-GNN | SSWL+ | LFWL(2) | SLFWL(2) | 2-DRFWL(2) GNN |
|---|---|---|---|---|---|---|---|
| $\mu$ | 0.493 | 0.493 | 0.476 | 0.418 | 0.439 | 0.435 | **0.346** |
| $\alpha$ | 0.78 | 0.27 | 0.27 | 0.271 | 0.315 | 0.289 | **0.222** |
| $\varepsilon_{\text{homo}}$ | 0.00321 | 0.00331 | 0.00337 | 0.00298 | 0.00332 | 0.00308 | **0.00226** |
| $\varepsilon_{\text{lumo}}$ | 0.00355 | 0.00350 | 0.00351 | 0.00291 | 0.00332 | 0.00322 | **0.00225** |
| $\Delta\varepsilon$ | 0.0049 | 0.0047 | 0.0048 | 0.00414 | 0.00455 | 0.00447 | **0.00324** |
| $R^2$ | 34.1 | 21.5 | 22.9 | 18.36 | 19.10 | 18.80 | **15.04** |
| ZPVE | 0.00124 | 0.00018 | 0.00019 | 0.00020 | 0.00022 | 0.00020 | **0.00017** |
| $U_0$ | 2.32 | **0.0357** | 0.0427 | 0.110 | 0.144 | 0.083 | 0.156 |
| $U$ | 2.08 | 0.107 | 0.111 | **0.106** | 0.143 | 0.121 | 0.153 |
| $H$ | 2.23 | 0.070 | **0.0419** | 0.120 | 0.164 | 0.124 | 0.145 |
| $G$ | 1.94 | 0.140 | **0.0469** | 0.115 | 0.164 | 0.103 | 0.156 |
| $C_v$ | 0.27 | 0.0989 | 0.0944 | 0.1083 | 0.1192 | 0.1167 | **0.0901** |

Table 10: Four-runs results on Long Range Graph Benchmark. Methods are categorized into: MPNNs, Transformer-based methods, our method.

| Method | Peptides-func (AP ↑) | Peptides-struct (MAE ↓) |
|---|---|---|
| GCN | $0.5930 \pm 0.0023$ | $0.3496 \pm 0.0013$ |
| GCNII | $0.5543 \pm 0.0078$ | $0.3471 \pm 0.0010$ |
| GINE | $0.5498 \pm 0.0079$ | $0.3547 \pm 0.0045$ |
| GatedGCN | $0.5864 \pm 0.0077$ | $0.3420 \pm 0.0013$ |
| GatedGCN+RWSE | $0.6069 \pm 0.0035$ | $0.3357 \pm 0.0006$ |
| Transformer+LapPE | $0.6326 \pm 0.0126$ | $\mathbf{0.2529} \pm 0.0016$ |
| SAN+LapPE | $0.6384 \pm 0.0121$ | $0.2683 \pm 0.0043$ |
| SAN+RWSE | $\mathbf{0.6439} \pm 0.0075$ | $0.2545 \pm 0.0012$ |
| 2-DRFWL(2) GNN | $0.5953 \pm 0.0048$ | $0.2594 \pm 0.0038$ |

GNN on QM9 with that of 1-2-GNN, 1-2-3-GNN, SSWL+ GNN, LFWL(2) GNN and SLFWL(2) GNN, among which the first two are proposed by Morris et al. [44], and the others by Zhang et al. [58]. We also include the result of 1-GNN for reference.

We present the results in Table 9. The baseline results for 1-GNN, 1-2-GNN and 1-2-3-GNN are from [44]. For experiments on SSWL+ GNN, LFWL(2) GNN and SLFWL(2) GNN, we implement the three models following the style of PPGN [42]; namely, we use a $B \times d \times n_{\max} \times n_{\max}$ matrix to store the embeddings of all 2-tuples within a batch, with $B$ being the batch size, $d$ being the size of the embedding dimension, and $n_{\max}$ being the maximal number of nodes among all graphs in the batch.

For each of SSWL+, LFWL(2) and SLFWL(2), we use 5 layers with embedding size 64, and apply layer normalization after each layer. We train for 400 epochs using Adam optimizer with initial learning rate searched from $\{0.001, 0.002\}$, and plateau scheduler with patience 10, decay factor 0.9 and minimum learning rate $10^{-5}$. The batch size is 64. We report the best MAE (mean absolute error) for each target. The experimental details for 2-DRFWL(2) GNN follow Appendix E.2.2.

From Table 9, we see that although 2-DRFWL(2) GNN outperforms all baseline methods on 8 out of the 12 targets, its performance is inferior to globally expressive models on the targets $U_0, U, H$ and $G$. This result implies that the ability of 2-DRFWL(2) GNN to capture long-range interactions is weaker than globally expressive models.

We also evaluate the performance of 2-DRFWL(2) GNN on the Long Range Graph Benchmark (LRGB) [21]. LRGB is a collection of graph datasets suitable for testing GNN models' power to learn from long-range interactions. We select the two graph-level property prediction datasets, Peptides-func and Peptides-struct, from LRGB. The task of Peptides-func is multi-label binary classification, and the average precision (AP) is adopted as the metric. The task of Peptides-struct is multi-label regression, and the mean absolute error (MAE) is adopted as the metric.

For experiments on both datasets, we use 5 2-DRFWL(2) GNN layers with hidden size 120. We train 200 epochs using Adam optimizer with initial learning rate 0.001, and plateau scheduler with patience 10, decay factor 0.5, and minimal learning rate $10^{-5}$. The batch size is 64. We also list all baseline results given in [21], including MPNNs like GCN [36], GCNII [13], GINE [30, 55] and GatedGCN [11], as well as Transformer-based models like fully connected Transformer [49] with Laplacian PE (LapPE) [19, 20] and SAN [38]. The results are shown in Table 10.

From Table 10, we see that 2-DRFWL(2) GNN outperforms almost all MPNNs. Surprisingly, on `Peptides-struct` the performance of 2-DRFWL(2) GNN is even comparable to Transformer-based methods (which inherently capture long-range information by treating the graph as fully connected). Therefore, we conclude that although designed to capture local structural information, our model still possesses the ability to learn long-range dependencies.

### F.5 Additional experiments on cycle counting power and scalability

**Datasets.** To further compare the cycle counting power and scalability of $d$-DRFWL(2) GNNs (especially for the case of $d = 2$) with other powerful GNNs, we perform node-level cycle counting tasks on two protein datasets, ProteinsDB and HomologyTAPE, collected from [29]. We evaluate the performance and empirical efficiency of 2-DRFWL(2) GNN as well as baseline models such as MPNN, NGNN, $I^2$-GNN and PPGN, on both datasets.

The official code of [29] provides raw data for three protein datasets—"Enzymes vs Non-Enzymes", "Scope 1.75" and "Protein Function". All three datasets contain graphs that represent protein molecules, in which nodes represent amino acids, and edges represent the peptide bonds or hydrogen bonds between amino acids. Each of the three datasets contains a number of *directed* graphs, in which only a small portion of edges are directed.

We collect the first two datasets from the three and convert all graphs to undirected graphs by adding inverse edges for the directed edges. Since we are only interested in cycle counting tasks, we remove node features (e.g., amino acid types, spatial positions of amino acids) and edge features (e.g., whether the bond is a peptide bond or a hydrogen bond). The node-level cycle counts are generated by the same method as the one described in Appendix E.2.1. We then rename the two processed datasets as ProteinsDB and HomologyTAPE respectively. (The new names of the datasets correspond to their original .zip file names.)

Some statistics of the two processed datasets are shown in Table 11. We also show the statistics of some other datasets we use in Table 12. From Table 11 and Table 12 we can see that the graphs in ProteinsDB and HomologyTAPE datasets are much larger than those in QM9, ZINC, ogbg-molhiv, ogbg-molpcba or the synthetic dataset, thus suitable for evaluating the scalability of GNN models.

Table 11: Statistics of two protein datasets.

| Dataset | Number of graphs | Average number of nodes | Average number of edges |
|---------|------------------|-------------------------|-------------------------|
| ProteinsDB | 1,178 | 475.9 | 714.8 |
| HomologyTAPE | 16,292 | 167.3 | 256.7 |

Table 12: Statistics of other datasets we use.

| Dataset | Number of graphs | Average number of nodes | Average number of edges |
|---------|------------------|-------------------------|-------------------------|
| Synthetic | 5,000 | 18.8 | 31.3 |
| QM9 | 130,831 | 18.0 | 18.7 |
| ZINC-12K | 12,000 | 23.2 | 24.9 |
| ZINC-250K | 249,456 | 23.2 | 24.9 |
| ogbg-molhiv | 41,127 | 25.5 | 27.5 |
| ogbg-molpcba | 437,929 | 26.0 | 28.1 |

Table 13: Average normalized MAE results of node-level counting cycles on ProteinsDB and HomologyTAPE datasets. The colored cell means an error less than 0.002 (for ProteinsDB) or 0.0002 (for HomologyTAPE). "Out of GPU Memory" means the method takes an amount of GPU memory more than 24 GB.

| Method | ProteinsDB | | | | HomologyTAPE | | | |
|---|---|---|---|---|---|---|---|---|
| | 3-Cycle | 4-Cycle | 5-Cycle | 6-Cycle | 3-Cycle | 4-Cycle | 5-Cycle | 6-Cycle |
| MPNN | 0.422208 | 0.223064 | 0.254188 | 0.203454 | 0.185634 | 0.162396 | 0.211261 | 0.158633 |
| NGNN | **0.000070** | **0.000189** | 0.003395 | 0.009955 | 0.000003 | 0.000016 | 0.003706 | 0.002541 |
| $I^2$-GNN | | Out of GPU Memory | | | 0.000002 | 0.000010 | 0.000069 | **0.000177** |
| PPGN | | Out of GPU Memory | | | | Out of GPU Memory | | |
| 2-DRFWL(2) GNN | 0.000077 | 0.000192 | **0.000317** | **0.001875** | **0.000001** | **0.000007** | **0.000021** | 0.000178 |

Table 14: Empirical efficiency on ProteinsDB and HomologyTAPE datasets. "OOM" means the method takes an amount of GPU memory more than 24 GB.

| Method | ProteinsDB | | | HomologyTAPE | | |
|---|---|---|---|---|---|---|
| | Memory (GB) | Pre. (s) | Train (s/epoch) | Memory (GB) | Pre. (s) | Train (s/epoch) |
| MPNN | 2.60 | 235.7 | 0.597 | 1.99 | 243.8 | 5.599 |
| NGNN | 16.94 | 941.8 | 2.763 | 8.44 | 1480.7 | 15.249 |
| $I^2$-GNN | OOM | 1293.4 | OOM | 21.97 | 3173.6 | 38.201 |
| PPGN | OOM | 235.7 | OOM | OOM | 243.8 | OOM |
| 2-DRFWL(2) GNN | 8.11 | 1843.7 | 3.809 | 3.82 | 2909.3 | 30.687 |

For ProteinsDB, we use 10-fold cross validation and report the average normalized MAE for each counting task. We use the splitting provided in the raw data.

For HomologyTAPE, there are three test splits in the raw data, called "test_fold", "test_family" and "test_superfamily" respectively. The training/validation/test_fold/test_family/test_superfamily splitting is 12,312/736/718/1,272/1,254. We report the weighted average normalized MAE on the three test splits for each counting task.

**Models.** We adopt MPNN, NGNN, $I^2$-GNN and PPGN as our baselines. The implementation details of 2-DRFWL(2) GNN as well as all baseline methods follow Appendix E.2.1, except that (i) for NGNN and $I^2$-GNN, the subgraph height is 3 for all tasks, and (ii) for PPGN, the number of PPGN layers is 5 and the embedding size is 64.

**Training settings.** We use Adam optimizer with initial learning rate 0.001, and use plateau scheduler with patience 10, decay factor 0.9 and minimum learning rate $10^{-5}$. We train all models for 1,500 epochs on ProteinsDB and 2,000 epochs on HomologyTAPE. The batch size is 32.

**Results.** From Table 13, we see that 2-DRFWL(2) GNN achieves relatively low errors (average normalized MAE $< 0.002$ for ProteinsDB, and $< 0.0002$ for HomologyTAPE) for counting all 3, 4, 5 and 6-cycles on both datasets. This again verifies our theoretical result (Theorem 4.5). Although $I^2$-GNN and PPGN are also provably capable of counting up to 6-cycles, PPGN fails to scale to either ProteinsDB or HomologyTAPE dataset, while $I^2$-GNN fails to scale to ProteinsDB. Therefore, 2-DRFWL(2) GNN maintains good scalability compared with other existing GNNs that can count up to 6-cycles.

We also measure (i) the maximal GPU memory usage during training, (ii) the preprocessing time, and (iii) the training time per epoch, for all models on both datasets. The result is shown in Table 14. We again see that 2-DRFWL(2) GNN uses much less GPU memory while training, compared with subgraph GNNs. Besides, the training time of 2-DRFWL(2) GNN is shorter than that of $I^2$-GNN, while the preprocessing time is comparable.

From Table 14, we also see one limitation of our implementation of 2-DRFWL(2) GNN, which we have mentioned in Section 7—the preprocessing of 2-DRFWL(2) GNN is not efficient enough on larger graphs or graphs with a larger average degree. We believe this limitation can be addressed by

developing more efficient, parallelized operators for the preprocessing of 2-DRFWL(2) GNN, and we leave the exploration of such implementations to future work.

