# OpenReview forum: "Distance-Restricted Folklore Weisfeiler-Leman GNNs with Provable Cycle Counting Power"
_NeurIPS.cc/2023/Conference — NeurIPS 2023 spotlight_

### Official Review · Reviewer_TSJX · 2023-06-27

**Soundness:** 2 fair
**Presentation:** 3 good
**Contribution:** 2 fair
**Rating:** 6
**Confidence:** 2

**Summary:**

The paper proposes a version of the 2-dimensional Weisfeiler-Leman algorithm that restricts the distance between nodes in the node tuples processed. A GNN based on this restriction is also proposed and investigated regarding its ability to count cycles and other substructures.

**Strengths:**

**S1** In general, the paper is well written and clearly structured.

**S2** The idea to restrict the distance is interesting and seems to work reasonably well for counting substructures.

**Weaknesses:**

**W1** The description of the main contribution d-DRFWL(2) could be improved, making the illustration clearer. The general concept behind d-DRFWL(2) could be explained better. What exactly is updated with what information? The formula represents it, but the corresponding text and the example (Figure 1) could be more precise. It does not help that the figure is colored and the colors are referenced. I cannot distinguish red/violet/pink here.

**W2** The theoretical investigation can be extended to show how the approach relates to k-WL and $\delta$-k-LWL.

**W3** More benchmark datasets should be used for evaluation.

**Questions:**

**Q1** Is the approach as expressive as FWL(2) theoretically?

**Q2** How does the proposed approach fit into the k-WL/ $\delta$-k-LWL hierarchy?

**Q3** Can you state the differences and connections to $\delta$-k-LWL more clearly?

**Q4** In d-DRFWL(k) what happens for larger values of d and k for theoretical expressiveness, running time and classification accuracy?

### Other remarks:
The spacing between the lines is off at some places (for example l108/109 where characters from different lines overlap).

**Limitations:**

To me the theoretical expressiveness did not get discussed to the right extend (see Q2).

---

> ### Author Rebuttal · Authors · 2023-08-10
>
> Thank you for your insightful comments and suggestions. We respond to all weaknesses and questions below.
>
> **Reply to W1.**
>
> Thanks for the valuable suggestion. To understand the concept behind $d$-DRFWL(2), one has to first understand FWL(2). FWL(2) works by (i) assigning each 2-tuple $(u,v)\in\mathcal{V}_G^2$ a representation $W(u,v)$; (ii) updating the representation $W(u,v)$ iteratively. The update of $W(u,v)$ relies on an additional node $w$, and information is collected from both $(u,w)$ and $(w,v)$; afterwards, the above-collected information from all $w\in\mathcal{V}_G$ is gathered into a multiset, and the multiset is hashed into value $W(u,v)$ as the updated representation of $(u,v)$.
>
> $d$-DRFWL(2) differs from FWL(2) in two points: (i) only $(u,v)$ with a limited distance $d(u,v)\leqslant d$ are given a representation; (ii) when updating $W(u,v)$, only information from $w$ with both $d(u,w)\leqslant d$ and $d(w,v)\leqslant d$ is gathered into the multiset. From the above description, one can see $d$-DRFWL(2) as a pruned version of FWL(2), in which representations for node pairs $(u,v)$ with $d(u,v)>d$ are deleted.
>
> So far we have explained how $d$-DRFWL(2) works, what information is collected and what is updated in $d$-DRFWL(2). Now let us explain Figure 1, which illustrates 2-DRFWL(2). As explained above, 2-DRFWL(2) first assigns a representation to all pair of nodes with mutual distance $\leqslant 2$. Then, these representations will be updated. Since in Figure 1 we have $d(u,v)=2$, $(u,v)$ is among the 2-tuples which will be assigned a representation. We exclusively focus on the update procedure of $W(u,v)$ in Figure 1. From point (ii), only those nodes $w$ with $d(u,w)\leqslant 2$ and $d(w,v)\leqslant 2$ play a role in the update procedure. Therefore, $w$ can be any of $u,v,x,y,z,t,r$ in Figure 1. We further classify the seven nodes based on their distances to $u$ and to $v$:
>
> |Nodes|Distance to $u$|Distance to $v$|Color|
> |:-:|:-:|:-:|:-:|
> |u|0|2|violet|
> |v|2|0|red|
> |x, y|1|1|green|
> |z|1|2|blue|
> |t|2|1|yellow|
> |r|2|2|pink|
>
> Each of these nodes contributes a piece of information like $(W(u,w), W(w,v))$, and they are then gathered to update $W(u,v)$. The colored lines in Figure 1 are simply depictions for node pairs $(u,w)$ and $(w,v)$, with $w\in\{u,v,x,y,z,t,r\}$.
>
> We will further clarify the text describing $d$-DRFWL(2), and change the colors in Figure 1 to more easily distinguishable ones in our revision.
>
> **Reply to W2.**
>
> Thanks for the suggestion. Since 2-WL is equally expressive as WL(1), and 3-WL is equally expressive as FWL(2) [a], our methods strictly lie between 2-WL and 3-WL.
>
> As for the $\delta$-$k$-LWL hierarchy, it is known from [b] that for $k=2$, $\delta$-2-LWL is as expressive as SSWL proposed in [b]. However, the relation in expressiveness between SSWL (or $\delta$-2-LWL) and $d$-DRFWL(2) is unknown for arbitrary $d$.
>
> Actually, for any finite $d$, two $(3d+1)$-cycles and one $(6d+2)$-cycle can be separated by SSWL($\delta$-2-LWL) but not $d$-DRFWL(2). This shows that **no $d$-DRFWL(2) with a fixed $d$ value can be more powerful than SSWL($\delta$-2-LWL)**. On the other hand, for graphs with diameter $\leqslant d$, $d$-DRFWL(2) has equal power to FWL(2). Since it is shown in [a] that there exist graph pairs separable by FWL(2) but not SSWL, with d becoming larger than the diameter of such graph pairs, $d$-DRFWL(2) can separate that pair of graphs. This shows that **with sufficiently large $d$, the power of $d$-DRFWL(2) is neither stronger nor weaker than SSWL($\delta$-2-LWL)**. Nevertheless, it remains an open question the relation in discriminating power between SSWL($\delta$-2-LWL) and $d$-DRFWL(2) **with a smaller $d$ value**, such as $d=2$ or $d=3$.
>
> Finally, since for larger $k$ values, the precise relationship in expressiveness between $\delta$-k-LWL and $k$-WL has not been established, the relation between $d$-DRFWL(2) and $\delta$-k-LWL also remains an open problem.
>
> [a] N. T. Huang and S. Villar. A short tutorial on the weisfeiler-lehman test and its variants. In ICASSP 2021 - 2021 IEEE International Conference on Acoustics, Speech and Signal Processing (ICASSP)
> [b] B. Zhang, G. Feng, Y. Du, D. He, and L. Wang. A complete expressiveness hierarchy for subgraph gnns via subgraph weisfeiler-lehman tests. arXiv preprint arXiv:2302.07090, 2023.
>
> **Reply to W3.**
>
> Thanks for the suggestion. We further evaluate 2-DRFWL(2) GNN on the two graph-level property prediction datasets, `Peptides-func` and `Peptides-struct` of Long Range Graph Benchmarks (LRGB) [a]. For experiments on both datasets, we use 5 2-DRFWL(2) GNN layers with hidden size 120. We train 200 epochs using Adam optimizer with initial learning rate 0.001, and plateau scheduler with patience 10, decay rate 0.5, and minimal learning rate 1e-5. The batch size is 64. The results are shown in Table 4 of the PDF.
>
> From the table, we see that our method outperforms almost all message passing GNNs, and even achieves comparable performance to transformer-based models on `Peptides-struct`. Therefore, we conclude that although designed to capture local structural information, our model still learns long-range dependencies well.
>
> [a] V. P. Dwivedi, L. Rampášek, M. Galkin, A. Parviz, G. Wolf, A. T. Luu, D. Beaini, Long Range Graph Benchmark
>
> **Reply to Questions**
>
> Due to space limit, we only provide brief responses to your questions. We are willing to append the full responses during the author-reviewer discussion.
>
> *Q1.* No, our method is strictly less powerful than FWL(2)
>
> *Q2.* Our method lies strictly between WL(2) and WL(3), while the comparison with $\delta$-k-LWL is left open
>
> Due to space limit, we have to defer our response to the other questions in the author-reviewer discussion phase. We are sorry for the inconvenience, and will add the remaining response immediately when the discussion period begins.

---

> > ### Author Response · Authors · 2023-08-12
> > **Author Response to Reviewer TSJX, continued**
> >
> > We now respond to the questions posed by Reviewer TSJX. Those responses are absent in the our rebuttal due to space limit.
> >
> > **Reply to Q1.**
> >
> > The answer is negative. In Theorem 3.2 of our paper, we have established the strict separation result between the power of FWL(2) and $d$-DRFWL(2), for any $d$.
> >
> > The intuition is that since $d$-DRFWL(2) only keeps the representations for 2-tuples $(u,v)$ with $d(u,v)\leqslant d$, information from 2-tuples $(u,v)$ with $u$ and $v$ sufficiently distant (compared with $d$) is inevitably lost. On the contrary, the update rule of FWL(2) is global, meaning that the information is gathered from the entire graph instead of the vicinity of the current 2-tuple. Therefore, FWL(2) can capture global information while $d$-DRFWL(2) cannot. This accounts for their expressiveness gap.
> >
> > **Reply to Q2.**
> >
> > See our response to **W2**.
> >
> > **Reply to Q3.**
> >
> > Both our method and $\delta$-k-LWL are trying to leverage sparsity to accelerate higher-order algorithms. However, there are two major differences between our method and $\delta$-k-LWL. First, our method is built upon FWL(2), while $\delta$-k-LWL is built upon k-WL. This results in different update rules between our method and $\delta$-k-LWL. In our method, a 2-tuple $(u,v)$ receives information of the form $(W(u,w),W(w,v))$, where $w$ is any node that has distance $\leqslant d$ to both $u$ and $v$. In $\delta$-k-LWL, a k-tuple $(u_1,\ldots, u_k)$ receives information from other k-tuples of the form $(u_1,\ldots, u_{j-1},w,u_{j+1},\ldots, u_k)$, where $w$ is a neighbor of $u_j$. Second, our model reduces both the messages to pass and the representations to store. In contrast, $\delta$-k-LWL only reduces the messages to pass, but the number of representations to store is not reduced. Therefore, both our model's time complexity and space complexity is lower than FWL(2), while $\delta$-k-LWL only lowers the time complexity of k-WL.
> >
> > **Reply to Q4.**
> >
> > If $k=2$ but $d$ increases, then the expressive power of d-DRFWL(2) strictly increases, as stated in Theorem 3.2 of our paper. The running time grows as $O(n\ \mathrm{deg}^{2d})$, where $n$ and $\mathrm{deg}$ are the number of nodes and the average degree, respectively. Generally, there is no simple relation between the value of $d$ and the classification accuracy, because the accuracy depends not only on the model expressiveness, but also data.
> >
> > If $k$ increases, the definition of "mutual distance" of a $k$-tuple for $k>2$ is not obvious. If the "mutual distance" within a k-tuple is defined as the maximal distance between nodes in it, then the resulting d-DRFWL(k) will form a similar expressiveness hierarchy both for increasing k and increasing d. The running time will grow as $O(n\ \mathrm{deg}^{kd})$. As above, no simple conclusion for classification accuracy can be drawn.
> >
> > **Reply to Other remarks**
> >
> > Thanks for the suggestion. We will fix it in our revision.
> >
> > **Reply to Limitations**
> >
> > We acknowledge that **Q2** has not been fully addressed. While we keep working on the comparison in expressiveness between $d$-DRFWL(2) GNN and other expressiveness hierarchies, we point out that this is not our major concern, as stated in the first point of our general response.

---

> > > ### Comment · Reviewer_TSJX · 2023-08-13
> > >
> > > I thank the authors for their detailed answers and explanations. Especially the clarifications on Figure 1 and the connections to $k$-WL were very insightful.
> > > The additional findings about the cycle counting power for $d \geq 3$ are really interesting and should be included in the final version.
> > >
> > > I updated my score accordingly.

---

> > > > ### Author Response · Authors · 2023-08-13
> > > >
> > > > We thank the reviewer for thoroughly reading our responses and giving the valuable comments. We will include the additional theoretical findings on the cycle counting power of $d$-DRFWL(2) with $d\geqslant 3$, as well as the empirical verification, into our revised version of the paper. Also, Figure 1 and its description will be made more discernible.

---

### Official Review · Reviewer_pvsr · 2023-07-04

**Soundness:** 3 good
**Presentation:** 3 good
**Contribution:** 3 good
**Rating:** 6
**Confidence:** 3

**Summary:**

The paper deals with constructing expressive GNNs (i.e., more expressive than 1-WL) with low complexity. The authors focus on cycle count ability, motivated by their importance in real life chemical datasets. With this problem and motivation at hand, the authors propose a family of algorithms possessing a strict hierarchy property called $d$-distcne restricted FWL or $d$-DRFWL(2).  They identify the ability to count cycles in the tuple based coloring and aggregation of the $2$-FWL algorithm compared to MPNNs which are bounded by $1$-WL and cannot even count a 3-cycle, but aim to avoid the $O(n^2)$ and $O(n^3)$ memory and time complexity respectively of $2$-FWL. To that end, they propose only coloring 2-tuples $(u,v)$ which are far from each other by ad most $d$ and propose a distance-dependent aggregation, this algorithm is named $d$-DRFWL(2). They then show that already for $d=2$ the proposed algorithm can count up to 6-cycles and that increasing $d$ strictly increases discriminative power. The proposed algorithm is the most efficient known model that can count 6-cycles, with time and memory complexities of $O(n\mathrm{deg}^4)$ and $O(n\mathrm{deg}^2)$, respectively.



**Strengths:**

The authors clearly present the problem, goal, and motivation for their work.

On the originality and novelty side, the authors design the most efficient model to this date that can count 6-cycles and show a thorough analysis of one of its instances, the $2$-DRFWL(2).

Due to the importance of cycle counts in chemical and biological structures, I think that this work is valuable to that community.



**Weaknesses:**

- Although taking a step towards more efficient and expressive GNNs, the efficiency of the method is only when graphs are sparse enough. To my understanding, the proposed method still did not break the limit of running on graphs of sizes $O(100)$ (e.g, ogbg-ppa dataset).
- Do the authors have any explanation for the large performance gap between their method and $I^2$-GNN on the QM9 dataset? Theoretically, they both can count the same sizes of cycles, so this clouds the motivation for the paper in my opinion, since cycle counting may not be what makes this method more successful.



**Questions:**

- Have the authors considered a baseline model where they add the cycle/substructure counts as features? Is the proposed method indeed better then the naive approach?
- An ablation that I find missing is to see the empirical effect of $d$. Have the authors experimented with larger $d$ values?
- How does the proposed hierarchy compare to LFWL(2) and SLFWL(2)? Due to this approaches being closely related, coloring 2-tuples in lower complexity, a more elaborate discussion would add to the work. Also, either theoretical or experimental comparison would be good.

**Limitations:**

The authors discuss the limitations of their work.

---

> ### Author Rebuttal · Authors · 2023-08-10
>
> Thank you for your thoughtful and thorough comments. We respond to all points below.
>
> **Reply to W1.**
>
> We do agree with your comment that the efficiency of our model is only guaranteed when graphs are sparse. Still, our model can handle graphs with even $n\approx 500$ nodes, as long as the average degree $\mathrm{deg}\sim O(1)$, as is verified by our experiments on ProteinsDB and HomologyTAPE datasets.
>
> On graphs with $n$ nodes and average degree $\mathrm{deg}$, $d$-DRFWL(2) GNN uses $O(n\ \mathrm{deg}^d)$ space and $O(nd\ \mathrm{deg}^{2d})$ time to store and update the embeddings for distance-restricted 2-tuples. Therefore, it is in general $\mathrm{deg}$ instead of $n$ that has a larger impact on our model's efficiency. This fact accounts for the failure of $d$-DRFWL(2) GNN on ogbg-ppa, where the average degree is about 10. The search for efficient and powerful methods pertaining to dense graphs remains open. However, we also notice that practical molecule datasets (where cycle counting matters most) only have an average degree between 2 and 3, making our model feasible in most cases.
>
> **Reply to W2.**
>
> Thanks for the valuable question. Below we list the relative performance gain, computed by
> $$\frac{\text{I}^2\text{-GNN MAE}-\text{2-DRFWL(2) GNN MAE}}{\text{I}^2\text{-GNN MAE}}$$
>
> |Target | Relative Performance Gain|
> |:---:|:---:|
> |$\mu$|0.192|
> |$\alpha$|0.035
> |$\varepsilon_\text{homo}$|0.134|
> |$\varepsilon_\text{lumo}$|0.157|
> |$\Delta\varepsilon$|0.147|
> |$R^2$|0.193|
> |$\text{ZPVE}$|-0.214|
> |$U_0$|0.261|
> |$U$|0.257|
> |$H$|0.461|
> |$G$|0.402|
> |$C_v$|-0.234|
>
> We see that the largest relative performance gains of 2-DRFWL(2) GNN compared with I$^2$-GNN occur on targets $U_0,U,H,G$ (each with gain $\geqslant$ 25%). These targets are macroscopic thermodynamic properties of molecules. For such properties, inter-molecular interactions (such as hydrogen bonds), as well as interactions between molecules and the environment can be as important as interactions within molecules. Such interactions may occur between two atoms that are graph-theoretically distant (for example, between the “head” of a molecule and the “tail” of a nearby molecule). Therefore, subgraph extraction operation inhibits information from propagating between such atom pairs. Although our models are also local, they still allow long-range message passing by running for multiple iterations. Therefore, we suspect that it is our model's stronger ability to capture long-range interactions that leads to the observed performance gap with I$^2$-GNN.
>
> **Reply to Q1.**
>
> We have included "Graph Substructure Networks" (GSN) [a] as a baseline model on ZINC-12k and ogbg-molhiv datasets. GSN augments the graph with node-level and edge-level induced substructure counts, and then runs a message passing GNN. As is shown in the following tables, the performance of our model is either better than or comparable with GSN on the two benchmarks.
>
> | Method |   ZINC-12k (MAE) |
> | :---:  |   :---:          |
> |GSN| 0.115 $\pm$ 0.012 |
> |$d$-DRFWL(2) GNN |  **0.077** $\pm$ 0.002 |
>
> | Method | ogbg-molhiv (ROC-AUC) |
> | :---: |:---:|
> |GSN (GIN+VN base)| 0.7799 $\pm$ 0.0100 |
> |GSN (DGN+substructures)| **0.8039** $\pm$ 0.0090|
> |$d$-DRFWL(2) GNN | 0.7818 $\pm$ 0.0219 |
>
> [a] Giorgos Bouritsas, Fabrizio Frasca, Stefanos Zafeiriou, and Michael M Bronstein. Improving graph neural network expressivity via subgraph isomorphism counting. IEEE Transactions on Pattern Analysis and Machine Intelligence, 45(1):657–668, 2022.
>
> **Reply to Q2.**
>
> Thanks for the question. We have included a comprehensive study of the effect of larger d values in the 2nd point of our general response.
>
> **Reply to Q3.**
>
> Thanks for the in-depth question. For any finite d, we can construct graph pairs that are separable by LFWL(2)/SLFWL(2) but not d-DRFWL(2). One of such graph pairs can be two $(3d+1)$-cycles and one $(6d+2)$-cycle, between which d-DRFWL(2) cannot discriminate. However, both LFWL(2) and SLFWL(2) can calculate distance between every pair of nodes, e.g. by simulating the Bellman-Ford shortest path algorithm. Therefore, it is easy for LFWL(2)/SLFWL(2) to distinguish between the aforementioned pair of graphs, since they have different diameters. This shows that **no d-DRFWL(2) with a fixed d value can be more powerful than LFWL(2)/SLFWL(2)**.
>
> On the other hand, for graphs with diameter $\leqslant d$, d-DRFWL(2) has equal power to FWL(2). Since it is shown in [a] that there exist graph pairs separable by FWL(2) but not LFWL(2)/SLFWL(2), once the value of $d$ grows larger than the diameter of such graph pairs, those graph pairs can be separated by d-DRFWL(2). This shows that **with sufficiently large d, the power of d-DRFWL(2) is neither stronger nor weaker than LFWL(2)/SLFWL(2)**.
>
> However, it remains underinvestigated the relation in discriminating power between LFWL(2)/SLFWL(2) and d-DRFWL(2) **with a practical d value**, such as d=2 or d=3. **Are d-DRFWL(2) with those smaller d values less powerful than LFWL(2)/SLFWL(2), or are they incomparable?** In an attempt to tackle this problem computationally, we implement in Python programs the construction procedure of the generalized Furer graph pairs described in [a]. It is claimed in [a] that the generalized Furer graph pairs constructed from Figure 9, 10, 11 of [a] cannot be separated by LFWL(2), while generalized Furer graph pairs constructed from Figure 10 of [a] cannot be separated by SLFWL(2). However, our experiments show that none of the mentioned generalized Furer graph pairs can be separated by d-DRFWL(2), with d=2 or d=3. Therefore, the question raised above remains open for future research.
>
> [a] B. Zhang, G. Feng, Y. Du, D. He, and L. Wang. A complete expressiveness hierarchy for subgraph gnns via subgraph weisfeiler-lehman tests. arXiv preprint arXiv:2302.07090, 2023.

---

> > ### Comment · Reviewer_pvsr · 2023-08-14
> >
> > I thank the authors for their response.
> >
> > I recommend adding the discussion on **Q3** to the revised version as well as the discussion on $d>2$..

---

> > > ### Author Response · Authors · 2023-08-15
> > >
> > > We thank the reviewer for thoroughly reading our responses and giving the constructive suggestions. We will include the discussion on **Q3** as well as the additional discussion on $d$-DRFWL(2) with $d>2$ in our revised version.

---

### Official Review · Reviewer_9V2p · 2023-07-05

**Soundness:** 3 good
**Presentation:** 3 good
**Contribution:** 3 good
**Rating:** 7
**Confidence:** 4

**Summary:**

The paper focuses on the task of counting or detecting substructures such as cycles or small cliques in graphs. This task is (theoretically and practically) impossible for standard GNNs. While higher-order GNNs are able to count such substructures, they are slow, using $n^2$ space and $n^3$ time which is infeasible for large graphs. The paper suggests to make such higher-order GNNs practical by only considering "local" pairs of nodes (in the case of a 2-GNN) as substructures such as cycles are inherently local too. The paper proves that such distance-restricted higher-order GNNs are able to count substructures and in particular distance 2 and a 2-GNN suffice to count 6-cycles. Experiments underline the strong substructure counting power of the suggested method while still achieving reasonable runtimes (in contrast to full 2-GNN).

**Strengths:**

Strong cycle counting power without being extremely slow: finally we have an efficient GNN that can not only count triangles (i have seen a few of that) but the much more complicated (and possibly interesting) 6-cycles. This holds both in theory and in practice. The authors provide a clear theory (I have to admit that I did not check the proofs in the appendix) and extensive experiments focusing on the main claim, namely the strong cycle and in general substructure counting power. In terms of efficiency I was surprised that the practical memory consumption is not far from what a standard MPGNN uses (even though in the molecular domain the maximum degree in each graph is typically 3 or 4 and thus not extremely large). Overall a very nice paper!

**Weaknesses:**

I don't have strong criticisms about the paper (but please see the questions below). Here is a list of things I noted while reading the paper, roughly in the order of appearance:
- 14: How is subgraph extraction avoided when using DRFWL? The DR-k-FWL procedure somehow needs to operate on small induced subgraphs.
- 48: I would say it is not obvious why 2-FWL encodes closed walks. It would be nice to have some additional intuition there
- 62: Many graphs have small diameter, for those the distance restriction does not help too much. And in particular, on the complete graph the distance restriction trivially is useless. The "deg" dependency is only introduced in the next paragraph such that the statement that DRFWL(2) should have a lower complexity than 2-FWL is unclear as this obviously does not hold for the worst-case complexity.
- the related work might also include methods that break the 1-WL barrier by aggregating from multiple distances at once (those methods often call themselves "higher-order" in the sense that they use a "higher-order neighborhood", often 2 or 3 hops)
- At Q4: please make explicit that this is a question to be answered empirically (the theory result is already there)
- How about long-range dependencies? (https://arxiv.org/abs/2206.08164)
- It would be nice to mention that the model is not suitable for node classification as the corresponding graphs typically have relatively large degrees (at least compared to molecules).


Typography:
- Leman asked to be written without the h. This is why some older papers still use the additional h while most newer ones do not.
- It might look nicer to use \bm for the $d$ in section headings
- I personally prefer it when equations are referenced as Eq. (6) instead of just (6). One way is to use \cref throughout the document
- 173: the "one the other hand" seems out of context
- It would be nice to have author names in addition to a citation [99] when [99] acts as the subject of that sentence. This happens several times in the related work section
- 51: := \ell -> =: \ell (as \ell is to be defined)
- 296: Similar -> A similar
- Thanks a lot for proper proofreading!

**Questions:**

- There seems to be a subtle difference between equations 3,4 and equations 6,7. In particular 6,7 explicitly create $M_{i,j}$ for every distance pair $i,j$ while equation 3,4 just create a single multiset of sift operations (colors). Would you like to comment on that small difference? I have the feeling that this change may make the distance-restricted method (with d>diameter) stronger than pure FWL, although your theorem states that this is not the case - effectively the DR variant seems to individualizes one node.

- is 2-DRFWL(2) well suited for long-range dependencies? https://arxiv.org/abs/2206.08164
- How does it perform on molpcba (which is more commonly evaluated against than molhiv)?
- Is there a simple intuitive reason why FWL(2) cannot count more than 7-cycles?

**Limitations:**

The authors addressed the main limitation of the work, namely that it becomes a lot slower when the degree in a graph increases, limiting the applicability to mostly the molecular domain and keeping it from working in the context of node classification.

---

> ### Author Rebuttal · Authors · 2023-08-10
>
> Thank you for acknowledging our contributions as well as providing the insightful suggestions.
>
> **Reply to W1.**
>
> $d$-DRFWL(2) GNNs process a graph G in two steps: (i) extract all 2-tuples $(u,v)\in\mathcal{V}_G^2$ that satisfy $d(u,v)\leqslant d$; (ii) initialize and update the embeddings of all such 2-tuples using the update rules (equations 10 and 11 of the paper).
>
> In the above procedure, the message passing between 2-tuples takes place directly on the entire graph G, without the need to extract subgraphs from G and view G as a "bag of subgraphs". Moreover, messages from any 2-tuple (u, v) will be propagated to the entire graph after a sufficiently large number of iterations. In both senses DRFWL(2) GNNs avoid extracting subgraphs.
>
> **Reply to W2.**
>
> Briefly speaking, the update rule of FWL(2) allows each 2-tuple (u,v) to be aware of the information from another two 2-tuples (u,w) and (w,v), for an arbitrary w. Now, let us assume that we have *somehow* trained an FWL(2) GNN such that for every 2-tuple (u,v), its embedding contains $\left(W_{d_1}(u,v), W_{d_2}(u,v), W_{d_3}(u,v)\right)$, where $W_k(u,v)$ means the number of k-walks starting from u and ending at v. Then, the update rule of FWL(2) would allow the following value to be calculated:
>
> $$W_{d_1}(u,v)\cdot\sum_{w\in\mathcal{V}_ G} W_{d_2}(u,w)W_{d_3}(w,v)$$
>
> If further summed over different $(d_1, d_2, d_3)$ combinations with fixed $l=d_1+d_2+d_3$, this yields the number of closed $l$-walks that pass u and v. In this sense, FWL(2) can encode closed walks, as long as information of shorter walks can be obtained. Since shorter walks are easy to capture by a few FWL(2) updates, our claim holds.
>
> **Reply to W3.**
>
> We acknowledge that for graphs with small diameters (i.e., diameter $\lesssim d$), the distance restriction of $d$-DRFWL(2) helps little in reducing computational cost. However, such graphs either have few nodes or are very dense (e.g. complete graphs). For the first case, the efficiency is usually not a major concern. The second case hardly happens since real-world graphs are usually sparse. We will explicitly state our assumption on sparsity in our revision, making the "deg" dependency clearer.
>
> **Reply to W4.**
>
> Thanks for the suggestion. We include a discussion on those methods here. As proposed in [a], such methods are often called K-hop message passing GNNs (K-hop MPNNs). Like our method, K-hop MPNNs also explicitly leverage distances to control message passing, thus making use of the sparsity of the graphs. Their discriminating power is also between WL(1) and FWL(2).
>
> However, a major deficiency of K-hop MPNNs is that messages are passed between **nodes** instead of **node pairs**. As is discussed in our paper, such kind of message passing is **incapable of detecting substructures, especially cycles**. Indeed, as shown in Figure 1 of [a], the pair of graphs in either Example 1 or Example 2 contain different numbers of 3-cycles; however, K-hop MPNNs with either shortest path distance kernel or graph diffusion kernel fails to distinguish between both graph pairs, implying that vanilla K-hop MPNNs **fail to graph-level count even the simplest 3-cycles**. In contrast, with d=2, d-DRFWL(2) GNNs can already node-level count up to 6-cycles.
>
> [a] Feng, J. & Chen, Y. & Li, F. & Sarkar, A. & Zhang, M. (2022). How Powerful are K-hop Message Passing Graph Neural Networks. 10.48550/arXiv.2205.13328.
>
> **Reply to W5 & W7.**
>
> Thanks for the suggestions. We will clarify both points in our revision.
>
> **Reply to W6.**
>
> Thanks for the question. We evaluate 2-DRFWL(2) GNN on the two graph-level property prediction datasets, `Peptides-func` and `Peptides-struct` of Long Range Graph Benchmarks (LRGB). For experiments on both datasets, we use 5 2-DRFWL(2) GNN layers with hidden size 120. We train 200 epochs using Adam optimizer with initial learning rate 0.001, and plateau scheduler with patience 10, decay rate 0.5, and minimal learning rate 1e-5. The batch size is 64. The results are shown in Table 4 of the PDF.
>
> From the table, we see that our method outperforms almost all message passing GNNs. Surprisingly, on `Peptides-struct` the performance of our model is **even comparable to Transformer-based models** (which inherently capture long-range information). Therefore, we conclude that although designed to capture local structural information, our model still learns long-range dependencies well.
>
> **Reply to Typography.**
>
> Thanks a lot for the meticulous checking! We will update them all in the revision.
>
> **Reply to Questions.**
>
> Due to space limit, we only provide brief responses to your questions. We are willing to append the full responses during the author-reviewer discussion.
>
> *Q1:* The fact is that with d > diameter, d-DRFWL(2) is **as powerful as** the FWL(2) test. The key is that FWL(2) can also calculate the distance between every node pair $(u,v)$, after which it can store the distance as an additional marking for the 2-tuple. Now it is easier to see the equivalence with equations 6,7, as long as d > diameter.
>
> *Q2:* Please see **W6**.
>
> *Q3:* The result of 2-DRFWL(2) GNN on molpcba is as follows.
>
> |Method | ogbg-molpcba (AP)|
> |:---:| :---:|
> | GIN+virtual node | 0.2703 $\pm$ 0.0023|
> | Nested GIN+virtual node | **0.2832** $\pm$ 0.0041 |
> | 2-DRFWL(2) GNN (ours) | 0.2076 $\pm$ 0.0026 |
>
> We see that our model performs poorly on ogbg-molpcba, which we have no proper explanation currently. While we will keep on fine-tuning the hyperparameters. We point out that none of the known higher-order GNNs (or their sparsified variants) have ever appeared on the leaderboard of ogbg-molpcba, indicating that ogbg-molpcba might be particularly hard to optimize or generalize for higher-order GNNs (including ours).
>
> *Q4:* The simplest answer is that Shrikhande graph and `4*4` Rook's graph (a well-known pair of FWL(2) inseparable graphs) have different numbers of 8~16 cycles. The intuitive explanation is deferred to author-reviewer discussion.

---

> > ### Author Response · Authors · 2023-08-12
> > **Author Response to Reviewer 9V2p, continued**
> >
> > Due to space limit, only brief responses are given to the questions posed by Reviewer 9V2p. We now append the full responses to those questions below.
> >
> > **Reply to Q1.**
> >
> > The fact is that with d greater than the diameter, d-DRFWL(2) is as powerful as the FWL(2) test. The key point is that FWL(2) can also individually deal with different distance pairs $i,j$, as in equations 6, 7. As stated in lines 174--176 of our paper, by a few FWL(2) iterations one can calculate the distance $d(u,v)$ between every pair of nodes $(u,v)$. Due to this fact, we can design an FWL(2) instance, such that it first computes distance between every pair of nodes, and then stores the distance $d(u,v)$ as an additional marking for the 2-tuple $(u,v)$. After executing the above operations, the contributions from each distance pairs $i,j$ are naturally disjoint, because $i,j$ has been labeled to $W^{(t-1)}(u,w), W^{(t-1)}(w,v)$ respectively. This way, it is easier to see the equivalence with equations 6, 7, as long as $d>$ diameter.
> >
> > **Reply to Q2.**
> >
> > See our response to **W6**.
> >
> > **Reply to Q3.**
> >
> > Thanks for the question. We evaluate 2-DRFWL(2) GNN on molpcba. We tune the hyperparameters by randomly searching the following space with four trials: (i) hidden size $\in[100,300]$; (ii) number of layers $\in[3,5]$; (iii) learning rate $\in[0.001, 0.008]$; (iv) dropout rate $\in\{0.2,0.5\}$; (v) plateau scheduler patience $\in\{10,20\}$; (vi) plateau scheduler decay $\in\{0.5, 0.8\}$. We also apply layer normalization after each 2-DRFWL(2) GNN layer. We train for 150 epochs with batch size 256. The best 4-runs result among all trials is already presented in our rebuttal.
> >
> > Our model performs poorly on ogbg-molpcba, which we have no proper explanation currently. While we will keep on fine-tuning the hyperparameters, we point out that none of the known higher-order GNNs (or their sparsified variants) have ever appeared on the leaderboard of ogbg-molpcba, indicating that ogbg-molpcba might be particularly hard to optimize or generalize on for higher-order GNNs (including ours).
> >
> > **Reply to Q4.**
> >
> > Thanks for the question. The simplest answer is that Shrikhande graph and `4*4` Rook's graph (a well-known pair of FWL(2) inseparable graphs) have different numbers of 8~16 cycles. The counterexamples for longer cycles are constructed in [a]. However, this proof lacks intuition.
> >
> > Nevertheless, intuitive explanation may be given for why our technique in proving the ability of FWL(2) to count 6 or 7-cycles fails on 8-cycles. We notice that to count a longer cycle, we always first split it into two distinct paths whose counts we can obtain. For example, a 6-cycle is splitted into a 2-path and a 4-path, with 2, 4-paths both known countable by FWL(2). Now if we want to count 8-cycles, we obviously should split an 8-cycle into two 4-paths. The next step is to enumerate all cases in which some of the nodes on the two paths coincide. For convenience, say the two 4-paths both start from u and end at v. Consider a coinciding case where the two 4-paths are of the form u->x->y->z->v and u->x->b->c->v. To count such case, one must count 6-cycles like x->y->z->v->c->b->x for **any** pair of x and v. Performing this count further requires the number of x->y->z->v->y->z->x, which appears as a coinciding case for counting the 6-cycles for x and v. However, this is generally **not countable by FWL(2)**, since (x,v) not only needs to know (x,y), (y,v) or (x,z), (z,v), but also has to assure that y and z are neighbors. We believe similar intuitions may be provided for longer cycles.
> >
> > [a] V. Arvind, F. Fuhlbrück, J. Köbler, and O. Verbitsky. On weisfeiler-leman invariance: Subgraph counts and related graph properties. Journal of Computer and System Sciences,

---

> > > ### Comment · Reviewer_9V2p · 2023-08-12
> > >
> > > I thank the authors for their very detailed responses, both to my questions as well as the ones raised by the other reviewers. Also the additional theoretical and empirical analyses about increasing $d$ will make the paper even stronger. I would like it if some of those explanations (e.g. about the difference between the equations 3/4 and 6/7 or the explanation on general cycle counting of FWL(2)) would make it into the main paper or at least into the appendix.
> > >
> > > I also have no idea why the presented method might work better on MolHIV than MolPCBA, although I would very much be interested in a reason for that.
> > >
> > > So thanks again for thoroughly answering all my open questions!

---

> > > > ### Author Response · Authors · 2023-08-12
> > > >
> > > > We thank the reviewer for diligently reading our responses and giving the valuable suggestions. In our revision, we will include the additional theoretical and empirical studies on $d$-DRFWL(2) with larger $d$ values either in the main paper or in the appendices, since they extend our current results in the paper. Also, brief clarifying remarks will be added to proper positions in our text.
> > > >
> > > > As for the empirical study on ogbg-molpcba, additional efforts will be devoted to tuning our model more extensively and finding explanations for the unexpected performance degradation of our model on ogbg-molpcba. Whatever further advancement will be included in our revision.

---

### Official Review · Reviewer_4wqe · 2023-07-06

**Soundness:** 3 good
**Presentation:** 4 excellent
**Contribution:** 3 good
**Rating:** 6
**Confidence:** 5

**Summary:**

In this paper, the authors introduce the d-DRFWL(2) GNNs to count certain graph substructures, including cycles, cliques, and paths, which is essential for various graph-mining tasks. The authors prove that with d = 2, d-DRFWL(2) GNNs can already count up to 6 cycles, retaining most of the cycle counting power of FWL(2). Since the d-DRFWL(2) GNNs restrict the distance of message propagation, they are much more efficient than other GNN models. Experiments on both synthetic datasets and real molecular datasets are consistent with the theoretical results.

**Strengths:**

S1. The experiment is conducted on both real and synthetic datasets, where the results are convincing.

S2. The theoretical results are sound and easy to follow.

S3. The paper is clear and easy to follow.

**Weaknesses:**

W1. In my opinion, the scope of this particular line of research, which examines the expressive capabilities of GNNs by quantifying subgraph structures, appears to be rather limited. As stated in [1], GNNs are Turing complete (LOCAL algorithm) under certain conditions, indicating that they can effectively address any graph mining task with the same power as traditional algorithms. Consequently, this line of research primarily relies on observations made in previous graph mining algorithms and attempts to reinterpret them in the framework of GNNs, albeit within specific model designs, i.e., data structure for processing the graph. It would be beneficial if the authors could provide a comprehensive explanation detailing how their approach represents an advancement in utilizing GNNs to solve the subgraph counting problem, surpassing conventional algorithms [2].

W2. Although the authors demonstrate that 2-DRFWL(2) can count 6-cycles, I am curious about the ability of 3-DRFWL(2) (or even larger $d$) to solve the cycle counting problem with larger subgraph sizes. If possible, it would be essential to understand the relationship between the minimal distance and the subgraph size. Additionally, exploring the impact of different values of K in the K-FWL test on the expressive power of GNNs would be interesting.

[1] Loukas, A. (2019, September). What graph neural networks cannot learn: depth vs width. In International Conference on Learning Representations.\
[2] Ribeiro, P., Paredes, P., Silva, M. E., Aparicio, D., & Silva, F. (2021). A survey on subgraph counting: concepts, algorithms, and applications to network motifs and graphlets. ACM Computing Surveys (CSUR), 54(2), 1-36.

**Questions:**

Q1. There are other essential subgraph structure such as k-plex and k-core. I wonder whether the proposed framework can count this structures.

Q2. The authors propose the existence of a clear hierarchy of expressiveness for d-DRFWL(2) GNNs as the value of $d$ increases. However, no experiments have been conducted to verify this claim across different values of $d$. Therefore, I would like to suggest the authors include a sensitivity test to support their theoretical findings.





**Limitations:**

Please check the weaknesses and questions.

---

> ### Author Rebuttal · Authors · 2023-08-09
>
> Thanks for your insightful comments and suggestions. We respond to all the weaknesses and questions below.
>
> **Reply to W1.**
>
> We thank the reviewer for providing relevant works [1][2]. Although [1] claims that message passing GNNs (MP-GNNs) can be Turing complete, the assertion is made under fairly strong assumptions, such as requiring **a unique identity labeling given to each node**, which explicitly breaks the permutation symmetry. As explained in Appendix D of [1], once we remove the node identity labeling, MP-GNNs are no longer Turing universal. Actually, MP-GNNs without node identity labeling are no more powerful than the WL(1) test (as stated in [a]), and cannot count any cycles. This fact indicates that they fail to **causally** recognize graph substructures.
>
> In many real-world tasks, such as molecular property prediction, it is impossible to assign unique labels to nodes (for example, it is hard to consistently determine which node should be labeled as "0"-th in a benzene ring); instead, only models that respect permutation symmetry are considered. As is verified by our theory and experiments, with $d=2$, $d$-DRFWL(2) GNNs do have an outstanding substructure counting power compared with other permutation-symmetric models. There exist works [b, c] that assign unique random features/labels to nodes (breaking permutation symmetry), but they all have bad generalization performance on large real-world datasets since even two isomorphic graphs may have different representations using random features.
>
> [2] surveys traditional algorithms addressing the substructure counting problem. Different from our method, they are generally not data-driven. As we are to point out below, these traditional methods are **not directly comparable** with our method. This is because the ultimate goal of designing more powerful GNNs is to **tackle the traditionally intractable tasks in a data-driven manner**. Therefore, those GNN models (including ours) are not tailored to **solely** solving the substructure counting problem. Indeed, if certain substructure counts (or known functions of them) are the only targets, one should always apply traditional approaches instead of training a GNN, since traditional approaches avoid expensive training and have better accuracy and explainability. However, if the task is intractable by traditional approaches (e.g. molecular property prediction), then GNNs with strong cycle counting power (like ours) can work well. Therefore, the traditional methods and our model **differ in their domains of application**, and should not be directly compared.
>
> Despite the above discussion, substructure counting power remains a practical metric to evaluate a GNN's expressiveness, as stated in the 1st point in our general response. It is from this perspective should our main contribution (proposing efficient GNNs with provable cycle counting power) be understood.
>
> [a] K. Xu, W. Hu, J. Leskovec, and S. Jegelka. How powerful are graph neural networks? In ICLR, 2018.
>
> [b] Abboud, Ralph, et al. "The surprising power of graph neural networks with random node initialization." arXiv preprint arXiv:2010.01179 (2020).
>
> [c] Sato, Ryoma, Makoto Yamada, and Hisashi Kashima. "Random features strengthen graph neural networks." SDM 2021.
>
> **Reply to W2.**
>
> Thanks for suggestion. As stated in the 2nd point of our general response, we have proved that increasing d from 2 to 3 makes d-DRFWL(2) able to count 7-cycles. However, further increasing d to $d\geqslant 4$ does not bring about stronger cycle counting power since even FWL(2) cannot graph-level count more-than-8-cycles.
>
> As for the distance-restricted version of FWL(k) with k>2, the meaning of "mutual distance" within k-tuples (k>2) is somewhat opaque. If the "mutual distance" within a k-tuple is defined as the maximal distance between nodes in it, then we believe that the resulting d-DRFWL(k) will form a similar expressiveness hierarchy with increasing d; moreover, their expressive power is similarly upper-bounded by FWL(k). However, since the cycle counting power of FWL(k) with k>2 is still open, the cycle counting power of d-DRFWL(k) defined above remains open.
>
> **Reply to Q1.**
>
> Thanks for raising the valuable questions. We first clarify the definitions of k-plexes and k-cores as following:
>
> * A k-plex in G is an induced subgraph of G in which every node has degree at least $|S|-k$, where S is the node set of the subgraph
> * A k-core in G is an induced subgraph of G in which every node has degree at least k
>
> For k-plexes, it is an NP-complete problem even to find the maximal k-plex of a graph G. [a] Therefore, it is generally impossible for our model to count k-plexes. Indeed, if k=1, as 1-plexes are simply cliques, our model fails to count 1-plexes with size larger than 4.
>
> For k-cores, notice that a k-core can have arbitrarily large diameter, which may make it "beyond the reach" of d-DRFWL(2) GNN. For example, if k=2, then any induced cycle of G is a 2-core in G. Taking the example of **G being two $(3d+1)$-cycles and H being a single $(6d+2)$-cycle** (between which d-DRFWL(2) GNNs fail to discriminate), we see that G has two 2-cores while H has one. Therefore, d-DRFWL(2) GNNs cannot count 2-cores for any d. We believe that counterexamples for larger k can be similarly constructed, by considering other regular graph pairs with large diameters.
>
> Summarizing the above discussion, the answer to **Q1** is generally negative.
>
> [a] Balasundaram, Balabhaskar & Butenko, Sergiy & Hicks, Illya V. Hicks. (2011) Clique Relaxations in Social Network Analysis: The Maximum k-Plex Problem. Operations Research. 59. 133-142. 10.1287/opre.1100.0851.
>
> **Reply to Q2.**
>
> Thanks for the constructive suggestion. We have included in the 2nd point of the "general response to all reviewers" a thorough experimental study of $d$-DRFWL(2) GNNs with two $d$ values: $d=2$ and $d=3$. The experimental results should have provided strong support to our theory.

---

> > ### Comment · Reviewer_4wqe · 2023-08-11
> >
> > I think the authors have clarified most of my concerns. I would be happy to adjust my rating.

---

> > > ### Author Response · Authors · 2023-08-12
> > >
> > > We thank the reviewer for diligently reading our responses and giving the affirmative feedbacks.

---

### Official Review · Reviewer_8bs4 · 2023-07-20

**Soundness:** 3 good
**Presentation:** 3 good
**Contribution:** 3 good
**Rating:** 6
**Confidence:** 4

**Summary:**

This work explores the ability of graph neural network to count certain graph substructures, especially cycles. While past works counts by collecting subgraphs, the work avoids such burdensome procedure and construct a local method. Theoretical analysis are presented to show that Folklore Weisfeiler-Lehman test. Accordingly, the work proposes d-Distance-Restricted FWL(2) GNNs (FWL(2) being 2-dimensional Folklore Weisfeiler-Lehman), which incorporates the Folklore Weisfeiler-Lehman test algorithm into the aggregation steps of a GNN. Subsequent theoretical analysis and experiemental evaluations of d-Distance-Restricted FWL(2) GNNs are provided. In particular, experiment results show that the proposed GNN shows superior results on counting cycles or other structures (in synthetic datasets) and better performance graph regression task on chemical datasets, as well as being empirically more efficient.

**Strengths:**

- The problem of analyzing the ability of GNN to count substructures on a graph is interesting. Intuitively, counting cycles in a graph is non-intuitive for node-centric algorithm, where the operations solely performed on each node which only receives information from its neighbors at each iteration.
- In light of above, the observation that counting cycles remains intrinsically \emph{local} is a good and noteworthy observation.
- The technical content is good and supported by the experiments.

**Weaknesses:**

### Neglected baseline comparisons with other higher-order GNNs
As explained by the author, [35] and [46] presents GNNs utilizing node tuples for better expressivity. As the adaptation of FWL(2) is the main component of this work, it is concerning that [35] and [46] are not included in the baseline comparisons. Since FWL is adopted into GNN to improve over sugraph GNNs, other (representative) works on higher-order GNNs should also be compared.

As it stands, there is no information in the paper on how this work compares to other higher-order GNNs. Does the proposed work have better performance and better efficiency? Does the proposed work trade-off performance for efficiency and space? We do not know.

In continuation of above, as [35] is a highly cited paper, further searching founds [a, b, c] that also mentions cycle counting in their paper, but is not discussed or mentioned in this work.
- [a] A New Perspective on "How Graph Neural Networks Go Beyond Weisfeiler-Lehman?" ICLR 2022
- [b] A Practical, Progressively-Expressive GNN. NeurIPS 2022
- [c] Graph Neural Networks with Local Graph Parameters. NeurIPS 2021


### Unanswered questions: ablation and trade-off for larger d
While the work presents d-DRFWL(2) GNNs, the theoretical analysis and experiments are all limited to d=2. Furthermore, Q4 is not satisfyingly answered, as the experiment results are very close. Why not use a sufficiently large d to represent the FWL(2) test?

### There are some places that are unclear and could be improved by providing more explanation:
- FWL is left unexplained in the abstract. The paper may improve its readability by briefly stating FWL is a 2-tuple coloring scheme that solves graph isomorphism tests, whereas this work proposes to limit the scope of 2-tuples from all node pairs to only those within d-distance.
- (line 108), "divided by a factor only depending on S." Isn't the factor simply the number of nodes in S?
- The syntax of using double curly brakets {{ }} for multisets should be noted when first appeared in Equation (1)
- (line 125) the meaning of rightarrow with a bar ($\mapsto$) is not clear.
- The meaning (or depiction) of the twelve targets of QM9 dataset is not provided in the main paper or the appendix


**Questions:**

Can experiments be provided for comparison between the proposed method with [35] and [46] for performance and empirical efficiency?

Can experiments and analysis be provided for $d\geq 2$?

Can intuitive explanations be provided for each proof and concepts (currently, only 6-cyle is provided with an intuition)?
- For example, it seems that the expressivity of up to 4-path and 6-cycle is correlated, as the furtherest node from a certain node in a 6-cycle requires a 4-path back get back to the certain node. Besides, is the 4-path attained by a pair of 2-tuples?




**Limitations:**

It is provided.

---

> ### Author Rebuttal · Authors · 2023-08-10
>
> Thank you for your insightful comments and thorough investigation of additional related works. We respond to all the weaknesses and questions below.
>
> **Reply to W1.**
>
> Thanks for introducing the related works and offering the suggestion. To compare the performance and efficiency of our model with the ones proposed in [35], we implement GNNs based on **SSWL, SSWL+, LFWL(2) and SLFWL(2)** (all of which are proposed in [35]), and run them on four real-world datasets (in the order of increasing graph size): (i) QM9, (ii) ogbg-molhiv, (iii) HomologyTAPE and (iv) ProteinsDB. Details of the datasets are already present in our paper.
>
> For all experiments, we use a dense $B\times d\times n\times n$ matrix to store the embeddings of all 2-tuples within a batch, where $B$ is the batch size, $d$ is the size of the embedding dimension, and $n$ is the maximal number of nodes among all graphs in the batch.
>
> On QM9 dataset, we use 5 layers with hidden size 64 for each model. We train for 400 epochs using Adam optimizer with initial learning rate 0.001, and plateau scheduler with patience 10, decay factor 0.9 and minimum learning rate 1e-5. The batch size is 64.
>
> Due to time and resource limitation, we compare the four methods with ours only on the first target (i.e., dipole moment $\mu$). The result is shown below.
>
> |Method| QM9 ( $\mathrm{MAE}_\mu$ ) |
> |:---:|:---:|
> |SSWL|0.438|
> |SSWL+|0.421|
> |LFWL(2)|0.439|
> |SLFWL(2)|0.435 |
> |2-DRFWL(2) (ours)|**0.346**|
>
> We see that on target $\mu$, our model outperforms all four methods proposed in [35]. We also compare the efficiency of our model with the four methods. The used metrics follow our paper. The results are shown below.
>
> |Method | Memory (GB) | Preprocess (s) | Train (s/epoch)|
> |:---:|:---:|:---:|:---:|
> |SSWL|3.97|  64|88.5|
> |SSWL+|4.39|64|86.4|
> |LFWL(2)|3.17|64|58.0|
> |SLFWL(2)|3.97|64|82.8|
> |2-DRFWL(2) (ours)|2.31|430|141.9|
>
> Our method has the best memory efficiency compared with all four methods proposed in [35]. However, both the preprocessing time and training time of our model are longer. This is because
> * The four methods proposed in [35] do not need preprocessing, but instead build dense representation matrices on the fly.
> * The graphs in QM9 are rather small (~18 nodes per graph). On graphs of such small scale, the advantage of our model's lower time complexity are greatly offset by the constant overhead brought about by scatter operation on sparse matrices; in contrast, the four methods proposed in [35] use dense matrix multiplication and `1*1` convolution, which are more optimized operations on GPU
>
> On ogbg-molhiv, HomologyTAPE and ProteinsDB, with the same hyperparameters as those chosen in Appendix E.3 & E.4, all of the four methods run out of GPU memory, indicating their inferior memory efficiency on large graphs.
>
> In summary, our experimental studies show that 2-DRFWL(2) GNNs usually have better efficiency and scalability than GNNs based on SSWL, SSWL+, LFWL(2) or SLFWL(2) except on fairly small graphs. Additionally, it is often easier for 2-DRFWL(2) GNNs to capture the inductive bias suitable for molecular tasks.
>
> Next, we analyze the methods proposed in [46]. [46] proposes $\delta$-$k$-GNN and $\delta$-$k$-LGNN, which are also among the class of sparsified higher-order GNNs. For the case of $k=2$, both $\delta$-2-GNN and $\delta$-2-LGNN are encompassed by the framework of SSWL-GNN in [35]. This is because $\delta$-2-GNN uses aggregation schemes $agg_{uv}^P$, $agg_u^L$, $agg_v^L$, $agg_u^G$ and $agg_v^G$, while $\delta$-2-LGNN uses $agg_{uv}^P$, $agg_u^L$ and $agg_v^L$. Therefore, **the study of $\delta$-2-GNN and $\delta$-2-LGNN can be subsumed into our above study of SSWL-GNN**. We do not discuss the case of larger k since both $\delta$-k-GNN and $\delta$-k-LGNN would be overly expensive.
>
> The above experiments and analysis should address the questions concerning a comparison with other higher-order GNNs, since [35] and [46] are among the most popular methods within this framework.
>
> As for [a, b, c], we include a discussion below.
>
> * [a] proposes GraphSNN, which uses heuristics based on 1-hop subgraphs around nodes as well as their overlap subgraphs to enhance message passing. By its construction, GraphSNN can count 3-cycles, but fails to capture cycles longer than 4.
>
> * [b] proposes ($k$,$c$)($\leqslant$)-SetGNNs, which greatly reduce the effective number of $k$-tuples in WL($k$) by (i) removing node order, (ii) removing duplicate nodes, and (iii) restricting the number of connected components. Like $d$-DRFWL(2), this work leverages **locality** to reduce storage. Nevertheless, there is **currently no theoretical guarantee** on their cycle counting power.
>
> * [c] proposes $\mathcal{F}$-MPNNs, augmenting message passing GNNs with homomorphism counts. This method resembles GSN [ref1], and inherently encodes substructure counts. However, it is rather different from our method since it uses hand-crafted features.
>
> [ref1] G. Bouritsas, F. Frasca, S. Zafeiriou, and M. M. Bronstein. Improving graph neural network expressivity via subgraph isomorphism counting. IEEE Transactions on Pattern Analysis and Machine Intelligence, 45(1):657–668, 2022.
>
> **Reply for W2.**
>
> Thanks for the constructive advice. We have included a comprehensive study on larger d in the 2nd point of our general response.
>
> Due to space limit, we have to defer our response to the other weaknesses/questions in the author-reviewer discussion phase. We are sorry for the inconvenience, and will add the remaining response immediately when the discussion period begins.

---

> ### Author Response · Authors · 2023-08-11
> **Author Response to Reviewer 8bs4, continued**
>
> We now respond to the other weaknesses/questions posed by Reviewer 8bs4. Those responses are not included in the our rebuttal due to space limit.
>
> **Reply for W3, point 1**
>
> Thanks for suggestion. We will add a brief introduction in our revision.
>
> **Reply for W3, point 2**
>
> Not always, since for some substructures S in which not all nodes are structurally equal (such as tailed triangles or chordal cycles), our definitions for C(S,u,G) (see Definitions 4.1 and 4.2) require the node u to lie only at some special position(s) in S. In such cases, the factor is the number of nodes at "legal positions" in S.
>
> **Reply for W3, point 3, 4, 5**
>
> Thanks for suggestion. We will add annotations to clarify these points in our revision. By the way, one can refer to, e.g. the PyTorch Geometric documentation of QM9 dataset for the meaning of the twelve targets.
>
> **Reply for Q1.**
>
> See our response to **W1**.
>
> **Reply for Q2.**
>
> See our response to **W2**.
>
> **Reply for Q3.**
>
> The observation made in the bullet point is correct. Intuitively, in FWL(2) (or its distance-restricted version) if there is a $d_1$ walk from u to v, a $d_2$-walk from u to w and a $d_3$-walk from w to v, then the algorithm can detect a closed $(d_1+d_2+d_3)$-walk (and also a $(d_2+d_3)$-walk) that passes $u$ and $v$, since the update rule concatenates $(u,w)$ and $(w,v)$. This intuition works for all our theorems regarding path and cycle counts. For example,
>
> * **Counting 2-paths**: Combining two 1-walk
> * **Counting 3-paths**: Combining a 2-walk and a 1-walk
> * **Counting 4-paths**: Combining two 2-walks
> * **Counting 3, 4, 5-cycles**: These tasks reduce to counting 2, 3, 4-paths.
> * **Counting 6-cycles**: Combining three 2-walks

---

> > ### Comment · Reviewer_8bs4 · 2023-08-18
> >
> > Thank you. I am satisfied with your explanations and have raised the score to weak accept.

---

> > > ### Author Response · Authors · 2023-08-18
> > >
> > > We thank the reviewer for thoroughly reading our responses and giving the affirmative feedbacks. By the way, we now provide **full** QM9 results (on all 12 tasks) of GNNs based on SSWL, SSWL+, LFWL(2) and SLFWL(2). For each model, we consistently use 5 layers with hidden dimension size 64, and apply layer normalization after each layer. We train for 400 epochs using Adam optimizer. The initial learning rate is searched from {0.001, 0.002}. We also use plateau scheduler with patience 10, decay factor 0.9 and minimum learning rate 1e-5. The batch size is 64. We report the best MAE (mean absolute error) result for each task.
> > >
> > > |Target|SSWL|SSWL+|LFWL(2) |SLFWL(2) | 2-DRFWL(2) (ours)|
> > > |:---:|:---:|:---:|:---:|:---:|:---:|
> > > |$\mu$|0.438| 0.418 |0.439|0.435|**0.346**|
> > > |$\alpha$|0.294|0.271|0.315|0.289|**0.222**|
> > > |$\varepsilon_\mathrm{homo}$|0.00302|0.00298|0.00332|0.00308|**0.00226**|
> > > |$\varepsilon_\mathrm{lumo}$|0.00318|0.00291|0.00332|0.00322|**0.00225**|
> > > |$\Delta\varepsilon$|0.00427|0.00414|0.00455|0.00447|**0.00324**|
> > > |$R^2$|19.31|18.36|19.10|18.80|**15.04**|
> > > |$\mathrm{ZPVE}$|0.00021|0.00020|0.00022|0.00020|**0.00017**|
> > > |$U_0$|0.151|0.110|0.144|**0.083**|0.156|
> > > |$U$|0.163|**0.106**|0.143|0.121|0.153|
> > > |$H$|0.143|**0.120**|0.164|0.124|0.145|
> > > |$G$|0.158|0.115|0.164|**0.103**|0.156|
> > > |$C_v$|0.1138|0.1083|0.1192|0.1167|**0.0901**|
> > >
> > > We see that 2-DRFWL(2) GNN greatly outperforms SSWL, SSWL+, LFWL(2) and SLFWL(2) on all tasks **except $U_0,U,H$ and $G$**. On those four tasks, the performance of SSWL+ and SLFWL(2) is much better.
> > >
> > > We may give an explanation to the phenomenon with some knowledge of the underlying physics. All four targets, $U_0,U,H$ and $G$, are macroscopic thermodynamic properties of molecules. For such properties, inter-molecular interactions (such as hydrogen bonds), as well as interactions between molecules and the environment can be as important as interactions within molecules. Such interactions may occur between two atoms that are graph-theoretically distant (for example, between the “head” of a molecule and the “tail” of a nearby molecule).
> > >
> > > Since 2-DRFWL(2) GNN only keeps embeddings for node pairs $(u,v)$ with $d(u,v)\leqslant 2$, it is harder for it to capture such long-range interactions between distant nodes. In contrast, SSWL, SSWL+, LFWL(2) and SLFWL(2) keep an embedding for every node pair $(u,v)$, no matter how far $u$ is from $v$. This makes them easier to learn long-range interactions, compared with 2-DRFWL(2) GNN.
> > >
> > > We also remark that as shown in Table 3 of our main paper, subgraph GNNs like NGNN and I$^2$-GNN perform even worse than 2-DRFWL(2) GNN on the targets $U_0,U,H$ and $G$. This may be because **the $k$-hop subgraph extraction procedure in subgraph GNNs greatly inhibits the propagation of information between distant nodes**. In subgraph GNNs, a node is even ignorant of the existence of a far-away node since that node simply does not exist in its $k$-hop subgraph. On the other hand, though, in 2-DRFWL(2) GNN such a far-away node is still perceptible since the receptive field enlarges as we stack more 2-DRFWL(2) GNN layers.
> > >
> > > The above discussion actually shows that our model strikes a balance between **generating fine representations for local substructures** and **capturing long-range interactions**, compared with either subgraph GNNs or the GNNs proposed in [35].

---

### Author Rebuttal · Authors · 2023-08-09

We thank all the reviewers for their insightful feedbacks and constructive suggestions. All the comments have been scrupulously considered, and we will integrate the suggestions into our revised version of the paper. To address the common concerns of the reviewers, below we restate the focus of our paper by answering a related question, and present our additional results obtained during the rebuttal period.

**1. Why to use cycle counting power as a metric for expressiveness of GNNs?**

Many reviewers raise the questions about how $d$-DRFWL(2) GNNs fit into other expressiveness hierarchies, such as $k$-WL [a], $\delta$-$k$-WL [b] and LFWL(2)/SLFWL(2) [c]. Although these questions are meaningful themselves, they deviate from our focus. The major concern of our paper is **to propose efficient GNNs with provably strong cycle counting power**. Therefore, cycle counting power is chosen as a metric to evaluate the expressiveness of GNNs. There are two practical reasons to use this metric instead of (the more widely used) discriminating power:

* Many real-world tasks (such as molecular property prediction) rely heavily on cycle counts. Therefore, GNNs with provable cycle counting power can capture important inductive bias closely related to such tasks, while GNNs that are incapable of counting cycles inevitably fail to make causal predictions on such tasks

* For many of the known expressiveness hierarchies such the WL hierarchy [a], the separation in expressiveness is often based on delicately crafted counterexamples, which provide little insight into the expressiveness exhibited in practical tasks. In contrast, cycle counting power remains a useful metric when we need to evaluate the practical expressiveness of a model on given datasets.

**2. Additional studies on $d$-DRFWL(2) GNNs with $d>2$.**

As pointed out by many reviewers, currently our paper only contains extensive studies on the cycle counting power of 2-DRFWL(2) GNNs. There is no investigation on the cycle counting power of $d$-DRFWL(2) GNNs with larger $d$, nor any experimental comparison between 2-DRFWL(2) GNNs and $d$-DRFWL(2) GNNs with larger $d$ values. During the rebuttal period, we have thoroughly studied, both theoretically and empirically, the discriminating power (related to **Q4** of our paper) and node-level cycle counting power of $d$-DRFWL(2) GNNs, with $d>2$.

Regarding the discriminating power of $d$-DRFWL(2) GNNs with larger $d$ values, we run 3-DRFWL(2) GNN on two synthetic datasets: (i) EXP and (ii) SR25. The result is shown in Table 1 of the PDF. We can see that (i) 3-DRFWL(2) GNN discriminates between **all graph pairs on EXP**, including those not separable by 2-DRFWL(2) GNN, verifying **the strict expressiveness gap between 2-DRFWL(2) GNN and 3-DRFWL(2) GNN**; (ii) 3-DRFWL(2) GNN fails on SR25 dataset with accuracy 6.67%.

Regarding the node-level cycle counting power, we have **fully unveiled** the theoretical cycle counting power for $d$-DRFWL(2) with **arbitrary** $d\geqslant 2$:

* 2-DRFWL(2) GNNs can node-level count 3, 4, 5, 6-cycles but cannot graph-level count 7-cycles (already proved in our paper)
* $d$-DRFWL(2) GNNs with arbitrary $d\geqslant 3$ can node-level count 3, 4, 5, 6, 7-cycles but cannot graph-level count 8-cycles (**newly added**)

A proof sketch of the second bullet point is given below. The negative result follows from the fact that $d$-DRFWL(2) GNNs are less powerful than FWL(2) and that FWL(2) cannot graph-level count 8-cycles [e]. For the positive result, it suffices to prove that 3-DRFWL(2) GNNs can node-level count 7-cycles. To calculate the count of 7-cycles, we first decompose a 7-cycle into a 3-path and a 4-path, both of which can be counted by 3-DRFWL(2) GNNs. Then, we enumerate all possible cases in which some of the nodes in the 3-path and the 4-path coincide. There are 12 cases in total, all of them shown in Figure 1 of the PDF. By examining each kind of substructure, we confirm that all of them can be counted by 3-DRFWL(2) GNNs, leading to the positive result.

To verify our theory, we conduct node-level cycle counting experiments on the Substructure Counting dataset, for both 2-DRFWL(2) GNN and 3-DRFWL(2) GNN. Particularly, we add the task of node-level counting 7-cycles. The result is shown in Table 3 of the PDF. We see that on the task of counting 3, 4, 5 and 6-cycles, 3-DRFWL(2) GNN achieves comparable performance with 2-DRFWL(2) GNN; on the task of counting 7-cycles, 3-DRFWL(2) GNN greatly outperforms 2-DRFWL(2) GNN, validating our theory.

We point out that the fact that 3-DRFWL(2) GNNs can node-level count 7-cycles **confirms the existence of GNNs with the same cycle-counting power as FWL(2) but strictly lower complexity**.

We further investigate the real-world performance of 3-DRFWL(2) GNNs by conducting ablation studies on the ZINC-12K dataset. The result is shown in Table 2 of the PDF. It is worth noticing that by only introducing part of the aggregations related to distance-3 tuples, a great performance gain can be observed.

[a] J. Cai, M. Fürer, and N. Immerman. An optimal lower bound on the number of variables for graph identification. Combinatorica, 12(4):389–410, 1992.

[b] C. Morris, G. Rattan, and P. Mutzel. Weisfeiler and leman go sparse: Towards scalable higher-order graph embeddings. Advances in Neural Information Processing Systems, 33:21824–21840, 2020.

[c] B. Zhang, G. Feng, Y. Du, D. He, and L. Wang. A complete expressiveness hierarchy for subgraph gnns via subgraph weisfeiler-lehman tests. arXiv preprint arXiv:2302.07090, 2023.

[d] Ribeiro, P., Paredes, P., Silva, M. E., Aparicio, D., & Silva, F. (2021). A survey on subgraph counting: concepts, algorithms, and applications to network motifs and graphlets. ACM Computing Surveys (CSUR), 54(2), 1-36.

[e] V. Arvind, F. Fuhlbrück, J. Köbler, and O. Verbitsky. On weisfeiler-leman invariance: Subgraph counts and related graph properties. Journal of Computer and System Sciences, 113:42–59, 2020.

---

### Decision · Program_Chairs · 2023-09-21

**Decision:**

Accept (spotlight)

**Comment:**

The reviewers first raised a few questions, including the methodology and the experimental work., on this paper. The authors properly addressed them during the rebuttal/discussion phase, and the reviewers were finally all positive about the work.
In light of the reviews, discussions, and scores, I am hence recommending this paper be accepted as a spotlight.